# Diffusion-based Annealed Boltzmann Generators : benefits, pitfalls and hopes

**Louis Grenioux**[1,2*]                                                    *lgrenioux@flatironinstitute.org*

**Maxence Noble**[1*]                                                    *maxence.noble@gmail.com*

[1] *CMAP, CNRS, École polytechnique, Institut Polytechnique de Paris, 91120 Palaiseau, France*
[2] *Center for Computational Mathematics, Flatiron Institute, New York, NY, USA*

**Reviewed on OpenReview:** *https://openreview.net/forum?id=la4FDaeIbw*

## Abstract

Sampling configurations at thermodynamic equilibrium is a central challenge in statistical physics. Boltzmann Generators (BGs) address this problem by pairing a generative model with a Monte Carlo (MC) correction scheme, yielding asymptotically consistent samples from an unnormalized target density. However, most existing BGs rely on classic MC mechanisms such as importance sampling, which (i) impose strong constraints on the backbone model (typically requiring exact and efficient likelihood evaluation) and (ii) suffer from severe scalability issues in high-dimensional, multi-modal settings. This work investigates BGs built around annealed Monte Carlo (aMC) schemes, which mitigate the limitations of classic MC by bridging a simple reference distribution to the target through a sequence of intermediate densities. In this context, diffusion models (DMs) are particularly appealing backbones: they are powerful generative models and naturally induce density paths that have been leveraged in prior aMC-based methods. We provide an empirical meta-analysis of this DM-based aMC-BG design choice on controlled yet challenging synthetic benchmarks based on multi-modal Gaussian mixtures, varying inter-mode separation, number of modes, and dimensionality. To disentangle learning effects from inference effects, we first study an idealized setting in which the DM is perfectly learned, and then turn to realistic settings where the DM is trained from data. Even in the idealized regime, we find that standard aMC integrations of DMs that rely only on first-order stochastic denoising kernels systematically fail in the proposed scenarios. In contrast, incorporating second-order denoising kernels can substantially improve performance when the required covariance information is available. Motivated by this gap, we propose an alternative aMC integration based on deterministic first-order transport maps derived from DMs; empirically, this approach consistently outperforms its stochastic first-order counterpart, albeit at increased computational cost. Overall, while results in the perfect-learning regime suggest that exploiting DM-induced dynamics within aMC is a promising route to building effective BGs, our experiments with learned DMs show that DM–aMC combinations still struggle to produce accurate BGs in practice. We attribute this limitation primarily to inaccuracies in DM log-density estimation. Code available at https://github.com/h2o64/dabg.

---

*Both authors contributed equally.

# 1 Introduction

Sampling configurations from the Boltzmann distribution of a system $\pi(x) \propto \exp(-\mathcal{E}(x))$, where $\mathcal{E}(x)$ denotes the potential energy of configuration $x$, is a foundational and long-standing challenge. Reliable access to samples from $\pi$ underpins the estimation of many key observables which, in turn, govern macroscopic behavior. Hence, efficient Boltzmann sampling is central to a broad range of applications, from characterizing biomolecular function and accelerating drug discovery to materials design and the study of complex statistical-physics models (Liu, 2001; Krauth, 2006; Stoltz et al., 2010; Ohno et al., 2018; Frenkel & Smit, 2023).

The core difficulty of sampling stems from the geometry of realistic energy landscapes. In many practical settings, the energy $\mathcal{E}$ is high-dimensional, non-smooth, and highly rugged, with numerous metastable basins (referred to as "modes") separated by high barriers. This structure severely challenges classical simulation-based approaches such as Molecular Dynamics (MD) and Markov Chain Monte Carlo (MCMC), whose generated samples follow dynamics prone to trapping in local minima, thus requiring a computationally prohibitive number of successive steps to mix across modes. The resulting samples are strongly correlated, leading to large statistical inefficiencies.

Boltzmann Generators (BGs) (Noé et al., 2019) address this bottleneck by amortizing sampling cost through training a generative model $p^\theta$ to approximate $\pi$, followed by a correction step that turns proposals from $p^\theta$ into samples from the target $\pi$. Modern BGs predominantly rely on normalizing flows (NFs), either discrete (DNFs) (Rezende & Mohamed, 2015; Papamakarios et al., 2021) or the more expressive continuous variant (CNFs) (Chen et al., 2018; Grathwohl et al., 2019), because they support efficient sampling and (in principle) tractable density evaluations. For NFs, the natural correction mechanism is to embed proposals into Monte Carlo (MC) schemes, most prominently Importance Sampling (IS) (Müller et al., 2019; Noé et al., 2019; Köhler et al., 2020; Klein et al., 2023; Klein & Noé, 2024) and MCMC (Albergo et al., 2019; Gabrié et al., 2022; Del Debbio et al., 2022; Brofos et al., 2022; Samsonov et al., 2022; Cabezas et al., 2024). However, these strategies are highly sensitive to the overlap between $p^\theta$ and $\pi$ (Agapiou et al., 2017; Grenioux et al., 2023): in high dimension or for highly multi-modal targets, even small modeling errors can yield extremely poor correction capabilities. Moreover, in the CNF setting, evaluating $p^\theta(x)$ accurately is itself expensive, as it requires solving a neural ODE.

Recently introduced Diffusion Models (DMs) (Sohl-Dickstein et al., 2015; Ho et al., 2020; Song et al., 2021) are generative models that have achieved state-of-the-art performance across many data modalities (Kong et al., 2021; Ho et al., 2022; Karras et al., 2024; Abramson et al., 2024), and thus provide a natural alternative to NFs as the backbone of Boltzmann generators. We review DMs in detail in Section 2.1. Their core principle is to learn how to remove noise from corrupted samples; training across many noise levels yields a sequential generation procedure that maps pure noise to structured data. While DMs often produce higher-fidelity samples than NFs on complex distributions, their inference mechanism does not integrate directly into classical BG pipelines, most notably because their likelihood is typically not available in a tractable form.

This work reviews and extends approaches that turn DMs into BGs by leveraging annealed Monte Carlo (aMC) methods, introduced in Section 2.2 as refinements of classical MC schemes. The key idea of aMC is to replace a hard sampling problem by a sequence of easier ones, relying on a user-defined path of intermediate densities that bridges a simple base distribution to the target $\pi$. While many such paths are possible, several recent works have shown that DMs suggest a particularly natural construction; we unify and review these strategies in Section 3. Our overarching objective is to address the following question:

*How can Diffusion Models yield accurate and efficient Boltzmann Generators?*

To explore this question, we examine two complementary experimental regimes:

**(A) Idealized regime:** we assume that the DM is perfectly learned, thus isolating the statistical inference errors induced by aMC from errors due to imperfect training;

**(B) Realistic regime:** the DM is trained from biased data, reflecting practical settings.

Our main contributions are the following:

- We present a unified review of existing approaches that integrate Diffusion Models into annealed Monte Carlo to build Boltzmann Generators. These methods exploit the sequence of marginal distributions induced by the DM's denoising process as intermediate densities in aMC. In idealized regime **(A)**, we show that such DM-informed constructions consistently outperform traditional aMC designs.

- We further analyze strategies that leverage the conditional structure of the denoising process, which is naturally available from DMs. In practice, this is achieved through Gaussian approximations of the conditional distributions between consecutive noise levels. We distinguish *first-order* approximations, which match only the conditional mean, from *second-order* approximations, which also incorporate covariance information. In idealized setting **(A)**, we find that first-order approximations offer no improvement over a naive, correlation-free baseline (i.e., using marginal densities alone), despite additional access to exact knowledge of conditional means, whereas second-order approximations yield substantial performance gains.

- We propose a complementary alternative to Gaussian approximations by introducing deterministic transport maps. Importantly, these maps integrate seamlessly into the aMC framework and require only access to the previously mentioned conditional mean. In idealized regime **(A)**, this deterministic approach achieves performance comparable to second-order stochastic methods, at the cost of a small computational overhead, but without requiring covariance estimates.

- In realistic regime **(B)**, where all DM's components are learned from data, we observe a significant performance degradation across all DM-based aMC-BG methods compared to idealized regime **(A)**. Our empirical results indicate that this gap could primarily be due to inaccurate approximations of DM's densities.

Although BGs are often benchmarked on molecular systems, we instead focus on controlled yet challenging Gaussian mixture distributions. These widely used targets enable systematic comparison under precisely controlled levels of difficulty (Grenioux et al., 2025; Noble et al., 2025), and crucially allow exact computation of the quantities required in idealized setting **(A)**.

> **Multi-modal Gaussian distributions under consideration**
>
> We consider: *(i)* the bimodal distribution of Grenioux et al. (2025), denoted *TwoModes*, which allows one to control both the system's dimensionality, denoted $d$, and the separation between imbalanced modes through a parameter $a > 0$ (larger $a$ implies a larger gap); and *(ii)* the multi-modal target of (Noble et al., 2025, Appendix H.1), denoted *ManyModes*, which features a variable number of modes with non-uniform weights. Formal definitions are recalled in Appendix D.1.

For each target family, we select three representative "edge-case" configurations that combine high dimensionality with strong multi-modality, and are therefore particularly challenging. For *TwoModes*, we consider: close modes in high dimension ($a = 1.0$, $d = 128$), distant modes in low dimension ($a = 10.0$, $d = 16$), and an intermediate case ($a = 5.0$, $d = 64$). For *ManyModes*, we use 4, 16, and 64 modes with dimension fixed to 32. In realistic regime **(B)**, we additionally evaluate the diffusion-based BGs on instances of the ManyWell distribution (Noé et al., 2019; Midgley et al., 2023b), which exhibit a substantially more challenging, non-Gaussian energy landscape than the preceding Gaussian mixtures. We consider dimensions 16, 32, and 64, with further details provided in Appendix D.1. To improve numerical stability and avoid target-specific hyperparameter tuning, all targets are standardized to have zero mean and unit covariance.

We evaluate sampling quality across all targets using three complementary metrics. Our primary metric is the Sliced Wasserstein Distance (Bonneel et al., 2015), denoted Sliced $W_2$, a popular choice that balances statistical accuracy and computational cost, computed between weighted generated and ground-truth samples. Following Grenioux et al. (2025); Noble et al. (2025), we also report a mode-weight estimation metric, which assesses whether generated samples populate target modes in the correct proportions (see Appendix D.1). Finally, we estimate the log-normalization constant $\log \mathcal{Z}$ (equal to 0 in our setting) to assess annealed sampling methods. As sliced $W_2$ is the most comprehensive of the three, the main paper reports only this metric, with the others deferred to Appendix D.3.

**Code.** All code required to reproduce the experiments and implement the algorithms presented in this paper is publicly available at `https://github.com/h2o64/dabg`.

**Notation.** For any measurable space $(X, \mathcal{X})$, we denote by $\mathcal{P}(X)$ the space of probability measures defined on $(X, \mathcal{X})$. Unless specified, if X is a topological space, then $\mathcal{X}$ is defined as the Borel $\sigma$-field of X. For simplicity, we use the same notation to refer both to a probability distribution and its density wrt the Lebesgue measure when it is defined. In our paper, $\pi^{\mathrm{base}}$ denotes a simple distribution that is easy to sample from (for instance, Gaussian), and is referred to as the "base" distribution. We denote $N(\mu, \Sigma)$ with $\mu \in \mathbb{R}^d$ and $\Sigma \in S_d^{++}$ the multivariate Gaussian distribution with mean $\mu$ and covariance $\Sigma$. For any Markov kernel $Q : \mathcal{B}(\mathbb{R}^d) \times \mathbb{R}^d \to [0, 1]$, we denote its conditional density $q(y|x) = Q(\mathrm{d}y, x)/\mathrm{d}y$ for any $(x, y) \in \mathbb{R}^d \times \mathbb{R}^d$. Moreover, for any probability distribution $\mu \in \mathcal{P}(\mathbb{R}^d)$, we denote by $\mu Q \in \mathcal{P}(\mathbb{R}^d)$ the distribution obtained by applying the kernel $Q$ to $\mu$, defined by

$$(\mu Q)(\mathrm{d}y) = \int_{\mathbb{R}^d} Q(\mathrm{d}y, x)\mathrm{d}\mu(x) .$$

For ease of reading, we may use the same notation for $Q(y, x)$ and $q(y|x)$ throughout the paper. For any $C^1$-diffeomorphism $T : \mathbb{R}^d \to \mathbb{R}^d$, we denote by $J_T(x)$ the Jacobian matrix of T evaluated at $x$, and by $T_{\#}\mu$ the pushforward of the distribution $\mu$ by $T$. Hence, if $X \sim \mu$, then $T(X) \sim T_{\#}\mu$. By the change-of-variable formula, the density of $T_{\#}\mu$ wrt the Lebesgue measure is given by

$$T_{\#}\mu(x) = \mu(T^{-1}(x)) \left|\det J_{T^{-1}}(x)\right| . \tag{1}$$

## 2 Background

Before detailing existing DM-based aMC-BG methods (Section 3) and presenting our deterministic version (Section 4), we first review the key ingredients that underpin these approaches: diffusion models (Section 2.1) and annealed sampling techniques (Section 2.2). Throughout this section, for both generative and sampling frameworks, $\pi$ and $\pi^{\mathrm{base}}$ will respectively refer to the target and the base distributions.

### 2.1 Diffusion models

**Forward process.** The stochastic "noising" process of DMs that gradually corrupts the data with increasing Gaussian noise is described by a linear SDE of the form

$$\mathrm{d}X_t = f(t)X_t\mathrm{d}t + g(t)\mathrm{d}W_t, \quad X_0 \sim \pi, \quad t \in [0, T] , \tag{2}$$

where $(W_t)_{t \geq 0}$ is a standard Brownian motion, and $f : [0, T] \to \mathbb{R}$ and $g : [0, T] \to (0, \infty)$ are given schedule functions. Marginally, this forward diffusion process can be explicitly defined by

$$X_t \overset{\mathrm{d}}{=} S(t)X_0 + S(t)\sigma(t)Z, \quad X_0 \sim \pi, \quad Z \sim N(0, I_d) , \tag{3}$$

where $S(t) = \exp(\int_0^t f(u)\mathrm{d}u)$ and $\sigma^2(t) = \int_0^t g^2(u)/S^2(u)\mathrm{d}u$. As a result, the marginal density of $X_t$, denoted by $p_t$, is a convolution of $\pi$ with a Gaussian kernel that writes as

$$p_t(x) = \int_{\mathbb{R}^d} N(x; S(t)x_0, S(t)^2\sigma^2(t)I_d)\mathrm{d}\pi(x_0) . \tag{4}$$

With an appropriate choice of schedules $f$ and $g$ (or equivalently, $S$ and $\sigma$), the forward process interpolates between $p_0 = \pi$ and $p_T = \pi^{\mathrm{base}}$, where $\pi^{\mathrm{base}}$ is a Gaussian distribution independent of $\pi$. We refer to Song et al. (2021) and Karras et al. (2022) for a detailed presentation of commonly chosen noising schemes. In Appendix B, we detail computations related to the widely used Variance Preserving (VP) and Variance Exploding (VE) settings. In practice, the integral in (4) generally cannot be computed in closed form, rendering the marginal density $p_t$ intractable for an arbitrary target distribution $\pi$.

**Backward process.** To generate new data, the idea is to reverse time in SDE (2) so as to denoise samples from $\pi^{\mathrm{base}}$ into samples from $\pi$. Under mild regularity conditions on $f$, $g$, and $\pi$, it can be shown (Anderson, 1982) that the reverse-time dynamics of the noising SDE is itself governed by another SDE, commonly referred to as the reverse-time or denoising SDE

$$\mathrm{d}X_t = \left[f(t)X_t - g^2(t)\nabla \log p_t(X_t)\right]\mathrm{d}t + g(t)\mathrm{d}\tilde{B}_t, \quad X_T \sim \pi^{\mathrm{base}} , \tag{5}$$

where $(\tilde{B}_t)_{t\geq 0}$ is a reverse-time standard Brownian motion. Interestingly, the stochastic process induced by the denoising SDE has the same marginal distributions $(p_t)_{t\in[0,T]}$ as the stochastic process induced by its deterministic counterpart, called the probability flow ODE (PF-ODE) (Song et al., 2021)

$$\mathrm{d}X_t = \left[ f(t)X_t - \tfrac{g^2(t)}{2}\nabla\log p_t(X_t) \right]\mathrm{d}t, \quad X_T \sim \pi^{\mathrm{base}} . \tag{6}$$

Thus, to obtain samples from $\pi$ at inference, one needs to either solve the SDE (5) or the ODE (6) backward in time (*i.e.*, from $t = T$ to $t = 0$), starting from noise samples drawn from $\pi^{\mathrm{base}}$. Below, we detail the denoising transition kernels and transport maps associated with approximate numerical solvers for, respectively, the SDE (5) and the ODE (6).

**Stochastic transition kernels.** For $0 \leq s < t \leq T$, the conditional distribution of $X_t$ given $X_s = x_s$ is a tractable Gaussian distribution $q_{t|s}(\cdot|x_s)$, called *noising* transition kernel, that writes as

$$q_{t|s}(\cdot \mid x_s) = \mathrm{N}(\alpha_{t|s}\, x_s, \sigma_{t|s}^2\, \mathrm{I}_d) , \tag{7}$$

where $\alpha_{t|s} = S(t)/S(s)$ and $\sigma_{t|s}^2 = S^2(t)[\sigma^2(t) - \sigma^2(s)]$. In contrast, the conditional distribution of $X_s$ given $X_t = x_t$ induced by the denoising SDE (5), denoted $q_{s|t}(\cdot|x_t)$ and called *denoising* transition kernel, does not have a closed-form expression in general and is usually approximated by a Gaussian distribution.

A classical way to approximate $q_{s|t}$ is to use a Gaussian distribution whose mean is given by Tweedie's formula (Robbins, 1992),

$$m_{s|t}(x_t) = \mathbb{E}[X_s \mid X_t = x_t] = \alpha_{t|s}^{-1}\left( x_t + \sigma_{t|s}^2\nabla\log p_t(x_t) \right). \tag{8}$$

A widely used instance of this approach is the Denoising Diffusion Probabilistic Model (DDPM) $\beta$-scheme (Ho et al., 2020), which underlies many large-scale diffusion model implementations and defines

$$q_{s|t}^{\mathrm{DDPM}}(\cdot \mid x_t) = \mathrm{N}(m_{s|t}(x_t), \sigma_{t|s}^2\mathrm{I}_d) . \tag{9}$$

Another possibility is to construct Gaussian kernels by numerically solving the denoising SDE (5), for example with Euler–Maruyama (EM) or Exponential Integration (EI) schemes, the latter often being more accurate than EM over large time intervals. Related computations are given in Appendix B. We call these methods *first-order* because they depend only on the score.

In contrast to first-order methods, *second-order* approximations of the denoising kernel $q_{s|t}(\cdot|x_t)$ also use information from the Hessian $\nabla^2\log p_t(x_t)$, in addition to the score $\nabla\log p_t(x_t)$. A natural construction is to keep the Tweedie mean (8), while replacing the fixed covariance by its second-order counterpart (Grenioux et al., 2024, Appendix A, Lemma 4). This yields the state-dependent covariance

$$\Sigma_{s|t}(x_t) = \mathrm{Cov}[X_s \mid X_t = x_t] = \alpha_{t|s}^{-2}\left( \sigma_{t|s}^2\mathrm{I}_d + \sigma_{t|s}^4\nabla^2\log p_t(x_t) \right) ,$$

and the Gaussian approximation

$$q_{s|t}^{\mathrm{DDPM\text{-}2}}(\cdot \mid x_t) = \mathrm{N}(m_{s|t}(x_t), \Sigma_{s|t}(x_t)) . \tag{10}$$

**Deterministic transition maps.** In the case of the PF-ODE (6), $q_{s|t}$ degenerates to a Dirac mass, *i.e.*, $X_s = \mathrm{T}_{s|t}(x_t)$ where $\mathrm{T}_{s|t} : \mathbb{R}^d \to \mathbb{R}^d$ is the deterministic map that solves the ODE (6) backward in time on $[s, t]$. In practice, $\mathrm{T}_{s|t}$ is intractable too, but may be approximated via first-order integration methods. For instance, using the Euler scheme leads to

$$\mathrm{T}_{s|t}^{\mathrm{EM}}(x_t) = x_t - f(t)(t - s)x_t + \tfrac{g^2(t)}{2}(t - s)\nabla\log p_t(x_t) . \tag{11}$$

Similarly to the stochastic setting, EI versions of such transition maps can be derived to reduce discretization error, see Appendix B for more details.

**Training DMs.** In practice, the score functions $\{\nabla \log p_t\}_{t \in [0,T]}$ and, for second-order methods, the corresponding Hessians $\{\nabla^2 \log p_t\}_{t \in [0,T]}$, are not available in closed form for general target distributions and must therefore be estimated. As a result, data generation relies on approximate dynamics: first, the SDE (5) or ODE (6) is approximated through estimated scores (yielding an *estimation error*); second, these approximate dynamics are numerically solved using the tools described above (yielding a *discretization error*).

Score functions are typically learned from data via score-matching losses (Hyvärinen, 2005; Vincent, 2011; Song et al., 2021; Bortoli et al., 2024). While Hessians can in principle be obtained by differentiating the learned score network, doing so is computationally prohibitive in practice. Early methods therefore relied on state-independent scalar approximations (Ho et al., 2020). More recent works instead learn diagonal or full-matrix approximations (optionally state-dependent) through dedicated objectives built on top of a pre-trained score model (Nichol & Dhariwal, 2021; Bao et al., 2022b;a); see Ou et al. (2025) for an overview.

Another line of research aims at rather approximating the log-densities $\{\log p_t\}_{t \in [0,T]}$ with neural networks, and then taking the derivative with respect to the input to obtain score or Hessian approximations. Various related objectives have been recently designed, either based on maximum likelihood (Gao et al., 2021; Zhang et al., 2023; Zhu et al., 2024; Noble et al., 2025), consistency via Fokker-Planck equation (Shi et al., 2024; Plainer et al., 2025), consistency via Bayes's rule (He et al., 2026) or multi-label classification (Yadin et al., 2024). In practice, the dominant strategy remains the score matching approach, which indirectly approximates the DM log-densities by training a neural network to match their gradient (Song & Kingma, 2021; Salimans & Ho, 2021; Du et al., 2023; Phillips et al., 2024; Thornton et al., 2025) or their time derivative (Guth et al., 2025a; Yu et al., 2025) : for the latter, we will refer to it as "time" score matching.

---

**Diffusion model under consideration**

In all experiments presented below, the noising diffusion process is chosen to be the linear Variance Preserving (VP) diffusion path (Song et al., 2021) with hyperparameters $(\beta_{\min}, \beta_{\max}, T) = (0.1, 20, 1)$, whose exact noising kernel (7) is computed in Lemma 14 (Appendix B.2). When using stochastic denoising kernels, we take by default the DDPM kernel (9) for first-order approaches (additional experiments being reported in Appendix D.4 with the EI scheme), and the DDPM-2 scheme (10) for second-order approaches. When using noising and denoising transport maps, we consider the EI-based ODE integration schemes detailed in Lemmas 16 and 17 (Appendix B.2). Moreover, we set the time discretization $\{t_k\}_{k=0}^K \subset [0,T]$ so as to be constant in log-SNR increments (see Appendix B.4 for more details). We will refer to the induced sequence of densities $\{p_{t_k}\}_{k=0}^{K-1}$ (also denoted $\{p_k\}_{k=0}^{K-1}$), marginally defined by (4), as the "diffusion" path.

---

## 2.2 Standard Monte Carlo & Annealed sampling

This section presents the Monte Carlo tools that are central to all BG methods presented below. We recall that the original purpose of these methods is to generate samples from $\pi$, with only access to its energy function $\mathcal{E}$ up to an additive constant. We begin by reviewing classic techniques, which serve as foundation for the aMC methods introduced afterwards.

**Importance Sampling.** Importance Sampling (IS) is a fundamental Monte Carlo method that approximates expectations taken under $\pi$ using samples drawn from a proposal distribution $\rho$ whose density is tractable. Assuming that $\mathrm{Supp}(\rho) \subset \mathrm{Supp}(\pi)$, any $\pi$-integrable function $\phi$ satisfies

$$\mathbb{E}_\pi[\phi(X)] = \mathbb{E}_\rho\left[w(X)\phi(X)\right], \quad \text{where } w(x) = \pi(x)/\rho(x) \text{ is the importance weight.}$$

In practice, this means that sampling from $\pi$ via IS reduces to (i) sample $N$ particles $\{x^i\}_{i=1}^N$ from $\rho$ and (ii) reweight them using the importance weights $\{w(x^i)\}_{i=1}^N$ [1]. Although IS is simple to implement, its accuracy critically depends on how well $\rho$ matches $\pi$. In particular, the variance of the importance weights can grow rapidly, potentially exponentially with the dimension, when the mismatch is large (Agapiou et al., 2017).

---

[1]When the density $\pi$ is only known up to a normalizing constant, as it is often the case in practice, one turns to the *self-normalized* weights $\bar{w}(x^i) = w(x^i)/\sum_{j=1}^N w(x^j)$, which however leads to a biased estimator.

**Markov Chain Monte Carlo.** Markov Chain Monte Carlo (MCMC) methods are designed to simulate a Markov chain whose stationary distribution is $\pi$, hence generating asymptotically accurate samples.

MCMC methods typically construct their transition mechanism using a proposal distribution $q(y|x)$, which suggests a new state $y$ from the current state $x$. The Metropolis-Hastings (MH) algorithm then corrects this proposal via an acceptance-rejection step to ensure that the chain targets the desired distribution $\pi$. Specifically, given $x$, the proposed $y \sim q(\cdot|x)$ is accepted with probability

$$\alpha(x, y) = \min\left(1, \frac{q(x|y)\pi(y)}{q(y|x)\pi(x)}\right) = \min\left(1, \frac{q(x|y)\exp(-\mathcal{E}(y))}{q(y|x)\exp(-\mathcal{E}(x))}\right) , \tag{12}$$

otherwise the new state is set as $x$. Note that the MH algorithm can be extended to the deterministic case, when $q(\cdot|x) = \delta_{\mathrm{T}(x)}$ for a diffeomorphism $\mathrm{T} : \mathbb{R}^d \to \mathbb{R}^d$ that is required to be involutive, *i.e.*, $\mathrm{T} \circ \mathrm{T} = \mathrm{I}_d$. In this case, the acceptance probability only depends on the previous state $x$ and writes

$$\alpha(x) = \min\left(1, \frac{\mathrm{T}_\#\pi(x)}{\pi(x)}\right) = \min\left(1, \frac{\exp(-\mathcal{E}(\mathrm{T}(x))|\det J_T(x)|}{\exp(-\mathcal{E}(x))}\right) . \tag{13}$$

This deterministic formulation encompasses the popular Hamiltonian Monte Carlo (HMC) algorithm (Neal, 2012). As with IS, the performance of such MH-based samplers hinges on the quality of the proposal. For instance, independent proposals scale poorly with dimension (Grenioux et al., 2023), and multi-modal targets pose additional challenges, as proposals must efficiently explore both within and across the modes. Modern MH variants (Metropolis et al., 1953; Duane et al., 1987), including the Metropolis-Adjusted Langevin Algorithm (MALA) (Roberts & Tweedie, 1996), leverage gradient information to improve local mixing but still struggle with global exploration.

While IS and MCMC are fundamental sampling tools, they often fail in high-dimensional or multi-modal settings. *Annealed* sampling specifically addresses this limitation by breaking the original sampling problem into $K$ sampling problems with gradual complexity, by introducing a sequence of distributions $\{p_k\}_{k=0}^K$ that smoothly bridge between a simple base distribution $p_K = \pi^{\mathrm{base}}$ and the target $p_0 = \pi$. We consider such sequence in the rest of this section. By leveraging correlations across this sequence, it is possible to gradually transform samples from $\pi^{\mathrm{base}}$ into samples from $\pi$ while avoiding the pitfalls of standard MC methods.

**Annealed Importance Sampling.** Annealed Importance Sampling (AIS) (Neal, 2001) extends classic IS by defining a joint target distribution $\pi_{0:K}$ over a sequence of variables $(x_0, \ldots, x_K)$ such that its 0-th marginal is the target distribution $\pi$. Similarly, a joint proposal distribution $\rho_{0:K}$ is built such that its $K$-th marginal is the base distribution $\pi_{\mathrm{prior}}$. Both of these joint distributions are designed recursively as follows

$$\pi_{0:K}(x_{0:K}) = \pi(x_0) \prod_{k=0}^{K-1} q_{k+1|k}(x_{k+1}|x_k), \quad \rho_{0:K}(x_{0:K}) = \pi^{\mathrm{base}}(x_K) \prod_{k=0}^{K-1} q_{k|k+1}(x_k|x_{k+1}) , \tag{14}$$

where $q_{k+1|k}$ and $q_{k|k+1}$ respectively denote *forward* and *backward* Markov transition kernels. In this case, the importance weights are defined by

$$w^{\mathrm{AIS}}(x_{0:K}) = \frac{\pi_{0:K}(x_{0:K})}{\rho_{0:K}(x_{0:K})} \tag{15}$$

Analogously to IS, sampling from $\pi$ reduces to (i) sample $N$ trajectories of particles $\{x_{0:K}^i\}_{i=1}^N$ from $\rho_{0:K}$ and (ii) reweight the particles $\{x_0^i\}_{i=1}^N$ with the importance weights $w^{\mathrm{AIS}}(x_{0:K})$ [2]. However, while easier to achieve than classic IS, the efficiency of AIS also depends on how closely $\rho_{0:K}$ matches $\pi_{0:K}$. In particular, if there exists a sequence of bridging distributions $\{p_k\}_{k=0}^K$ (*i.e.*, such that $p_0 = \pi$ and $p_K = \pi^{\mathrm{base}}$) for which the forward and backward kernels satisfy the Bayes rule defined as

$$p_k(x_k)q_{k+1|k}(x_{k+1}|x_k) = p_{k+1}(x_{k+1})q_{k|k+1}(x_k|x_{k+1}), \quad \forall k \in \{0, \ldots, K-1\} , \tag{16}$$

then it holds exactly that $\pi_{0:K} = \rho_{0:K}$, *i.e.*, the estimator has minimal variance.

In standard AIS (Neal, 2001), the forward and backward kernels are typically chosen to be identical reversible MCMC kernels with respect to a given density path $\{p_k\}_{k=0}^K$ interpolating $\pi$ to $\pi^{\mathrm{base}}$, which simplifies the importance weights given in (15) but violates the Bayes consistency condition (16).

---

[2]In practice, these weights are also self-normalized as in classic IS.

**Sequential Monte Carlo.** Sequential Monte Carlo (SMC) methods (Doucet et al., 2001; Del Moral et al., 2006) address a major limitation of AIS, namely weight degeneracy, where importance weights progressively concentrate on a few particles—an effect that is particularly severe in high-dimensional settings. While SMC relies on the same forward and backward kernels as AIS, it introduces intermediate resampling steps that effectively decompose a single long AIS trajectory from $p_K$ to $p_0$ into two consecutive AIS procedures. Concretely, an initial AIS run propagates particles from $p_K$ to an intermediate distribution $p_k$ for some $k \in \{1, \ldots, K-1\}$; particles are then resampled according to their importance weights to obtain a population representative of $p_k$. A second AIS run, initialized from these resampled particles, subsequently propagates the system from $p_k$ to $p_0$. This mid-trajectory realignment prevents particle collapse, maintains diversity, and significantly reduces weight degeneracy. The construction naturally extends to multiple resampling points by partitioning the path between $\pi^{\text{base}}$ and $\pi$ into shorter AIS segments, which substantially reduces the variance of the AIS estimator without increasing the cost of importance-weight evaluations. In practice, SMC methods are often further augmented with MCMC rejuvenation steps at each stage to better align particles with the intermediate distributions, at the expense of additional computational cost.

**Replica Exchange.** Replica Exchange (RE) (Swendsen & Wang, 1986; Geyer et al., 1991; Hukushima & Nemoto, 1996) is an annealed sampling method that predates AIS and SMC. Unlike these sequential methods, RE correlates the distributions $\{p_k\}_{k=0}^{K}$ in parallel, rather than through a recursion. The goal is to construct a MCMC algorithm targeting the extended distribution $\bar{\pi}_{0:K}(x_{0:K}) = p_0(x_0)p_1(x_1)\ldots p_K(x_K)$. Its transition kernel is composed of two parts: (i) an exploration kernel that independently applies standard MCMC updates to each $p_k$ in parallel, and (ii) a communication kernel that correlates the different marginals. A basic communication move consists of a deterministic "swap" between two consecutive levels $k$ and $k+1$, mapping $(x_0, \ldots, x_k, x_{k+1}, \ldots, x_K)$ to $(x_0, \ldots, x_{k+1}, x_k, \ldots, x_K)$. Since this mapping is involutive, it can be used within the Metropolis–Hastings correction to ensure that the joint distribution $\bar{\pi}_{0:K}$ is stationary. The corresponding acceptance rate obtained from (13) is given by

$$\alpha_k^{\text{RE}}(x_{0:K}) = \min\left(1, \frac{p_{k+1}(x_k)p_k(x_{k+1})}{p_k(x_k)p_{k+1}(x_{k+1})}\right). \tag{17}$$

By applying these MH-calibrated swaps in parallel between even or odd pairs of indices in $\{0, \ldots, K\}$, one defines the even and odd communication kernels, respectively. These are commonly combined using a uniform mixture to build the full communication kernel. However, recent work suggests that deterministically alternating between even and odd kernels is more effective (Okabe et al., 2001; Syed et al., 2022). We adopt this so-called *non-reversible* strategy in the rest of the paper.

**Standard designs of interpolation density paths.** A central component of all aMC methods is the design of the interpolation density path. This path is critical to ensure good performance: in AIS and SMC, it governs the overlap between consecutive distributions, which directly affects the variance of the estimators; in RE, the consecutive overlap controls the probability of accepting swap moves between adjacent levels. When only the unnormalized density of $\pi$ is available, a common choice is the geometric interpolation path (Neal, 2001; Gelman & Meng, 1998), defined for all $x \in \mathbb{R}^d$ by

$$p_k(x) \propto \pi(x)^{\beta_k}\pi^{\text{base}}(x)^{1-\beta_k}, \tag{18}$$

where the annealing schedule $\{\beta_k\}_{k=0}^{K}$ is decreasing, and satisfies $(\beta_0, \beta_K) = (1, 0)$. We will refer to the collection of unnormalized densities obtained via (18) as the "tempering" path. The major benefit of these paths is their computational efficiency, as they allow for simple evaluations of the scores $\{\nabla \log p_k\}_{k=0}^{K}$, which are frequently required in MCMC transition kernels, via a linear combination of $\nabla \log \pi$ and $\nabla \log \pi^{\text{base}}$.

However, these paths are usually pathological for multi-modal targets, as they suffer from mass teleportation (also referred to as mode switching), which reflects sudden shifts in probability mass between modes along the interpolation path (Woodard et al., 2009; Máté & Fleuret, 2023). In practice, such sudden shifts undermine the assumed proximity between bridging densities, leading to instability in aMC. Mitigating this issue usually requires either carefully tuning the annealing schedule $\{\beta_k\}_{k=0}^{K}$ for each target or using a large number of intermediate levels $K$, which can incur significant computational cost.

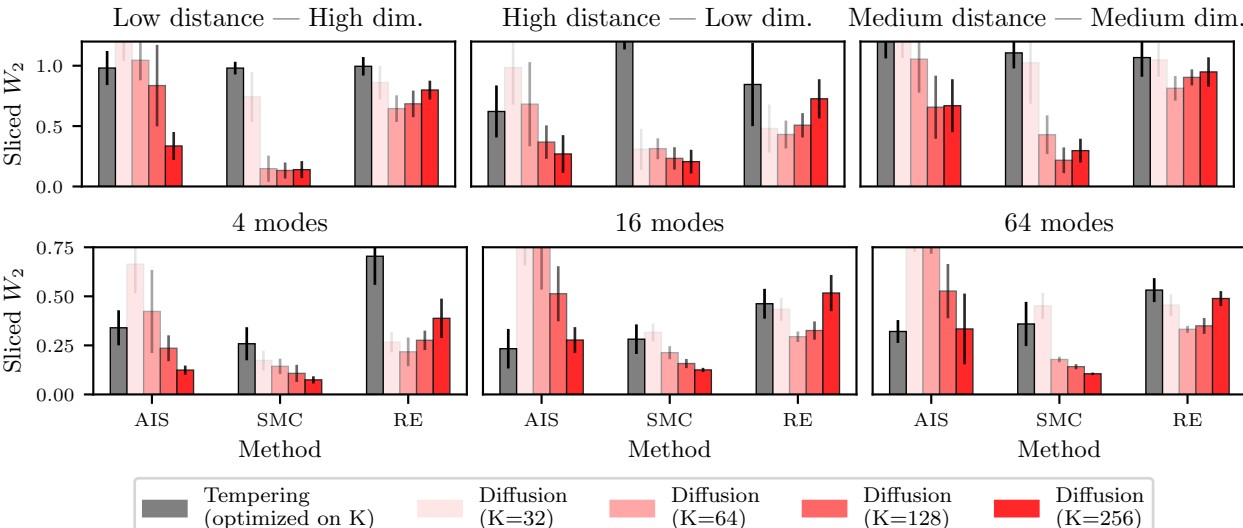

Figure 1: **Sampling results for classic annealed samplers with diffusion (red) and tempering (grey) paths**, when targeting *TwoModes* (**Top**) and *ManyModes* (**Bottom**) in idealized setting (**A**). For tempering paths, we display the best-performing result among all values of $K$. For diffusion paths, we display the results for $K \in \{32, 64, 128, 256\}$ : the darker the bar, the higher $K$. In particular, these configurations do not share the same computational budget. Each result is averaged over 8 runs with 8,192 samples per run.

The question of how to optimize the annealing schedule has been studied by Syed et al. (2021; 2022; 2025), who introduce the *global barrier* $\Lambda$ quantifying the intrinsic difficulty of sampling along a density path, and propose to spread this difficulty by approximating the inverse $\Lambda$ function. For a fixed number of levels $K$, this yields a schedule with constant barrier increments, reducing the variance of the log normalizing constant estimate in AIS/SMC and improving the number of round-trip in RE. In practice, they rely on a progressive sampling phase to estimate the barrier $\Lambda$ and deduce the corresponding schedule, which following their terminology we call the $\Lambda$-optimal schedule. We use it systematically in all tempering-path experiments, giving tempering-based methods their most favorable setting.

## 3 Diffusion-based aMC as a Boltzmann Generator backbone : benefits and pitfalls

Diffusion models are a natural fit for aMC schemes, as they inherently define a sequence of intermediate densities that can be leveraged in sampling algorithms such as AIS, SMC, or RE. In Section 3.1, we show that even a naive integration, simply using the sequence of DM densities as a direct replacement for the classic tempering sequence, can already deliver strong performance, thanks to the favorable properties of the Gaussian convolution paths induced by DMs. In Section 3.2, we review related methodologies, that additionally propose to "enhance" standard aMC tools using DM stochastic transition kernels. However, we demonstrate in Section 3.3 that those designs are fundamentally limited in challenging multi-modal scenarios.

We emphasize that, although the presented methods involve different hyperparameters, we focus our numerical evaluation solely on the effect of the number of annealing levels (defined as $K \in \{32, 64, 128, 256\}$), common to all methods, because it directly controls the overlap between consecutive distributions along the annealing path, a factor highlighted as crucial to the performance of aMC.

### 3.1 Of the interest of diffusion-based density paths

As noted by Máté & Fleuret (2023), diffusion paths are typically well conditioned and avoid common pitfalls of tempering paths, such as abrupt mode switching. In particular, they preserve the relative mass of the modes throughout the annealing process, leading to more stable sampling dynamics. This explains why diffusion paths consistently outperform tempering paths in aMC, as shown in idealized setting (**A**) by Figure 1.

Across all standard aMC methods considered in Section 2.2 and across all targets, the perfectly learned diffusion path outperforms the optimally tuned tempering path (18) *for a large range of annealing levels $K$*.[3] For AIS and SMC, performance generally improves with increasing $K$, with best results attained at the largest value tested ($K = 256$). For RE-based samplers, the dependence on $K$ is less monotonic: while larger $K$ improves local overlap and facilitates swaps, it can also hinder long-range communication between levels, leading to degraded performance beyond a certain point. These empirical conclusions are further supported by the log-normalization estimates reported in Appendix D.3, which show an even clearer and larger gap between tempering and diffusion-based aMC samplers, with the same dependence on $K$. Overall, our experiments highlight the strength of diffusion over tempering paths, motivating their use when a learned DM is available.

### 3.2 Review of existing diffusion-based aMC-BGs

Interestingly, DMs provide more than a sequence of intermediate densities: they also grant access to noising and denoising stochastic transition kernels (see, e.g., (7) and (9), (10)), which can be strategically exploited to improve both efficiency and robustness. In this section, we review existing extensions of aMC that leverage this additional structure. These approaches assume access to a DM defined on a discrete time grid $\{t_k\}_{k=0}^K \subset [0, T]$, enabling the additional evaluation of the associated noising kernels $\{q_{k+1|k}\}_{k=0}^{K-1}$ and denoising kernels $\{q_{k|k+1}\}_{k=0}^{K-1}$.

**Diffusion-based AIS.** DMs have been successfully integrated into AIS frameworks in recent work (Zhang et al., 2024; 2025). The core idea consists in using the exact noising transition kernels (7) as forward kernels, and *first-order* denoising transition kernels[4], similar to (9), as backward kernels, to respectively define the extended target and proposal distributions, see (14). By doing so, only the score functions are needed, not the log-densities. A key advantage of this approach is that, when the backward kernels match the exact denoising kernels, the forward and backward transitions satisfy the optimal Bayes condition (16), which ensures that the importance weights exhibit minimal variance.

**Diffusion-based SMC.** The exact same use of DM transition kernels has recently been extended to the SMC setting through the *Particle Denoising Diffusion Sampler* (PDDS) (Phillips et al., 2024), with the EI kernel considered in their numerical experiments. In contrast to AIS, however, the SMC formulation additionally requires the intermediate log-densities, up to normalizing constants, in order to perform resampling.

**Diffusion-based RE.** In the spirit of PDDS, Zhang et al. (2026) lately explored the use of DM transition kernels within the RE framework to propose the *Diffusion-based Accelerated Parallel Tempering* (Diff-APT) sampler. In Diff-APT, the traditional RE swaps between adjacent levels are combined with stochastic refinements inherited from those kernels. Given current states $x_k$ and $x_{k+1}$ at levels $k$ and $k+1$, Diff-APT first samples proposal states $y_{k+1} \sim q_{k+1|k}(\cdot|x_k)$ and $y_k \sim q_{k|k+1}(\cdot|x_{k+1})$, where $q_{k+1|k}$ and $q_{k|k+1}$ respectively denote the exact noising (forward) kernel, see (7), and a *first-order* denoising (backward) kernel, taken as the EM scheme in their experiment, between times $t_k$ and $t_{k+1}$. By exploiting the underlying correlation between noise levels, each chain is moved closer to its corresponding target distribution, respectively $p_{k+1}$ and $p_k$. Then, this stochastic-based swap is calibrated using the MH correction, resulting in the following acceptance probability

$$\alpha_k^{\mathrm{RE}}(x_{0:K}, y_{0:K}) = \min\left(1, \frac{p_k(y_k)p_{k+1}(y_{k+1})q_{k+1|k}(x_{k+1}|y_k)q_{k|k+1}(x_k|y_{k+1})}{p_k(x_k)p_{k+1}(x_{k+1})q_{k+1|k}(y_{k+1}|x_k)q_{k|k+1}(y_k|x_{k+1})}\right), \tag{19}$$

defined for any $(x_{0:K}, y_{0,K}) \in \mathbb{R}^{(K+1)d} \times \mathbb{R}^{(K+1)d}$. Compared to the standard RE acceptance ratio (17), this novel expression features four additional terms, which correspond to symmetric evaluations of forward and backward kernels. As in AIS and SMC, if the forward and backward kernels satisfy the Bayes condition (16) the proposed swap is systematically accepted, *i.e.*, the acceptance probability (19) always equals one.

---

[3]The only exception is AIS on the *ManyModes* target with 16 and 64 modes, where diffusion achieves performance comparable to tempering only for large values of $K$.

[4]Although Zhang et al. (2025) propose to adjust the covariance of the denoising kernels via additional learning, we still consider this approach as 'first-order' as it does not rely on the Hessian functions $\{\nabla^2 \log p_k\}_{k=0}^K$.

### 3.3 First-order approaches fail to bring informative transition information between annealing levels

Although theoretically well motivated, the existing DM-based aMC-BGs reviewed in Section 3.2 do not yield noticeable improvements over the standard baseline studied in Section 3.1, in idealized setting **(A)** where both log-densities and score functions are assumed to be perfectly known.

In Figure 3, we report sampling errors in the perfect-learning regime across all *TwoModes* and *ManyModes* targets. We compare the classical aMC baseline (red bars, the same as in Figure 1) with the aforementioned methods combined with the DDPM scheme (9) (blue bars). We find that first-order AIS and SMC methods systematically fail to improve over their respective baseline, while first-order RE yields only marginal gains in most cases; however, its overall performance remains substantially worse than that of AIS and SMC. One might ask whether DDPM is the right choice for first-order transition kernels. In Appendix D.4, we show that alternative SDE-based denoising kernels from prior work actually degrade performance, suggesting the issue lies within the choice of *first-order* backward kernels in aMC schemes rather than with DDPM specifically.

To validate this claim, we also consider *second-order* denoising kernels based on the DDPM-2 scheme (10) (green bars), assuming access to the Hessian functions.[5] These kernels consistently yield substantial gains over both the baseline and their first-order counterparts, highlighting the value of higher-order information for guiding transitions along the diffusion density path. Notably, all three second-order aMC samplers reach comparably strong performance and are far less sensitive to $K$: AIS plateaus at $K \geq 128$, SMC at $K \geq 64$, and RE is essentially flat across all $K$. For SMC and RE, we also tested multi-step transition kernels in place of the default single-step kernels, following the RE methodology of Zhang et al. (2026); results are reported in Appendix D.4. Under a fixed computational budget, multi-step kernels actually degrade sampling performance within both first- and second-order variants, while leaving unchanged the overall superiority of second-order methods. In practice, using second-order kernels nonetheless requires additional covariance estimation (Ou et al., 2025), which is beyond the scope of most DM training methods, where only approximations of log-densities and/or scores are available.

## 4 Exploiting deterministic transitions of DMs in aMC methods : a new hope ?

In this section, we propose investigating the design of a deterministic diffusion-based aMC-BG. We first describe its general principle in Section 4.1 and detail in Section 4.2 how to instantiate it concretely for DMs. We demonstrate that, in idealized setting **(A)**, our method outperforms previous first-order approaches, while being on par with the second-order stochastic ones. Similarly to Section 3.2, we assume that we have access to scores and log-densities from a DM associated to a certain time discretization $\{t_k\}_{k=0}^K \subset [0, T]$.

### 4.1 General methodology

**From stochastic to deterministic DM dynamics.** To further exploit the potential of aMC sampling methods, we propose to use *deterministic* kernels, by replacing stochastic transition kernels with their deterministic counterparts, which approximate the PF-ODE (6) rather than the denoising SDE (5). Below, we explain how the aMC framework presented in Section 2.2 naturally extends to this setting.

**Annealed samplers with deterministic transitions.** In this paragraph, we consider $K$ pairs of candidate transport maps, divided between *forward* maps $\{T_{k+1|k}\}_{k=0}^{K-1}$ and *backward* maps $\{T_{k|k+1}\}_{k=0}^{K-1}$. Moreover, we assume that **(a)** these maps are $C^1$-diffeomorphisms, and **(b)** verify the *per-level mutual invertibility* property, defined for any $k \in \{0, \dots, K-1\}$ by

$$T_{k+1|k} \circ T_{k|k+1} = T_{k|k+1} \circ T_{k+1|k} = \text{Id} . \tag{20}$$

To exploit the use of these transport maps into aMC samplers, we simply propose to set the forward Markov kernels $\{q_{k+1|k}\}_{k=0}^{K-1}$ and backward Markov kernels $\{q_{k|k+1}\}_{k=0}^{K-1}$ (used as transition kernels between adjacent levels in aMC methods) as Dirac masses defined for any $k \in \{0, \dots, K-1\}$ by $q_{k+1|k} = \delta_{T_{k+1|k}}$ and $q_{k|k+1} = \delta_{T_{k|k+1}}$ respectively.

---

[5]In our experiments, we only use the diagonal of the exact Hessians, which provides a good compromise between accuracy and computational efficiency in high dimension.

*Adaptation to AIS/SMC instance.* Under this setting, the AIS framework boils down to standard IS targeting $\pi$ with the push-forward of $\pi^{\text{base}}$ through all backward maps as proposal. Using the change-of-variables formula, the AIS weight (15) admits the following deterministic version, solely depending on the state $x_0$ :

$$w^{\text{AIS}}(x_0) = \frac{\pi(x_0)}{(\text{T}_{0:K})_{\#}\pi^{\text{base}}(x_0)}, \quad \text{with } \text{T}_{0:K} = \text{T}_{0|1} \circ \text{T}_{1|2} \ldots \circ \text{T}_{K-1|K}. \tag{21}$$

Using the chain rule, the determinant of the Jacobian of the full map $\text{T}_{0:K}$ appearing in $(\text{T}_{0:K})_{\#}\pi^{\text{base}}$ (see (1)) can be written as a product of the determinants of Jacobian of the individual maps $\text{T}_{k|k+1}$ for $k \in \{0, K-1\}$.

*Adaptation to RE instance.* By substituting Markov kernels with Dirac masses, the resulting swap in RE sampling procedure defines a deterministic map on the full extended space

$$\bar{\text{T}}_k(x_{0:K}) = (x_0, \ldots, \text{T}_{k|k+1}(x_{k+1}), \text{T}_{k+1|k}(x_k), \ldots, x_K) ,$$

which is guaranteed to be involutive due to assumption **(b)**. In particular, this property ensures that $\bar{\text{T}}_k$ can effectively be integrated within the Metropolis–Hastings algorithm with deterministic proposal, see (13). Using the identity $(\bar{\text{T}}_k)_{\#}\bar{\pi}(x_{0:K}) = p_0(x_0) \ldots (\text{T}_{k|k+1})_{\#}p_{k+1}(x_k)(\text{T}_{k+1|k})_{\#}p_k(x_{k+1}) \ldots p_K(x_K)$, we obtain the following acceptance probability:

$$\alpha_k^{\text{RE}}(x_{0:K}) = \min\left(1, \frac{(\text{T}_{k|k+1})_{\#}p_{k+1}(x_k)(\text{T}_{k+1|k})_{\#}p_k(x_{k+1})}{p_k(x_k)p_{k+1}(x_{k+1})}\right) . \tag{22}$$

This swapping mechanism is illustrated in Figure 2. Note that setting both $\text{T}_{k|k+1}$ and $\text{T}_{k+1|k}$ as the identity map recovers the standard RE algorithm as a special case.

**Effective application to diffusion models.** For all aMC methods, the choice of the forward maps $\{\text{T}_{k+1|k}\}_{k=0}^{K-1}$ and backward maps $\{\text{T}_{k|k+1}\}_{k=0}^{K-1}$ is optimal if those verify the deterministic version of the Bayes rule (16) given by

$$(\text{T}_{k+1|k})_{\#}p_k = p_{k+1}, \quad (\text{T}_{k|k+1})_{\#}p_{k+1} = p_k, \quad \forall k \in \{0, \ldots, K-1\} .$$

Indeed, in the case of AIS/SMC samplers, satisfying this identity would enable to get zero-variance in the estimator, while this would ensure to maximize the acceptance rate in the RE sampler. Intuitively, this rule reflects the fact that the maps should be chosen so as to perfectly transport particles between adjacent levels to match their target distribution.

The next section discusses two key challenges that arise when implementing these methods in practice using DM's ingredients :

1. *How to design transition maps that verify the invertibility condition* (20) *?*

2. *Given those maps, how to compute efficiently the push-forward densities appearing in* (21) *and* (22) *?*

### 4.2 The key components needed for efficient implementation

In this section, we first describe how to construct invertible transport maps that approximately solve the probability flow ODE (6). We then present a practical methodology, inspired by residual NFs, for obtaining estimates of the push-forward density terms appearing in (21) and (22). As made explicit by the change-of-variables formula (1), this approach requires (i) the ability to evaluate the transport maps and (ii) the computation (or unbiased estimation) of their Jacobian determinants. In the latter case, the resulting stochasticity is handled via the penalty correction of Ceperley & Dewing (1999), which preserves the consistency of the IS-based weights (in AIS/SMC) and the invariant measure of the MH-based swap (in RE). The construction directly extends to the case where the score in the PF-ODE is replaced by an estimate, by simply substituting the estimated score throughout. Our main contributions are summarized in Table 1. In what follows, we focus on two adjacent noise levels $k$ and $k+1$.

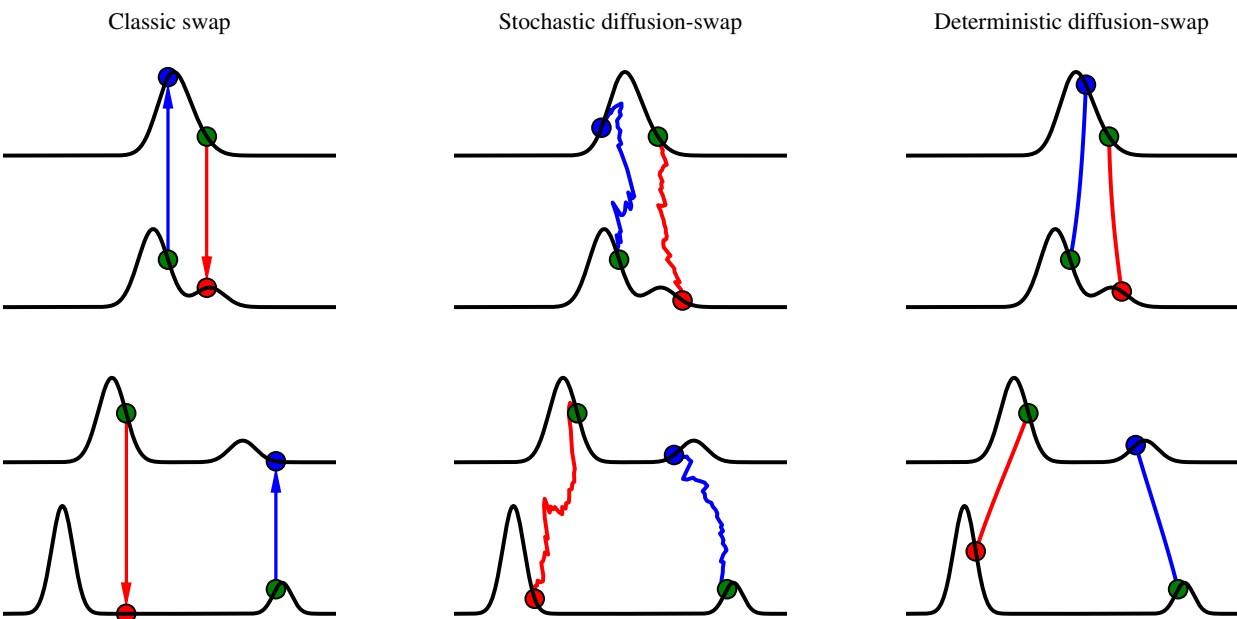

Figure 2: **Different diffusion-based swapping mechanisms for Replica Exchange. (Left)** standard swap scheme, see Section 2.2, where samples are exchanged directly across noise levels without guidance, potentially moving into low-probability regions. **(Middle)** DM-based swaps using forward and backward Markov kernels, as proposed by Zhang et al. (2026) (coined Diff-APT), see Section 3.2. **(Right)** DM-based swaps using forward and backward transport maps under the deterministic framework introduced in Section 4.1. In each panel, black lines denote noise levels; green dots mark the original samples; blue and red paths indicate forward (low to high noise) and backward (high to low noise) trajectories, respectively; and the swapped samples are shown as the resulting blue and green dots. The DM-based swaps better preserve high-probability regions during exchange, enabling theoretically more effective sampling.

**Building invertible transport maps.** To guarantee that the forward map $T_{k+1|k}$ and the backward map $T_{k|k+1}$ are mutually inverse, see (20), we cannot simply rely on *explicit* ODE integrators of the form (11). Indeed, forward and backward maps inherited from such first-order approximations do not, in general, compose to the identity[6]. This motivates us to move towards the class of *implicit* integrators, which are extensively used to simulate Hamiltonian dynamics where trajectory invertibility is often a desirable feature. In particular, we propose to design our transition maps via the *Implicit Midpoint* (IM) integrator, as presented in Proposition 1 for the Euler scheme.

**Proposition 1** (IM integrator with Euler scheme). *Let* $T_{k+1|k} : \mathbb{R}^d \to \mathbb{R}^d$ *and* $T_{k|k+1} : \mathbb{R}^d \to \mathbb{R}^d$ *be implicitly defined as*

$$T_{k+1|k} : x_k \mapsto x_k + \delta_k v\left(t_{k+1/2}, \frac{x_k + T_{k+1|k}(x_k)}{2}\right) , \quad T_{k|k+1} : x_{k+1} \mapsto x_{k+1} - \delta_k v\left(t_{k+1/2}, \frac{T_{k|k+1}(x_{k+1}) + x_{k+1}}{2}\right) ,$$

*where* $t_{k+1/2} = (t_k + t_{k+1})/2$, $\delta_k = t_{k+1} - t_k$ *and* $v(t,x) = f(t)x - (g(t)^2/2)\nabla \log p_t(x)$ *is the velocity field of PF-ODE* (6). *Then, these maps are valid forward and backward integrators of PF-ODE* (6) *on time interval* $[t_k, t_{k+1}]$ *and satisfy the mutual invertibility condition* (20).

In Appendix C, we provide the proof of the above proposition along with its generalization using the Exponential Integration scheme in Appendix B, which offers improved accuracy compared to the Euler scheme when $\delta_k$ is relatively large. Although the maps defined in Proposition 1 cannot be evaluated in closed form as they are by nature *implicit*, they can still be approximated in practice using fixed-point iterations as described in Proposition 2, which guarantees convergence of this scheme under certain assumptions detailed below. We refer to Appendix C for the proof of this result as well as Appendix B for its EI generalization.

---

[6]Note that this reasoning also applies in the case of EI-based first-order integrators.

**Assumption 1** (Score smoothness & discretization error). *(a) There exists $L_k > 0$ such that $\nabla \log p_{t_{k+1/2}}$ is $L_k$-Lipschitz and (b) the step-size $\delta_k$ is sufficiently small [7], that is $\delta_k = O(1/L_k)$.*

**Proposition 2** (Fixed-point approximation of the IM integrator). *Following the same notation as in Proposition 1, under Assumption 1, for any inputs $x_k$ and $x_{k+1}$, the sequences $\{\mathrm{T}_{k+1|k}^{(n)}(x_k)\}_{n \in \mathbb{N}}$ and $\{\mathrm{T}_{k|k+1}^{(n)}(x_{k+1})\}_{n \in \mathbb{N}}$ that are recursively defined as*

$$\mathrm{T}_{k+1|k}^{(0)}(x_k) = x_k, \qquad \mathrm{T}_{k+1|k}^{(n+1)}(x_k) = x_k + \delta_k v\left(t_{k+1/2}, \frac{x_k + \mathrm{T}_{k+1|k}^{(n)}(x_k)}{2}\right), \tag{23}$$

$$\mathrm{T}_{k|k+1}^{(0)}(x_{k+1}) = x_{k+1}, \qquad \mathrm{T}_{k|k+1}^{(n+1)}(x_{k+1}) = x_{k+1} - \delta_k v\left(t_{k+1/2}, \frac{\mathrm{T}_{k|k+1}^{(n)}(x_{k+1}) + x_{k+1}}{2}\right), \tag{24}$$

*converge linearly to $\mathrm{T}_{k+1|k}(x_k)$ and $\mathrm{T}_{k|k+1}(x_{k+1})$, respectively.*

In practice, we only compute the sequences from Proposition 2 up to a range $M \geq 1$ that ensures a prescribed fixed-point convergence tolerance $\varepsilon > 0$, that is, $M$ is of the first order such that

$$\left\|\mathrm{T}_{k+1|k}^{(M+1)}(x_k) - \mathrm{T}_{k+1|k}^{(M)}(x_k)\right\|_2 \leq \varepsilon \ \text{ and } \ \left\|\mathrm{T}_{k|k+1}^{(M+1)}(x_{k+1}) - \mathrm{T}_{k|k+1}^{(M)}(x_{k+1})\right\|_2 \leq \varepsilon, \tag{25}$$

and we approximate $\mathrm{T}_{k+1|k}(x_k)$, resp. $\mathrm{T}_{k|k+1}(x_{k+1})$, by the $(M+1)$-th term $\mathrm{T}_{k+1|k}^{(M)}(x_k)$, resp. $\mathrm{T}_{k|k+1}^{(M)}(x_{k+1})$. While this iterative scheme may introduce numerical errors, we note that potential violations of the invertibility property (20) could be mitigated through an additional optional rejection step as proposed by Noble et al. (2023). We leave the implementation of such a safeguard to future work. In Appendix D.4, we ablate the choice of $M$ across all aMC variants and find that sampling performance is largely insensitive to it; the default value used in our experiments is reported in Appendix D.2.

**Estimating the Jacobian determinants.** We now turn to the second component of (1): computing the Jacobian determinants of the transition maps. Since these quantities are generally intractable, we propose a numerical approximation tailored to the recursive structure of the IM integrators introduced in Proposition 1. Specifically, we first express their *log-determinant* as a power series, following techniques previously used for contractive residual normalizing flows (Behrmann et al., 2019; Chen et al., 2019). This yields the following proposition, the proof of which is given in Appendix C.

**Proposition 3** (Approximation of the Jacobian log-determinants via power series). *Following the same notation as in Propositions 1 and 2, under Assumption 1, for any inputs $x_k$ and $x_{k+1}$, and any prescribed fixed-point range $M \geq 1$ satisfying (25), the following approximation holds*

$$\log\left|\det J_{\mathrm{T}_{k+1|k}}(x_k)\right| \approx \sum_{i=0}^{I} a_{k,i} \mathrm{Tr}([A^{(M)}(x_k)]^i), \quad \log\left|\det J_{\mathrm{T}_{k|k+1}}(x_{k+1})\right| \approx \sum_{i=0}^{I} b_{k,i} \mathrm{Tr}([B^{(M)}(x_{k+1})]^i),$$

*where $I \geq 1$ is a prescribed truncation order,*

$$A^{(M)}(x_k) = \mathbf{H}_{t_{k+1/2}}\left(\frac{x_k + \mathrm{T}_{k+1|k}^{(M)}(x_k)}{2}\right), \quad B^{(M)}(x_{k+1}) = \mathbf{H}_{t_{k+1/2}}\left(\frac{x_{k+1} + \mathrm{T}_{k|k+1}^{(M)}(x_{k+1})}{2}\right),$$

*$\mathbf{H}_{t_{k+1/2}}$ is the Hessian of $\log p_{t_{k+1/2}}$ and $\{a_{k,i}, b_{k,i}\}_{i=0}^{I}$ are given in Proposition 13 (see Appendix B.1).*

Implementing Proposition 3 requires estimating traces of the powered midpoint Hessians. When these Hessians are available, we approximate the Jacobian determinants by simply exponentiating the log-determinant expansion, yielding second-order deterministic aMC methods. Otherwise, we estimate the traces using the Hutchinson identity $\mathrm{Tr}(\mathrm{M}) = \mathbb{E}_v[v^\top \mathrm{M} v]$, with $v \sim \mathrm{N}(0, \mathrm{I}_d)$ (Hutchinson, 1989; Avron & Toledo, 2011); in practice, we rather use the lower-variance Hutch++ estimator (Meyer et al., 2021). This only requires Jacobian–vector products, which can be computed efficiently by reverse-mode automatic differentiation. Since the resulting log-determinant estimates are stochastic, it is not immediately clear that IS and MH algorithms remain consistent when importance weights or acceptance probabilities are themselves random quantities.

---

[7]We provide the exact numerical constants related to this informal assumption in Appendix C.

| Transition method | Forward design | Backward design | Needs $\nabla^2 \log p_k$ |
|---|---|---|---|
| 1st order kernel | Exact noising kernel (7) | DDPM approx. (9) | ✗ |
| 2nd order kernel | Exact noising kernel (7) | DDPM-2 approx. (10) | ✓ |
| IM map via Hutchinson | Fixed-point approx. (23) | Fixed-point approx. (24) | ✗ |
| IM map via Hessian | Fixed-point approx. (23) | Fixed-point approx. (24) | ✓ |

Table 1: **Summary of DM-based transitions used in annealed sampling methods.** The top two rows correspond to stochastic transitions: the "1st order" row recovers prior work. The bottom two rows correspond to the deterministic transitions developed in Section 4. The last column specifies whether access to the Hessians of the log-densities is required. We recall that the acronym IM stands for *Implicit Midpoint*.

To address this, we follow the penalty method of Ceperley & Dewing (1999), a principled framework that ensures AIS/SMC importance weights remain unbiased and consistent, and that RE MH acceptance probabilities preserve the correct invariant distribution, despite this stochasticity. The induced aMC methods are thus of first-order. Details are deferred to Appendix C.1.

Our deterministic formulation introduces two further hyperparameters: the truncation order $I$ of the log-determinant power series (Proposition 3) and the number of Hutchinson random variables (for the first-order variant only). We ablate both in Appendix D.4: (i) sampling performance is largely insensitive to $I$, and (ii) after the penalty correction, the residual stochasticity of the Hutchinson estimator introduces no noticeable bias relative to the deterministic Hessian-based variant. Default values are reported in Appendix D.2, under which the deterministic variants of the aMC samplers incur only a limited computational overhead compared to their stochastic counterparts.

### 4.3 Empirical comparison between DM-based stochastic and deterministic transitions in aMC samplers

In Figure 3, we evaluate the deterministic methodology within AIS, SMC, and RE in idealized setting **(A)**, across all *TwoModes* and *ManyModes* targets. We compare it against approaches based on stochastic kernels, both first-order and second-order. Based on the results, we make the following observations:

(i) *When the Hessian is available*, using deterministic transitions (pink bars) performs on par with the second-order stochastic approach (green bars) across all aMC variants, for each value of $K$. Interestingly, the deterministic method provides even better results for low values of $K$ with AIS/SMC samplers.

(ii) *When the Hessian is not available*, the first-order deterministic variant relying solely on the score functions via the Hutchinson estimator (yellow bars) consistently improves over the standard baseline (red bars) and the use of first-order stochastic kernels (blue bars) presented in prior work, for each value of $K$. This highlights the promise of deterministic mappings in aMC samplers. Remarkably, across all multi-modal scenarios, *the performance gap with the second-order deterministic scheme is barely noticeable* for AIS at large $K$, and is even negligible for SMC, when $K \geq 64$, and for RE, across all values of $K$, proving the effectiveness of our proposed Hutchinson-based statistical estimation to fully exploit first-order information.

*Remark on second-order stochastic kernels.* For the Gaussian denoising kernels given by (10), the Hessian appears in the covariance term. As a result, sampling only involves Jacobian–vector products, which can be handled with standard automatic differentiation tools. In contrast, likelihood evaluation additionally requires inverse–Jacobian–vector products through the term $[\Sigma_{s|t}(x_t)]^{-1}(x_s - m_{s|t}(x_t))$, which is substantially more challenging to implement efficiently. While recent work has begun to address this computational bottleneck (Siskind, 2019), developing practical implementations is an open and promising direction for future research.

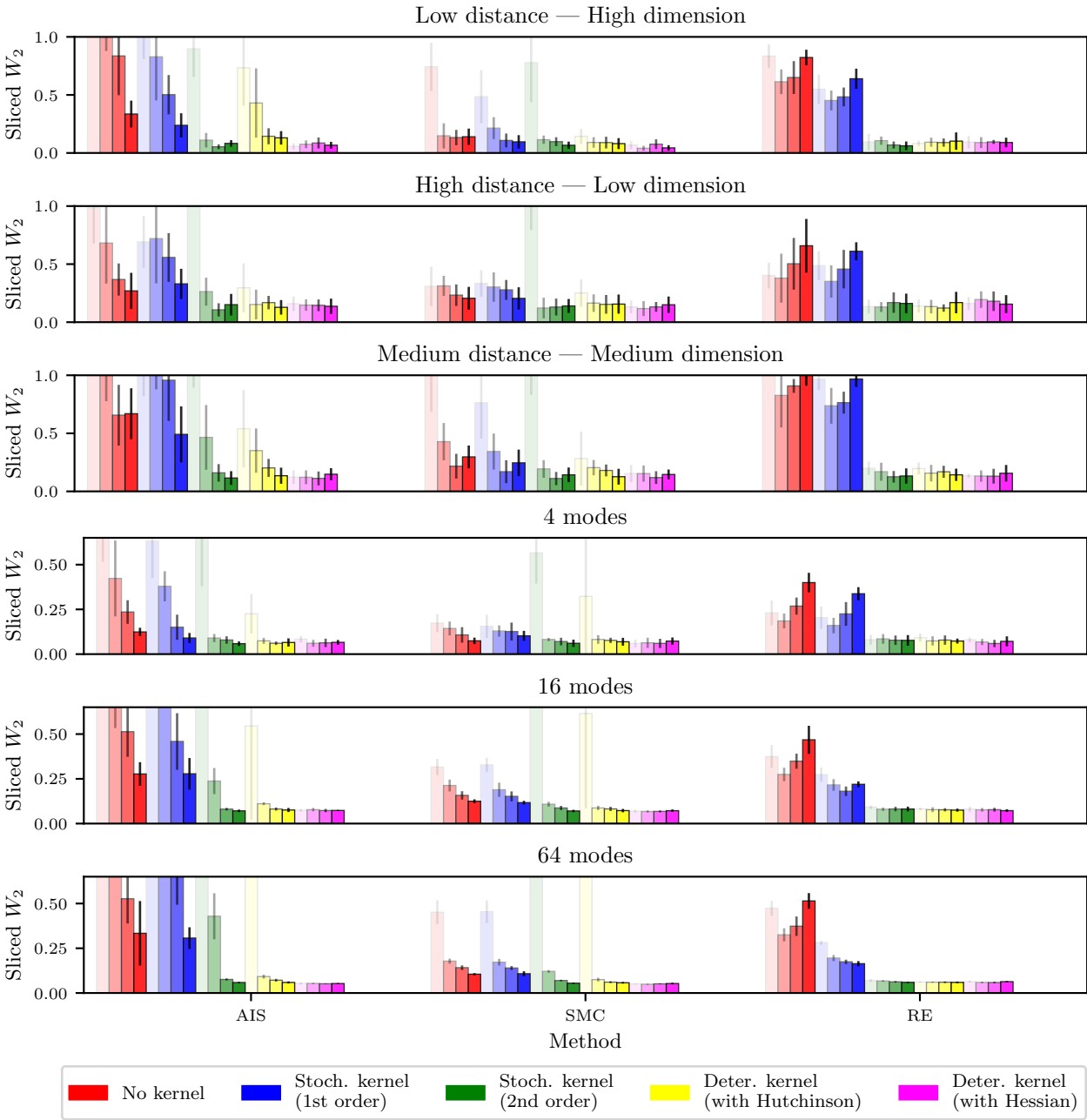

Figure 3: **DM-based aMC-BG results with annealed samplers using different mechanisms**, when targeting *TwoModes* (**Top three rows**) and *ManyModes* (**Bottom three rows**) distributions in idealized setting **(A)**. Each group of bars with the same color corresponds to a specific aMC method: red bars refer to methods that do not use DM-based transition kernels (standard baseline); blue and green refer to methods that exploit 1st-order (prior work) and 2nd-order stochastic kernels; pink and yellow correspond to variants with deterministic maps, where the log-determinant term is respectively computed from the ground truth diagonal Hessian or via the Hutchinson trick (see Section 4). For all settings, we display the results for $K \in \{32, 64, 128, 256\}$ : the darker the bar, the higher $K$ (same range as in Figure 1). In particular, configurations within each bar group do not share the same computational budget. Each result is averaged over 8 runs with 8,192 samples per run. We observe that using 1st-order stochastic kernels does not always lead to better performance than the baseline, while 2nd-order stochastic or deterministic kernels provide consistent improvements. Remarkably, using deterministic transitions combined with the Hutchinson trick, which only requires score evaluations, achieve performance comparable to second-order aMC variants.

## 5 Related Works

**Normalizing flows into annealed sampling.** Normalizing flows have been previously integrated into aMC frameworks. For instance, Arbel et al. (2021) and Matthews et al. (2022) incorporate flows as forward and backward kernels within AIS and SMC algorithms combined with tempering density paths. Other works such as Midgley et al. (2021; 2023b;a) consider AIS schemes where the sequence of densities use $\pi^{\text{base}}$ as a NF, allowing for better conditioned path. On the other hand, Invernizzi et al. (2022) propose an extension of RE of the form of (22), with the key difference being that their deterministic transformations are parameterized by NFs rather than derived from DM-based dynamics.

**Using DMs in aMC for sampling.** This idea has recently seen a growing interest in the generative modeling community. Some works have built upon the AIS backbone with specific choices of transition operators. For instance, Zhang et al. (2024) propose to design both forward and backward stochastic kernels as a mix of exact noising kernels and first-order *explicit* integrators of the PF-ODE, in order to take advantage of the efficiency of deterministic mappings. On the other hand, Zhang et al. (2025) design the backward transition kernels as Gaussian denoising kernels with a flexible scalar variance that is learned, in the same spirit as second-order kernels. Taking SMC as a sampling backbone, Phillips et al. (2024) present a end-to-end algorithm that aims to sample from a target distribution by learning the corresponding DM. This procedure alternates between (i) building a BG toward the target via DM-based SMC (here, the backward transitions are defined as first-order EI kernels) and (ii) updating this DM by minimizing a score matching objective with the samples from stage (i). To be able to evaluate the intermediate log-densities, the DM is parameterized as a multi-level energy-based model. More recently, Zhang et al. (2026) explore the use of DM-based kernels as forward and backward stochastic transitions within a RE framework. Similarly to Phillips et al. (2024), they propose an iterative sampling approach , that involves RE combined with first-order stochastic kernels.

**Combination of annealed sampling and DMs beyond BGs.** Diffusion models have also been combined with aMC methods, though not primarily for building BGs. Instead, these approaches leverage DMs for various downstream tasks. For instance, SMC-based approaches have been proposed for conditional generation (Wu et al., 2023), posterior sampling in Bayesian inverse problems (Cardoso et al., 2024; Dou & Song, 2024; Janati et al., 2024; 2025), reward-guided generation and fine-tuning (Uehara et al., 2024; Kim et al., 2025; Singhal et al., 2025), as well as compositional and controlled generation tasks (Thornton et al., 2025; Skreta et al., 2025). While these methods use advanced sampling, their primary focus lies in enhancing/extending generation capabilities rather than reweighting DMs with respect to a given target unnormalized density.

## 6 Numerical experiments in a realistic setting

In this section, we evaluate the performance of DM-based aMC-BGs in realistic setting **(B)**. This implies that the true dynamics are no longer available and are instead replaced by estimated dynamics driven by learned log-densities and scores. In particular, we assume that we do not have access to second-order information (*i.e.*, the Hessians of the log-densities), as it is often the case in practice. This restricts us to only using zeroth and first order diffusion-based aMC samplers. The purpose of this approach is to compare the practical performance of these samplers with their ideal counterparts described in Sections 3 and 4, which are affected only by statistical and time-discretization errors.

### 6.1 Log-density and score learning framework

**Architecture design.** To evaluate DM-aMC BGs under realistic constraints, we first learn DM log-densities and scores simultaneously using available samples. To do so, we model the log-density $\log p_t(x)$ by a scalar-valued neural network $(t, x) \mapsto -\mathcal{E}_t^\theta(x)$, and deduce an approximation of the score function $\nabla \log p_t(x)$ by taking the negative gradient of $\mathcal{E}^\theta$, denoted by $\mathbf{s}_t^\theta(x) = -\nabla_x \mathcal{E}_t^\theta(x)$. To ensure correctness at $t = t_0$ close to 0, we compare two common architectures.

(a) *Pinned*: we first consider the pinned architecture (Phillips et al., 2024; Zhang et al., 2026), defined as

$$\mathcal{E}_t^\theta(x) = (1 - f^\theta(t) + f^\theta(t_0))\mathcal{E}(x) + (f^\theta(t) - f^\theta(t_0))g_t^\theta(x) , \tag{26}$$

where $f^\theta : [0, T] \to \mathbb{R}$ is a neural network that solely takes time as input, and $g^\theta : [0, T] \times \mathbb{R}^d \to \mathbb{R}$ is another neural network conditioned on both $t$ and $x$.

While this setting ensures exact recovery of the target distribution $\pi$ at $t_0$, it is known to be difficult to train (Du et al., 2025), motivating the consideration of the next architecture.

(b) *Hardcoded*: the second architecture is an unconstrained variant inspired by the preconditioned score network used in (Karras et al., 2022; Thornton et al., 2025). Since it does not enforce any boundary condition at $t = t_0$, we explicitly correct this during sampling by replacing $\mathbf{s}_{t_0}^\theta$ with $-\nabla\mathcal{E}$ and $\mathcal{E}_{t_0}^\theta$ with $\mathcal{E}$. While this approach offers more flexibility during training, it may lead to inaccurate behavior at inference.

**Loss design.** For each neural network, we consider seven learning approaches. We restate their expression in Appendix A.3 and provide training details in Appendix D.2. These losses are denoted as follows:

- (DSM): *Denoising Score Matching* objective (Song et al., 2021; Karras et al., 2022),
- (TSM+DSM): *Target Score Matching* objective (Bortoli et al., 2024) with DSM regularization,
- (tSM+DSM): *Time Score Matching* objective (Yu et al., 2025; Guth et al., 2025b) with DSM regularization,
- (LFPE+DSM): DSM objective with *Log-density Fokker Planck Equation* regularization (Shi et al., 2024),
- (aLFPE+DSM): DSM objective with *approximated LFPE* regularization (Plainer et al., 2025),
- (RNE+DSM): DSM objective with *Radon-Nikodym Estimator* regularization (He et al., 2026),
- (DiffCLF+DSM): DSM objective with *Diffusive Classification* regularization (OuYang et al., 2026).

## 6.2 DM-BGs via aMC seem inherently limited by log-density approximation

**DM-BGs fail in practice.** Figure 4 compares zeroth and first order DM-based aMC-BGs in realistic setting **(B)** on the *TwoModes* intermediate difficulty target, for all DM training objectives introduced above. We consider: (a) the standard aMC setting (red bars); (b) aMC samplers based on first-order stochastic transition kernels (blue bars); and (c) aMC samplers based on first-order deterministic transition maps (yellow bars). Each BG is combined with both the hardcoded architecture (bar hatching) and the pinned architecture (dot hatching). We also report:

- classical tempering-based aMC samplers (grey bars), that were shown to be less accurate than DM-based standard aMC samplers in idealized setting **(A)**; see Section 3.1;
- simulations of the reverse SDE (5) and ODE (6) driven by the approximate score;
- semi-realistic DM-based aMC-BGs (no hatching), where the diffusion-path densities are the analytic ones, as in idealized setting **(A)**, while the learned score is used.

Within each setting described above, we report the best result over $K \in \{32, 64, 128, 256\}$ for readability. Overall, for each aMC class and each architecture, the three realistic DM-based BG variants achieve nearly indistinguishable performance, with no clear improvement over the tempering baseline and, in some cases, a degradation. This contrasts sharply with the idealized setting, where the diffusion-based deterministic approach consistently outperformed both the tempering path and the other diffusion-based alternatives across all aMC variants. More precisely, we observe that:

(i) *For the Hardcoded architecture*, the resulting BGs perform noticeably worse than the learned reverse SDE/ODE baselines. This suggests that the poor performance is not primarily due to the learned scores, but rather to inaccuracies in the learned log-densities. This interpretation is further supported by the semi-ideal experiments: when only densities are exact, the behavior of the aMC samplers improves substantially, and the results better match those observed in the idealized regime.

(ii) *For the Pinned architecture*, the conclusions are even less favorable. This setting is highly prone to training failures, and the learned reverse SDE/ODE simulations already produce strongly biased samples, indicating that the score functions themselves are poorly learned in the considered multi-modal settings.

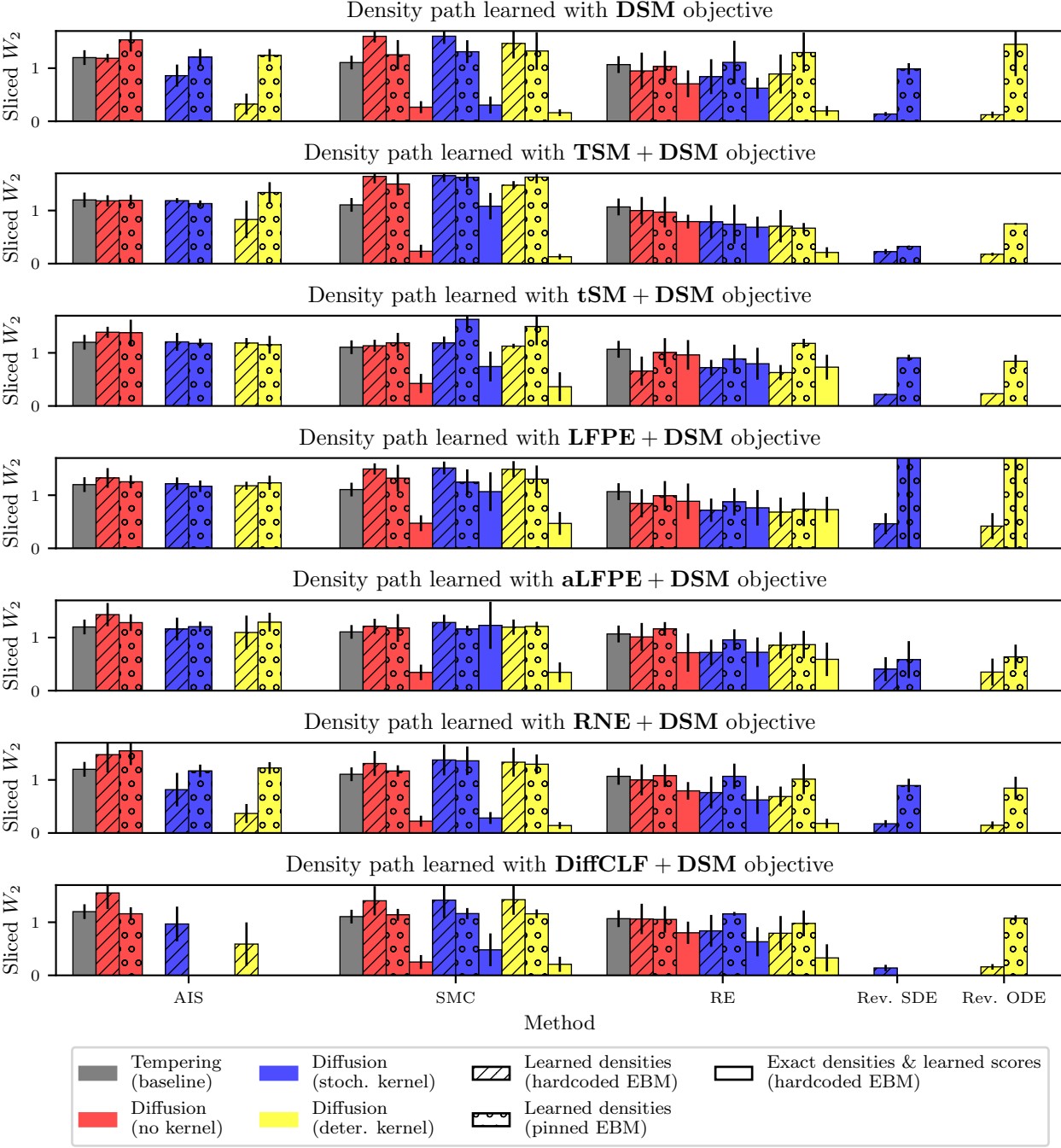

Figure 4: **Realistic results of DM-based aMC-BG**, when targeting *TwoModes* distribution with intermediate difficulty (Medium distance – Medium dimension) in setting **(B)** : **(From top to bottom)** the DM is trained via TSM+DSM, tSM+DSM, aLFPE+DSM, RNE+DSM or DiffCLF+DSM objective with identical computational budget. Each group of bars with the same color corresponds to a specific aMC method, except for the last two groups on the right which shows the baseline obtained by directly simulating the reverse SDE (5) and ODE (6). Bar colors are consistent with those displayed in Figures 1 and 3. On the other hand, hatching denotes the nature of neural approximator. For each method and objective, the number of levels $K$ was optimized individually so as to display the best expected result. Each result is averaged over 8 runs with 8,192 samples per run. Missing bars correspond to numerical failures during training. We observe that DM-BGs generally underperform compared to directly leveraging the DM alone (*i.e.*, simulating the reverse SDE/ODE), and rarely surpass the classic tempering methods. Running the same realistic experiments on the remaining *TwoModes* and *ManyModes* targets led to the same negative conclusions for all density-learning methods. To avoid overloading the manuscript with redundant results, we only report the numerical results for the *ManyModes* instance with 16 modes in Appendix D.3.

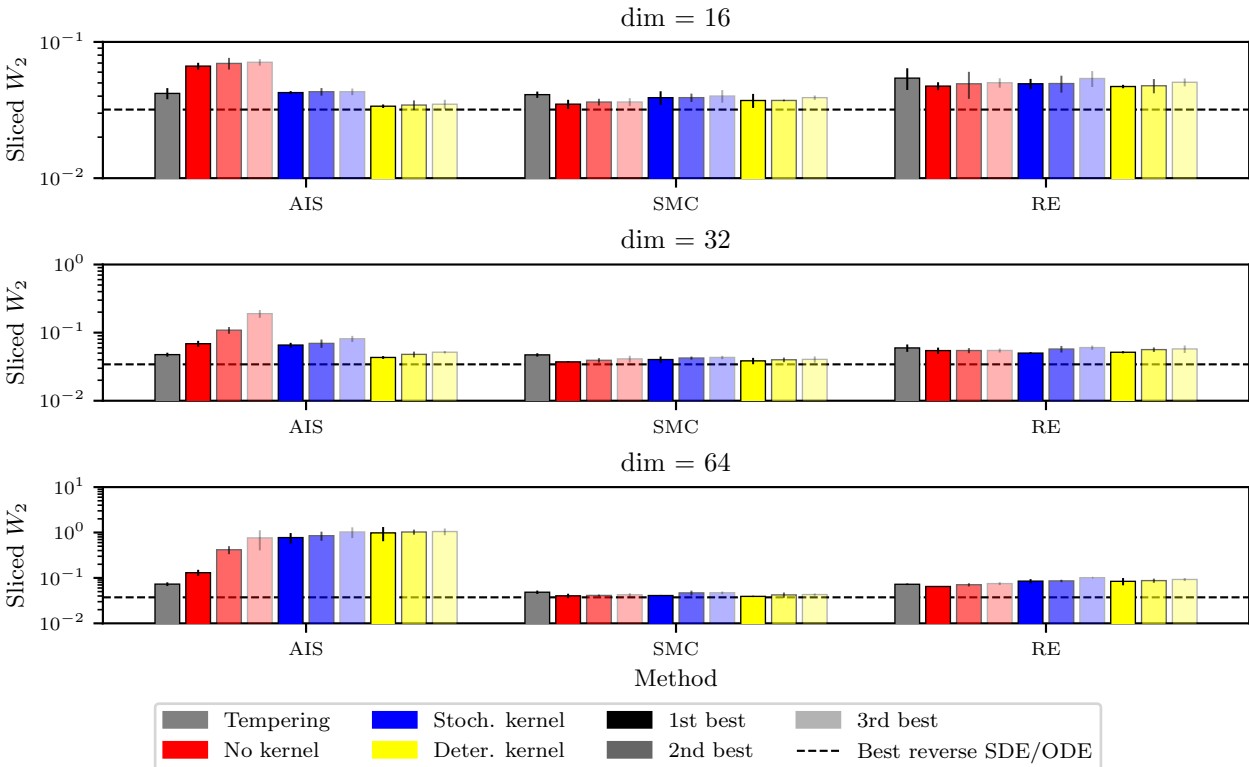

Figure 5: **Performance of DM-based aMC-BGs on *ManyWell*** in practical setting **(B)**, for $d = 16$ (top), $d = 32$ (middle), $d = 64$ (bottom). Colors denote AIS, SMC, or RE configurations, as in Figures 1, 3 and 4; gray marks conventional tempering baselines. The dotted line gives the better of the reverse ODE (6) and SDE (5) baselines. Results are aggregated over density-learning objectives, architectures, and numbers of levels, showing only the top three configurations per dimension and aMC-BG variant (decreasing intensity from best to third). Each bar averages 8 runs of 8,192 samples. Uncalibrated diffusion sampling consistently beats the tempering baselines, but its extension to BGs brings only marginal, largely configuration-independent gains: overall, diffusion-based BGs perform on par with, or slightly better than, tempering, while remaining substantially worse than the ODE/SDE baselines. Additional metrics are given in Appendix D.3.

We complement the practical experiments on Gaussian-mixture targets with results on several instances of the *ManyWell* distribution, for which no corresponding idealized regime is available; see Figure 5. We retain the same color coding for the DM-based aMC variants as in Figure 4, and additionally report the corresponding tempering results and the best reverse ODE/SDE result obtained across density-learning methods. Since architecture and training objective prove to be of secondary importance, we do not present separate results for every combination as in Figure 4; instead, for each class of DM-based aMC methods, we display only the three best-performing configurations.

The ManyWell results largely mirror those obtained in the practical Gaussian-mixture setting: across most aMC methods, the three DM-based BG variants achieve nearly indistinguishable performance, with AIS a partial exception, as the ordering observed in the idealized setting appears to persist for $d \in \{16, 32\}$. Diffusion-based BGs do not yield a clear improvement over their tempering baselines (the largest, though still marginal, gains appear for SMC) and none of the variants outperforms the full reverse ODE/SDE simulations. These findings highlight a key practical limitation of diffusion-based BGs on challenging multi-modal targets.

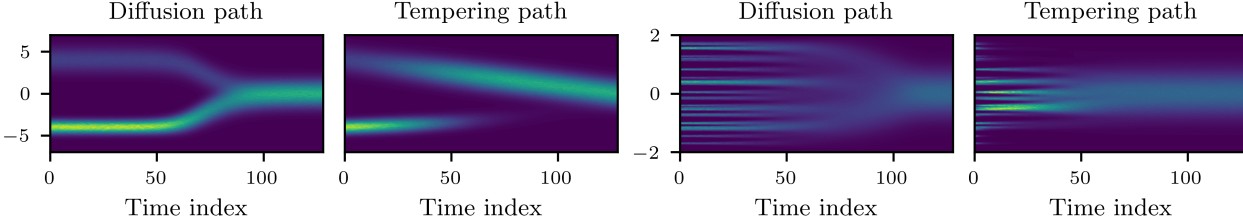

Figure 6: **Exact density paths bridging $\pi^{\mathbf{base}}$ (last time index) to 1D Gaussian mixtures (first time index). (Left)**: the target is an instance of *TwoModes* defined as $(3/4)\mathrm{N}(-4, 0.5^2) + (1/4)\mathrm{N}(+4, 1)$, **(Right)** the target is the 1D instance of *ManyModes* with 32 modes, **(First and third columns)** diffusion density path, **(Second and fourth columns)** tempering density path. Transport is made on time interval $[0, 1]$ with 128 timesteps. We observe that the tempering path shows clear mode switching for both of the targets: in the case of the *TwoModes* target, the strongest mode emerges abruptly, while the weakest modes appear rapidly for the *ManyModes* target. On the other hand, the mode weights in the diffusion path remain stable over time, making it more favorable for aMC.

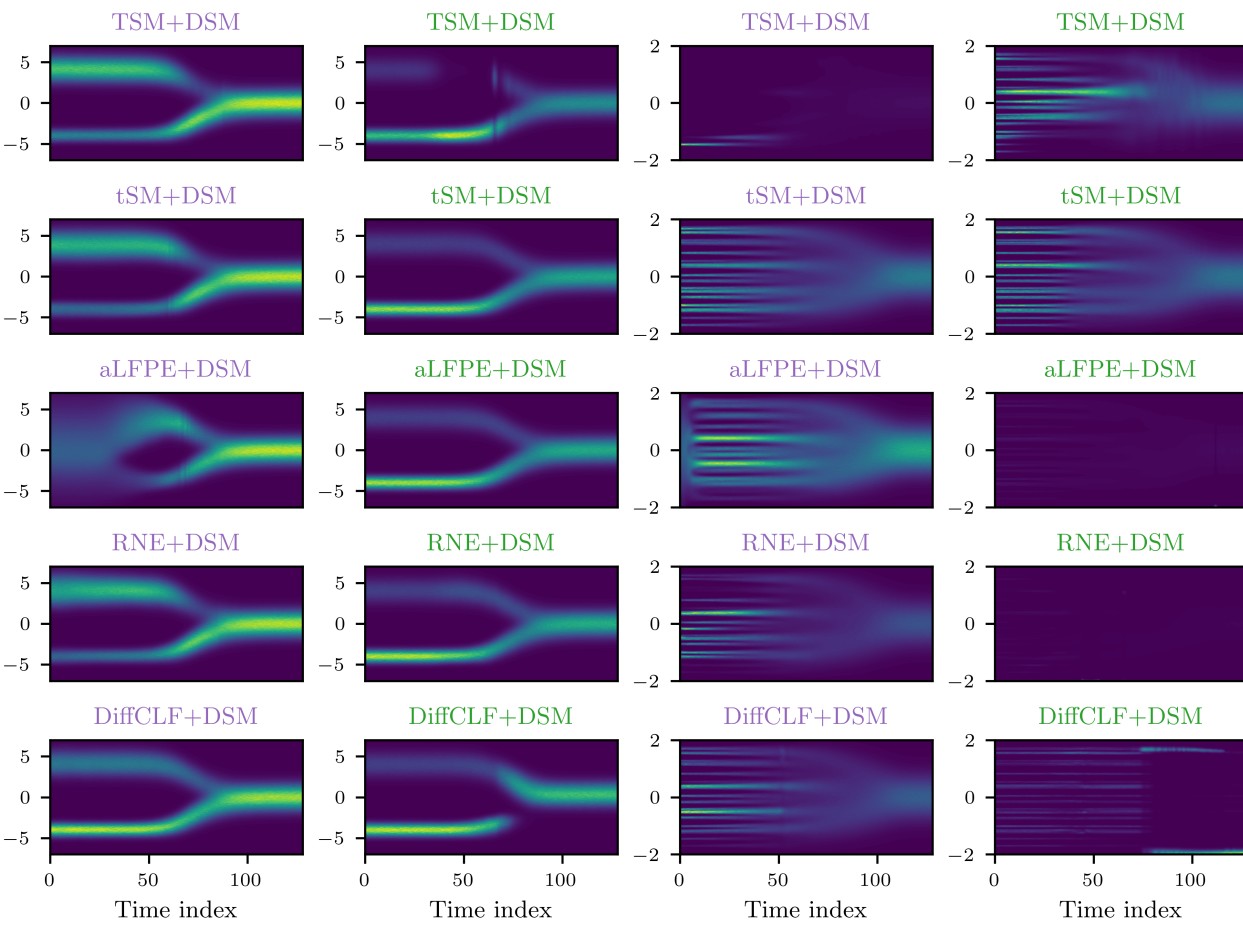

Figure 7: **Learned diffusion density paths bridging $\pi^{\mathbf{base}}$ (last time index) to the same targets as in Figure 6. (From top to bottom)** the DM is trained via TSM+DSM, tSM+DSM, aLFPE+DSM, RNE+DSM or DiffCLF+DSM objective, with identical computational budget, **(Left)** *TwoModes* target, **(Right)** *ManyModes* target, **(First and third columns)** use of hardcoded EBM, **(Second and fourth columns)** use of pinned EBM. The same plotting configuration as in Figure 6 is used here. When using the hardcoded architecture for both targets, we observe that the density is well learned for large times (near $\pi^{\mathbf{base}}$), but often fails to recover the exact mode weights for small times, thus highlighting the mode blindness of related training objectives. While using the pinned architecture enables to closely recover the exact diffusion path for the *TwoModes* target, we observe that this strategy is not successful when increasing the number of modes, with the notable exception of tSM+DSM objective.

**DM-BG failure cases could be attributed to mode switching in learned log-densities.** The semi-ideal experiments of Figure 4, where exact diffusion-path densities are paired with learned scores, point directly at the log-density estimation as the bottleneck: aMC performance recovers substantially as soon as densities are exact, even though scores remain learned. We hypothesize that the underlying cause is the inherent *mode blindness* of most of the DM training objectives considered here, which likely induces mode switching in the learned density path. This limitation of score-based learning is well documented (Wenliang & Kanagawa, 2021; Zhang et al., 2022; Shi et al., 2024) and affects all divergences derived from the Fisher divergence or Stein discrepancy: on multi-modal distributions with well-separated modes, these divergences cannot distinguish distributions sharing the same mode locations but differing in mode proportions. Indeed, the score is independent of the normalizing constant and, when evaluated within a single mode, is unaware of the others. Score matching therefore tends to recover the correct shape within each mode (*i.e.*, accurate gradients) but with incorrect relative weights. Figure 7 illustrates this phenomenon for the same subset of DM training objectives as in Figure 4 on 1D Gaussian mixtures (full results in Appendix D.3), and can be directly compared to the ground-truth diffusion and tempering density paths in Figure 6.

> **The ubiquitous curse of mode blindness for log-density estimation ?**
>
> Although mode blindness has been empirically documented for the DSM objective, we emphasize that the same issue also affects TSM, due to its score-based formulation. Nevertheless, recent diffusion-based sampling methods have used TSM to learn log-densities (Phillips et al., 2024; Zhang et al., 2026) in order to incorporate DMs within aMC samplers. By the same reasoning, mode blindness also affects log-density estimation based on score-distillation losses (Thornton et al., 2025; Akhound-Sadegh et al., 2026), which may harm the accuracy of SMC sampling in related inference-time alignment tasks.
>
> The mode blindness of the other training objectives considered in this paper has been recently investigated by OuYang et al. (2026): their Appendix C theoretically establishes that score matching, time score matching, and Fokker-Planck regularization all suffer from this issue, while their Proposition A.1 shows that RNE coincides with the LFPE objective in the dense-schedule limit, implying the same blindness. The case of tSM is more nuanced, as the authors argue that its susceptibility to mode blindness depends on the target – a mixed behavior also visible in Figure 7 with the hardcoded architecture, where tSM predicts incorrect mode weights on *TwoModes* but accurate ones on *ManyModes*. To address these limitations, OuYang et al. (2026) propose DiffCLF. While Figure 7 suggests that DiffCLF indeed mitigates mode blindness in simpler cases, the same figure shows it struggles on the *ManyModes* target, consistent with the authors' own observation.

Overall, we conjecture that mode switching significantly hinders aMC methods, as well for learned diffusion density paths as for tempering density paths. In SMC (including DM-enhanced variants), resampling must continually correct for imbalanced mode weights, which becomes increasingly challenging in high dimensions. Similarly, in RE, communication between chains is disrupted when mode alignment across levels is inconsistent, although we observe that it may be compensated for by the possibility of moving back and forth between levels during sampling procedure. This instability explains the poor performance of the learned path in Figure 4, even when the forward and backward transition kernels (both deterministic and stochastic) are accurate due to well-learned scores.

# 7 Conclusion & Limitations

This work revisits the design of Boltzmann Generators by replacing the standard normalizing-flow/importance-sampling backbone with a diffusion-model backbone embedded in annealed Monte Carlo. We first unify and review prior DM-aMC approaches, which exploit diffusion-induced *stochastic* denoising kernels to facilitate transitions between annealing levels, and we then introduce and study *deterministic* counterparts based on diffusion-derived transport maps. To compare these methods, we conduct an empirical study on multi-modal target distributions, emphasizing challenging characteristics such as inter-mode separation, number of modes, and dimensionality. Our analysis proceeds in two stages: we (i) isolate inference effects by assuming a perfectly learned DM, and (ii) turn to a realistic setting where the DM is trained from data.

In the idealized regime, empirical metrics reveal a non-zero discrepancy between the ground-truth target and the distribution induced by the resulting BG, despite perfect model knowledge. This indicates that aMC inference error alone can produce measurable bias. In this setting, *first-order* stochastic denoising kernels (score-only) often fail to improve over standard aMC baselines, whereas second-order kernels (incorporating Hessian information) and deterministic transitions yield substantially better results. Importantly, our deterministic construction based on a Hutchinson-type estimator remains competitive even without explicit Hessian access, suggesting that deterministic transport can recover much of the benefit of second-order information while relaxing its most demanding requirement.

In the learned regime (e.g., score-matching-like training objectives), the picture changes markedly: the resulting BGs systematically fail across our multi-modal benchmarks, even when the learned scores appear accurate. Our MoG experiments point to inaccuracies in DM log-density estimation as the primary culprit. Specifically, the obtained estimates are mostly mode-blind, as they fail to accurately represent relative mode proportions along the diffusion path in regions where the modes are well separated. As a consequence, such errors directly disrupt sampling and can dominate any gains from improved transitions. In other words, high-quality score estimates are not sufficient to guarantee successful BG construction when the correction step relies on unreliable log-density approximations.

In the spirit of Grenioux et al. (2025), our goal is not to demonstrate scalability but to expose and analyze the fundamental limitations of diffusion-based aMC-BGs in a simple, fully controlled benchmark. The underlying rationale is that methods that do not succeed in these elementary multi-modal settings are unlikely to behave reliably on more complex targets with many modes or ill-conditioned energy landscapes. Accordingly, this work emphasizes failure mechanisms over performance claims, consistent with our largely negative conclusions.

A natural direction for future work concerns the modeling side: the main bottleneck in realistic settings is the mode blindness of current DM log-density estimation techniques, and addressing it appears necessary for reliable sampling. Beyond building a single BG, training schemes that mitigate or eliminate this issue would open the door to using the proposed aMC machinery as an inner loop in iterative diffusion-based training procedures tailored for sampling – departing from the one-shot correction perspective, in the spirit of adaptive, data-free training strategies (Gabrié et al., 2022; Phillips et al., 2024; Akhound-Sadegh et al., 2024).

## Acknowledgments

We warmly thank Marylou Gabrié, Alain Durmus, and José Miguel Hernández-Lobato for the insightful discussions and reflections that helped shape and refine this work. Their perspectives and feedback have been invaluable throughout the development of the ideas presented here. This work was performed using HPC resources from GENCI–IDRIS (AD011014860R1, AD011014860R2 and AD011015234R1). This work received government funding managed by the French National Research Agency under France 2030, reference ANR-23-IACL-0005.

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

## Organization of the supplementary

The appendix is organized as follows. Appendix A summarizes general facts that will be useful for proofs and corresponding computations. In Appendix B, we describe the general framework of noising diffusion processes, as well as the particular case of Variance-Preserving (consistently used in our experiments) and Variance-Exploding schemes: we notably detail there the formulas related to SDE/ODE integrators and to the computation of the log-determinant terms arising from the use of deterministic transitions in DM-aMC samplers (see Section 4). In Appendix C, we provide the proofs of all theoretical results dispensed in Section 4. Finally, we precisely detail our experimental setting in Appendix D, along with additional numerical results.

## A    Preliminaries

### A.1    Useful lemmas

**Lemma 4** (Power series expansion of the matrix logarithm)**.** *Let* $(\alpha, \beta) \in \mathbb{R}^2$. *For any matrix* $\mathrm{M} \in \mathbb{R}^{d \times d}$ *satisfying* $\|\mathrm{M}\| < \min(1/|\alpha|, 1/|\beta|)$, *the following identities hold*

$$
\log\left[(\mathrm{I}_d - \beta\mathrm{M})^{-1}(\mathrm{I}_d + \alpha\mathrm{M})\right] = \sum_{i=1}^{\infty} \frac{\beta^i - (-1)^i \alpha^i}{i} \mathrm{M}^i, \quad \log\left[(\mathrm{I}_d + \beta\mathrm{M})^{-1}(\mathrm{I}_d - \alpha\mathrm{M})\right] = \sum_{i=1}^{\infty} \frac{(-1)^i \beta^i - \alpha^i}{i} \mathrm{M}^i,
$$

*where* $\log$ *denotes the matrix logarithm.*

*Proof.* This is an immediate corollary from (Hall, 2015, Theorem 3.6). $\qquad\square$

**Corollary 5.** *Let* $(c_1, c_2, c_3) \in \mathbb{R}^3$ *with* $c_1 \neq 0$. *Let* $\mathrm{M} \in \mathbb{R}^{d \times d}$ *be a matrix satisfying* $\|\mathrm{M}\| < \min(1/|c_2|, 1/|c_3|)$. *Define the matrices*

$$
\mathrm{M}_1 = c_1(\mathrm{I}_d + c_2\mathrm{M})^{-1}(\mathrm{I}_d - c_3\mathrm{M}), \; \mathrm{M}_2 = c_1^{-1}(\mathrm{I}_d - c_3\mathrm{M})^{-1}(\mathrm{I}_d + c_2\mathrm{M}).
$$

*Then we have*

$$
\log|\det \mathrm{M}_1| = d\log|c_1| + \sum_{i=1}^{\infty} \frac{(-1)^i c_2^i - c_3^i}{i} \operatorname{Tr}[\mathrm{M}^i], \; \log|\det \mathrm{M}_2| = -d\log|c_1| + \sum_{i=1}^{\infty} \frac{c_3^i - (-1)^i c_2^i}{i} \operatorname{Tr}[\mathrm{M}^i].
$$

*Proof.* Consider such $(c_1, c_2, c_3)$ and such matrix $\mathrm{M}$. Note that the assumption on $c_2$ and $c_3$ guarantees the invertibility of $\mathrm{I}_d + c_2\mathrm{M}$ and $\mathrm{I}_d - c_3\mathrm{M}$. Regarding $\mathrm{M}_1$, we have

$$
\begin{aligned}
\log|\det M_1| &= d\log|c_1| + \log\left|\det\left((\mathrm{I}_d + c_2\mathrm{M})^{-1}(\mathrm{I}_d - c_3\mathrm{M})\right)\right| \\
&= d\log|c_1| + \log\left|\det(\mathrm{I}_d + c_2\mathrm{M})^{-1}\det(\mathrm{I}_d - c_3\mathrm{M})\right| \\
&= d\log|c_1| - \log|\det(\mathrm{I}_d + c_2\mathrm{M})| + \log|\det(\mathrm{I}_d - c_3\mathrm{M})| \\
&= d\log|c_1| - \log\det(\mathrm{I}_d + c_2\mathrm{M}) + \log\det(\mathrm{I}_d - c_3\mathrm{M}) && \text{(Behrmann et al., 2019, Lemma 6)} \\
&= d\log|c_1| + \log\det\left((\mathrm{I}_d + c_2\mathrm{M})^{-1}(\mathrm{I}_d - c_3\mathrm{M})\right) \\
&= d\log|c_1| + \operatorname{Tr}\log\left((\mathrm{I}_d + c_2\mathrm{M})^{-1}(\mathrm{I}_d - c_3\mathrm{M})\right). && \text{(Hall, 2015, Theorem 3.10)}
\end{aligned}
$$

Hence, we obtain the first result of Corollary 5 by using the second statement of Lemma 4 with $\beta = c_2$ and $\alpha = c_3$. Similar computations with $\mathrm{M}_2$ lead to the second result. $\qquad\square$

### A.2    General results on SDE/ODE Exponential Integration

**Assumption 2** (Integrability conditions on $f$ and $g$)**.** *Coefficients* $f : [0, T] \to \mathbb{R}$ *and* $g : [0, T] \to (0, \infty)$ *are such that (a)* $f$ *is integrable on* $(0, T)$ *and (b)* $g$ *is integrable on* $(0, T)$.

**Lemma 6** (SDE Exponential Integration). *Let $T > 0$ and $b \in \mathbb{R}^d$. Consider the SDE defined on $[0, T]$ by $\mathrm{d}Y_t = f(t)(Y_t + b)\mathrm{d}t + g(t)\mathrm{d}B_t$, where coefficients $f$ and $g$ verify Assumption 2. Then, for any pair of time-steps $(s, t)$ such that $T \geq t > s \geq 0$, the conditional distribution of $Y_t$ given $Y_s = y_s \in \mathbb{R}^d$, denoted by $q_{t|s}(\cdot|y_s)$, verifies*

$$q_{t|s}(\cdot|y_s) = \mathrm{N}\left(\exp(\textstyle\int_s^t f(u)\mathrm{d}u)y_s + \left(\exp(\textstyle\int_s^t f(u)\mathrm{d}u) - 1\right)b, \textstyle\int_s^t g^2(u)\exp(2\textstyle\int_u^t f(r)\mathrm{d}r)\mathrm{d}u\,\mathrm{I}_d\right).$$

*Proof.* Assume Assumption 2. Define the function $\zeta : t \in [0, T] \to \exp(-\int_0^t f(u)\mathrm{d}u)$ and consider the stochastic process $(Z_t)_{t \in [0,T]}$ defined by $Z_t = \zeta(t)Y_t$ for any $t \in [0, T]$. By Îto's formula, we have $\mathrm{d}Z_t = f(t)\zeta(t)b\mathrm{d}t + \zeta(t)g(t)\mathrm{d}B_t = -\dot{\zeta}(t)b\mathrm{d}t + \zeta(t)g(t)\mathrm{d}B_t$. Therefore, for any time-steps $(s, t)$ such that $T \geq t > s \geq 0$, we have

$$\zeta(t)Y_t - \zeta(s)Y_s = \{\zeta(s) - \zeta(t)\}b + \textstyle\int_s^t \zeta(u)g(u)\mathrm{d}B_u,$$

and then

$$Y_t = \exp(\textstyle\int_s^t f(u)\mathrm{d}u)Y_s + \left(\exp(\textstyle\int_s^t f(u)\mathrm{d}u) - 1\right)b + \textstyle\int_s^t g(u)\exp(\textstyle\int_u^t f(r)\mathrm{d}r)\mathrm{d}B_u,$$

which gives the result using Îto's isometry and that $Y_s$ is independent from $(B_t - B_s)_{t \in [s,T]}$. $\square$

The following lemma can be seen as the limit of Lemma 6 in the deterministic regime, *i.e.*, when $g(t) = 0$ for any $t \in [0, T]$.

**Lemma 7** (ODE Exponential Integration). *Let $T > 0$ and $b \in \mathbb{R}^d$. Consider the ODE defined on $[0, T]$ by $\mathrm{d}Y_t = f(t)[Y_t + b]\mathrm{d}t$, where coefficient $f$ verifies Assumption 2. Then, for any pair of time-steps $(s, t)$ such that $T \geq t > s \geq 0$, the ODE solution $Y_t$ given $Y_s = y_s \in \mathbb{R}^d$ is defined by*

$$Y_t = \exp\left(\textstyle\int_s^t f(u)\mathrm{d}u\right)y_s + \left(\exp\left(\textstyle\int_s^t f(u)\mathrm{d}u\right) - 1\right)b.$$

*Proof.* Let $0 \leq s < t \leq T$, set $Z_t = \exp(-F(t))Y_t$, where $F(t) = \int_0^t f(u)\mathrm{d}u$, then

$$\mathrm{d}Z_t = f(t)\exp(-F(t))b\mathrm{d}t,$$

which implies that

$$Z_t = Z + (\exp(-F(s)) - \exp(-F(t)))\,b,$$

which gives the result. $\square$

## A.3 Review of score and energy matching methods

Consider the noising diffusion process given by the SDE (2). In this section, we review a selection of methods used to learn the scores, the time scores and/or the log-densities (*i.e.*, energies) of the marginal distributions $(p_t)_{t \in [0,T]}$ associated to this process, based on samples from the target distribution $\pi$ with unnormalized density $\gamma$. While the presented score matching (and time score matching) techniques have widely been experimented within the diffusion model community, the evoked log-density estimation (also referred to as energy matching) approaches are much more recent, and only provide a small glimpse into the pretty young field of research to which they belong. In the following, we will denote $r(t) = S(t)\sigma(t)$, where coefficients $S$ and $\sigma$ are introduced in (3) to marginally characterize diffusion models. We adopt a consistent notation for neural networks: $\mathbf{U}_t^\theta$ is used to learn the log-density $\log p_t$, $\mathbf{s}_t^\theta$ to learn the score $\nabla \log p_t$, and $\mathbf{u}_t^\theta$ to learn the time score $\partial_t \log p_t$.

**Denoising Score Matching (DSM) (Song et al., 2021).** This is the standard score matching loss used in the diffusion-based generative modeling literature. It relies on the so-called Tweedie identity

$$\nabla \log p_t(x_t) = \mathrm{E}[\nabla \log q_{t|0}(x_t|X_0)] \ , \ X_0 \sim q_{0|t}(\cdot|x_t) \ ,$$

where $q_{t|0}$ is the *tractable* noising transition kernel between times 0 and $t$, see (7), and $q_{0|t}$ is the related denoising transition kernel, which verifies by Bayes property $q_{0|t}(x_0|x_t) \propto \gamma(x_0)q_{t|0}(x_t|x_0)$. This gives rise to the following objective for estimating the score function $(t, x) \mapsto \nabla \log p_t(x)$ by a neural network $(t, x) \mapsto \mathbf{s}_t^\theta(x)$

$$\mathcal{L}_{\mathrm{DSM}}(\theta) = \mathrm{E}\left[\left\|\mathbf{s}_t^\theta(X_t) + \frac{Z}{r(t)}\right\|_2^2\right] \ , \ X_t = S(t)X_0 + r(t)Z \ ,$$

where $t \sim \mathrm{U}(0, T)$, $X_0 \sim \pi$ and $Z \sim \mathrm{N}(0, \mathrm{I}_d)$. In practice, one rather uses a reweighted version of this objective given by

$$\tilde{\mathcal{L}}_{\mathrm{DSM}}(\theta) = \mathrm{E}\left[r^2(t)\left\|\mathbf{s}_t^\theta(X_t) + \frac{Z}{r(t)}\right\|_2^2\right] = \mathrm{E}\left[\left\|r(t)\mathbf{s}_t^\theta(X_t) + Z\right\|_2^2\right] \ .$$

To further reduce its variance with respect to the noise variable, one may consider applying the antithetic trick on $Z$ variable and thus obtain the loss function

$$\tilde{\mathcal{L}}_{\mathrm{DSM}}^{\mathrm{anti}}(\theta) = \mathrm{E}\left[\frac{1}{2}\left\|r(t)\mathbf{s}_t^\theta(X_t) + Z\right\|_2^2 + \frac{1}{2}\left\|r(t)\mathbf{s}_t^\theta(X_t^-) - Z\right\|_2^2\right] \ ,$$

$$\text{with } X_t = S(t)X_0 + r(t)Z \ , \ X_t^- = S(t)X_0 - r(t)Z.$$

In practice, the DSM objective described above still exhibits high variance. To ensure its robustness and stability for large-scale applications, an equivalent objective, coined EDM, was proposed by Karras et al. (2022), which specifically relies on preconditioning guidelines for the neural network $\mathbf{s}^\theta$. In our experiments, the "DSM objective" will systematically refer to this specific EDM training loss, enhanced with the antithetic trick, whose success has been widely proven over the last few years for generative tasks.

**Target Score Matching (TSM) (Bortoli et al., 2024).** Alternatively, by operating a change-of-variable in the Tweedie's formula, the following identity also holds

$$\nabla \log p_t(x_t) = \frac{1}{S(t)}\mathrm{E}[\nabla \log \gamma(X_0)] \ , \ X_0 \sim q_{0|t}(\cdot|x_t) \ .$$

This gives rise to the following objective for estimating the score function $(t, x) \mapsto \nabla \log p_t(x)$ by a neural network $(t, x) \mapsto \mathbf{s}_t^\theta(x)$

$$\mathcal{L}_{\mathrm{TSM}}(\theta) = \mathrm{E}\left[\left\|\mathbf{s}_t^\theta(X_t) - \frac{\nabla \log \gamma(X_0)}{S(t)}\right\|_2^2\right] \ , \ X_t = S(t)X_0 + r(t)Z \ ,$$

where $t \sim \mathrm{U}(0, T)$, $X_0 \sim \pi$ and $Z \sim \mathrm{N}(0, \mathrm{I}_d)$. In practice, one rather uses a reweighted version of this objective given by

$$\tilde{\mathcal{L}}_{\mathrm{TSM}}(\theta) = \mathrm{E}\left[S^2(t)\left\|\mathbf{s}_t^\theta(X_t) - \frac{\nabla \log \gamma(X_0)}{S(t)}\right\|_2^2\right] = \mathrm{E}\left[\left\|S(t)\mathbf{s}_t^\theta(X_t) - \nabla \log \gamma(X_0)\right\|_2^2\right] \ ,$$

which itself can be improved via the antithetic trick as

$$\tilde{\mathcal{L}}_{\mathrm{TSM}}^{\mathrm{anti}}(\theta) = \mathrm{E}\left[\frac{1}{2}\left\|S(t)\mathbf{s}_t^\theta(X_t) - \nabla \log \gamma(X_0)\right\|_2^2 + \frac{1}{2}\left\|S(t)\mathbf{s}_t^\theta(X_t^-) - \nabla \log \gamma(X_0)\right\|_2^2\right] \ ,$$

$$\text{with } X_t = S(t)X_0 + r(t)Z \ , \ X_t^- = S(t)X_0 - r(t)Z.$$

In our experiments, the "TSM objective" will systematically refer to the training loss function $\tilde{\mathcal{L}}_{\mathrm{TSM}}^{\mathrm{anti}}$. As originally proposed by Bortoli et al. (2024), this loss can also be combined with preconditioning schemes to reduce its variance in practice; however, since those are not compatible with the preconditioning directives from Karras et al. (2022), we do not integrate them in our numerical experiments.

**Time Score Matching (tSM)(Guth et al., 2025a; Yu et al., 2025).** Interestingly, the time score function has a similar decomposition

$$\partial_t \log p_t(x_t) = \mathrm{E}[\partial_t \log q_{t|0}(x_t|X_0)] \ , \ X_0 \sim q_{0|t}(\cdot|x_t) \ .$$

Since the conditional time derivative $\partial_t \log q_{t|0}$ is as tractable as the conditional score $\nabla \log q_{t|0}$, this gives rise to the following objective for estimating the time score function $(t,x) \mapsto \partial_t \log p_t(x)$ by a neural network $(t,x) \mapsto \mathbf{u}_t^\theta(x)$

$$\mathcal{L}_{\mathrm{tSM}}(\theta) = \mathrm{E}\left[\left\|\mathbf{u}_t^\theta(X_t) - u_{\mathrm{tSM}}^{\mathrm{target}}(t, X_0, Z)\right\|_2^2\right] \ , \ X_t = S(t)X_0 + r(t)Z \ ,$$

where $u_{\mathrm{tSM}}^{\mathrm{target}}(t, x_0, z) = -[\dot{r}(t)/r(t)]\left(d - \|z\|^2\right) + [\dot{S}(t)/r(t)]x_0^\top z$, $t \sim \mathrm{U}(0,T)$, $X_0 \sim \pi$ and $Z \sim \mathrm{N}(0, \mathrm{I}_d)$. Similarly, this objective may include antithetic trick and rewrite as

$$\mathcal{L}_{\mathrm{tSM}}^{\mathrm{anti}}(\theta) = \mathrm{E}\left[\frac{1}{2}\left\|\mathbf{u}_t^\theta(X_t) - u_{\mathrm{tsM}}^{\mathrm{target}}(t, X_0, Z)\right\|_2^2 + \frac{1}{2}\left\|\mathbf{u}_t^\theta(X_t^-) - u_{\mathrm{tsM}}^{\mathrm{target}}(t, X_0, -Z)\right\|_2^2\right] \ ,$$
$$\text{with } X_t = S(t)X_0 + r(t)Z \ , \ X_t^- = S(t)X_0 - r(t)Z.$$

As such, it has been observed that the derived objective exhibits very high variance, even more than score matching methods. While Guth et al. (2025a) explore a reweighting precisely adjusted to the VE noising scheme, Yu et al. (2025) propose an alternative reweighting in the context of the VP noising scheme; we implement the latter formulation with the antithetic trick, to which the "tSM objective" will systematically refer in our experiments. Note that by including a change-of-variable into the expression of the time score, we may obtain a target-like version of the tSM objective given by

$$\mathcal{L}_{\mathrm{tTSM}}(\theta) = \mathrm{E}\left[\left\|\mathbf{u}_t^\theta(X_t) - u_{\mathrm{tTSM}}^{\mathrm{target}}(t, X_0, Z)\right\|_2^2\right] \ , \ X_t = S(t)X_0 + r(t)Z \ ,$$

where $u_{\mathrm{tTSM}}^{\mathrm{target}}(t, x_0, z) = -\left\{[\dot{S}(t)/S(t)]x_0 + [\sigma(t)\dot{r}(t)/r(t)]z\right\}^\top \nabla \log \gamma(x_0)$, $t \sim \mathrm{U}(0,T)$, $X_0 \sim \pi$ and $Z \sim \mathrm{N}(0, \mathrm{I}_d)$, along with its antithetic-like version

$$\mathcal{L}_{\mathrm{tTSM}}^{\mathrm{anti}}(\theta) = \mathrm{E}\left[\frac{1}{2}\left\|\mathbf{u}_t^\theta(X_t) - u_{\mathrm{tTSM}}^{\mathrm{target}}(t, X_0, Z)\right\|_2^2 + \frac{1}{2}\left\|\mathbf{u}_t^\theta(X_t^-) - u_{\mathrm{tTSM}}^{\mathrm{target}}(t, X_0, -Z)\right\|_2^2\right] \ ,$$
$$\text{with } X_t = S(t)X_0 + r(t)Z \ , \ X_t^- = S(t)X_0 - r(t)Z.$$

While this objective is enriched with the information of the target score $\nabla \log \gamma$, we did not use this objective in our experiments due to its variance instability during training procedure.

**Log-density Fokker-Planck-Equation (LFPE) (Lai et al., 2023; Shi et al., 2024; Sun et al., 2024).** A key property of the noising SDE (2) is that the induced log-densities $(p_t)_{t \in [0,1]}$ can be described by a partial differential equation called the *Fokker-Planck equation* (Øksendal, 2003), whose formulation can be written as

$$\partial_t \log p_t(x) = \mathcal{F}[\log p](t, x) \overset{\text{def}}{=} \frac{1}{2}g^2(t)\left[\mathrm{div}(\nabla \log p_t)(x) + \|\nabla \log p_t(x)\|_2^2\right] - f(t)\{d + x^\top \nabla \log p_t(x)\} \ ,$$

where div denotes the divergence operator defined by $\mathrm{div}\,\mathrm{F} = \mathrm{Tr}[\nabla \mathrm{F}]$. This gives rise to the following objective for estimating the log-density $(t,x) \mapsto \log p_t(x)$ by a neural network $(t,x) \mapsto \mathbf{U}_t^\theta(x)$

$$\mathcal{L}_{\mathrm{LFPE}}(\theta) = \mathrm{E}\left[\left\|\partial_t \mathbf{U}_t^\theta(X_t) - \mathrm{sg}\left\{\mathcal{F}[\mathbf{U}^\theta](t, X_t)\right\}\right\|_2^2\right] \ , \ X_t = S(t)X_0 + r(t)Z \ ,$$

where sg denotes the stop-gradient[8] operator with respect to parameter $\theta$, $t \sim \mathrm{U}(0, T)$, $X_0 \sim \pi$ and $Z \sim \mathrm{N}(0, \mathrm{I}_d)$. In this case too, we can derive an objective based on the antithetic trick

$$\mathcal{L}_{\mathrm{LFPE}}^{\mathrm{anti}}(\theta) = \mathrm{E}\left[\frac{1}{2}\left\|\partial_t \mathbf{U}_t^\theta\left(X_t\right) - \mathrm{sg}\left\{\mathcal{F}[\mathbf{U}^\theta](t, X_t)\right\}\right\|_2^2 + \frac{1}{2}\left\|\partial_t \mathbf{U}_t^\theta\left(X_t^-\right) - \mathrm{sg}\left\{\mathcal{F}[\mathbf{U}^\theta](t, X_t^-)\right\}\right\|_2^2\right],$$

$$\text{with } X_t = S(t)X_0 + r(t)Z \ , \ X_t^- = S(t)X_0 - r(t)Z.$$

In our experiments, the "LFPE objective" will always refer to the training loss $\mathcal{L}_{\mathrm{LFPE}}^{\mathrm{anti}}$.

**Approximate LFPE (aLFPE) (Plainer et al., 2025).** In the case where the target term $\mathcal{F}[\mathbf{U}^\theta]$ is not detached with respect to $\theta$ in $\mathcal{L}_{\mathrm{LFPE}}$, the main numerical burden lies in the computation of the divergence term $\mathrm{div}(\nabla \mathbf{U}^\theta)$ when the dimension is large. To reduce this overhead, Plainer et al. (2025) propose to instead consider a first-order statistical estimation of the residual term $\mathcal{R}^\theta(t, x) = \mathcal{F}[\mathbf{U}^\theta](t, x) - \partial_t \mathbf{U}_t^\theta(x)$ given by $\tilde{\mathcal{R}}^\theta(t, x) = \mathrm{E}_v[\tilde{\mathcal{R}}^\theta(t, x; v)]$, with $v \sim \mathrm{N}(0, \sigma^2 \mathrm{I}_d)$ for a small $\sigma > 0$, where[9]

$$\tilde{\mathcal{R}}^\theta(t, x; v) = \frac{1}{2}g^2(t)\left[\left(\frac{v}{\sigma}\right)^\top \frac{\nabla \mathbf{U}_t^\theta(x+v) - \nabla \mathbf{U}_t^\theta(x-v)}{2\sigma}\right]$$

$$+ \frac{1}{2}\left[\frac{1}{2}g^2(t)\left\|\nabla \mathbf{U}_t^\theta(x+v)\right\|_2^2 - f(t)\{d + (x+v)^\top \nabla \mathbf{U}_t^\theta(x+v)\} - \partial_t \mathbf{U}_t^\theta(x+v)\right]$$

$$+ \frac{1}{2}\left[\frac{1}{2}g^2(t)\left\|\nabla \mathbf{U}_t^\theta(x-v)\right\|_2^2 - f(t)\{d + (x-v)^\top \nabla \mathbf{U}_t^\theta(x-v)\} - \partial_t \mathbf{U}_t^\theta(x-v)\right].$$

This gives rise to the following objective

$$\mathcal{L}_{\mathrm{aLFPE}}(\theta) = \mathrm{E}\left[\left(\sum_{i=1}^N \tilde{\mathcal{R}}^\theta(t, X_t; v_i^{X_t})\right)\left(\sum_{j=1}^N \tilde{\mathcal{R}}^\theta(t, X_t; v_j^{X_t})\right)\right] \ , \ X_t = S(t)X_0 + r(t)Z \ ,$$

where $t \sim \mathrm{U}(0, T)$, $X_0 \sim \pi$, $Z \sim \mathrm{N}(0, \mathrm{I}_d)$ and $\{v_i^{X_t}, v_j^{X_t}\}_{j=1}^N$ are $2N$ independent samples from $\mathrm{N}(0, \sigma^2 \mathrm{I}_d)$ defined for each input $X_t$. Overall, this formulation avoids the need of the divergence computation (while maintaining backpropagation through the scores), at the cost of non-negligible statistical error. Following the guidelines from Plainer et al. (2025), we consistently set in our experiments $\sigma = 0.0001$, but choose $N = 64$ (instead of $N = 1$ as originally proposed) to reduce the variance of the loss, and bring it into the most favorable setting. We also choose to keep the use of auto-differentiation to compute the time derivative $\partial_t \mathbf{U}^\theta$ instead of using finite difference approximation as suggested by Plainer et al. (2025), as it brings more stability during training. In our experiments, we will systematically refer to this version of $\mathcal{L}_{\mathrm{aLFPE}}$ as the "aLFPE objective".

**Radon-Nikodym Estimator (RNE) (He et al., 2026).** Alternatively, a discrete-time formulation of the LFPE objective has been proposed to learn the log-densities $(\log p_t)_{t \in [0, T]}$, based on the Bayes's rule (ideally satisfied by DMs) stating that for any times $(s, t) \in [0, T]^2$ and any inputs $x_s$ and $x_t$, we have $p_t(x_t)q_{s|t}(x_s|x_t) = p_s(x_s)q_{t|s}(x_t|x_s)$, where $q_{s|t}$ and $q_{t|s}$ correspond to related stochastic transition kernels (see Section 2.1). Enforcing this consistency with log-densities can thus be translated into the following objective for estimating the log-density $(t, x) \mapsto \log p_t(x)$ by a neural network $(t, x) \mapsto \mathbf{U}_t^\theta(x)$

$$\mathcal{L}_{\mathrm{RNE}}(\theta) = \mathrm{E}\left[\left\|\mathbf{U}_t^\theta(X_t) - \mathbf{U}_s^\theta(X_s) - \mathrm{sg}\{\log q_{t|s}(X_t|X_s) - \log q_{s|t}^\theta(X_s|X_t)\}\right\|_2^2\right],$$

$$\text{with } X_s = S(s)X_0 + r(s)Z \ , \ X_t = [S(t)/S(s)]X_s + r(t)\{1 - \sigma^2(s)/\sigma^2(t)\}^{1/2}\tilde{Z} \ ,$$

---

[8]While cited related works did not consider detaching the term $\mathcal{F}[\mathbf{U}^\theta]$ with respect to $\theta$ in their respective formulation, we made this choice to avoid backpropagation through both first and second-order derivatives of $\mathbf{U}^\theta$, which was computationally infeasible in the high-dimensional settings considered in this paper. Nonetheless, we emphasize that, in our early experiments, we observed unchanged results on pure log-density estimation tasks for small dimensional settings, thereby suggesting that our methodology remains sound.

[9]Even though this objective features an additional term compared to the one stated in Equation (12) from Plainer et al. (2025), it is consistent with the related code available at `https://github.com/noegroup/ScoreMD`. This extra term actually originates from the use of the antithetic trick on the Gaussian variable $v$.

where sg denotes the stop-gradient operator with respect to parameter $\theta$, $(s,t) \sim \mathrm{U}\left(\{(t_k, t_{k+1}) : k \in [0, K-1]\}\right)$ with $\{t_k\}_{k=0}^{K-1}$ being a discretization of time interval $[0, T]$, $X_0 \sim \pi$ and $(Z, \tilde{Z}) \sim \mathrm{N}(0, \mathrm{I}_d) \otimes \mathrm{N}(0, \mathrm{I}_d)$. While $q_{t|s}$ denotes a *noising* transition kernel[10], that is tractable by (7), $q_{s|t}^\theta$ is an approximate Gaussian *denoising* transition kernel, that may be computed via the learned score $\nabla \mathbf{U}^\theta$. In this case too, one may consider the variant featuring the antithetic trick

$$
\begin{aligned}
\mathcal{L}_{\mathrm{RNE}}^{\mathrm{anti}}(\theta) = \mathrm{E}\Bigg[ & \frac{1}{2} \left\| \mathbf{U}_t^\theta\left(X_t\right) - \mathbf{U}_s^\theta\left(X_s\right) - \mathrm{sg}\{\log q_{t|s}(X_t|X_s) - \log q_{s|t}^\theta(X_s|X_t)\} \right\|_2^2 \\
& + \frac{1}{2} \left\| \mathbf{U}_t^\theta\left(X_t^-\right) - \mathbf{U}_s^\theta\left(X_s^-\right) - \mathrm{sg}\{\log q_{t|s}(X_t^-|X_s^-) - \log q_{s|t}^\theta(X_s^-|X_t^-)\} \right\|_2^2 \Bigg],
\end{aligned}
$$

$$
\text{with } \ X_s = S(s)X_0 + r(s)Z \ , \ X_t = [S(t)/S(s)]X_s + r(t)\{1 - \sigma^2(s)/\sigma^2(t)\}^{1/2}\tilde{Z} \ ,
$$

$$
\text{and } \ X_s^- = S(s)X_0 - r(s)Z \ , \ X_t^- = [S(t)/S(s)]X_s^- + r(t)\{1 - \sigma^2(s)/\sigma^2(t)\}^{1/2}\tilde{Z} \ .
$$

Contrary to the LFPE objective, the obtained loss function does not require backpropagation through the time derivative $\partial_t \mathbf{U}^\theta$, which represents a significant computational advantage. However, $\mathcal{L}_{\mathrm{RNE}}$ suffers from a severe bias-variance tradeoff with respect to the time gap $\delta = t - s$ for selected times $s$ and $t$: if $\delta$ is too large, then the denoising approximation obtained via $q_{s|t}^\theta$ may be ineffective and bring much bias; on the other hand, if $\delta$ is too small, the resulting objective may be prone to high variance. While He et al. (2026) propose to use the Euler-Maruyama estimation for $q_{s|t}^\theta$, see Lemma 9, we rather consider the Exponential Integration, see Appendix B.2 and Appendix B.3 for the formulas, which provides better accuracy for larger gap $\delta$. In our experiments, we will systematically refer to this version of $\mathcal{L}_{\mathrm{RNE}}^{\mathrm{anti}}$ as the "RNE objective".

**Diffusive Classification (DiffCLF) (OuYang et al., 2026).** Rather than relying on differential constraints, an alternative strategy for learning the log-densities $(\log p_t)_{t \in [0,T]}$ consists in enforcing self-consistency through a classification objective across noise levels. Given a collection of times $\{t_i\}_{i=1}^N$ discretizing the interval $[0, T]$, the underlying idea is to treat a sample $y$ associated with a label $c = i$ as being drawn from the marginal distribution $p_{t_i}$, and to model the resulting class-conditional probabilities via a parametric energy-based family $p_t^\theta(y) \propto \exp(-\mathbf{U}_t^\theta(y))/\mathcal{Z}_t(\theta)$, where the log-normalizing constant is learned as an additional time-dependent scalar parameter (in practice implemented as a bias on the last layer of $\mathbf{U}^\theta$). Under the uniform prior $p(c = i) = 1/N$, the posterior probabilities derived from Bayes' rule are $p^\theta(c = i|y) = p_{t_i}^\theta(y)/\sum_{j=1}^N p_{t_j}^\theta(y)$, and the associated categorical cross-entropy gives rise to the following objective for estimating the log-density $\log p_t$

$$
\mathcal{L}_{\mathrm{DiffCLF}}(\theta) = -\mathrm{E}\left[\frac{1}{N} \sum_{i=1}^N \log \frac{p_{t_i}^\theta(X_{t_i})}{\sum_{j=1}^N p_{t_j}^\theta(X_{t_i})}\right] \ , \ X_{t_i} = S(t_i)X_0 + r(t_i)Z_i \ ,
$$

where $(t_1, \ldots, t_N) \sim \mathrm{U}([0, T])^N$, $X_0 \sim \pi$ and $(Z_1, \ldots, Z_N) \sim \mathrm{N}(0, \mathrm{I}_d)^{\otimes N}$. A key feature of this objective is that, unlike score-based losses or their time-derivative counterparts, it directly probes log-density values across noise levels and thus circumvents the mode blindness pathology inherent to gradient-only formulations (Wenliang & Kanagawa, 2021; Zhang et al., 2022): distributions sharing identical modes but differing mixture weights yield distinguishable classification posteriors. As shown in OuYang et al. (2026), while the true marginals $(p_t)_{t \in [0,T]}$ are a minimizer of $\mathcal{L}_{\mathrm{DiffCLF}}$, uniqueness only holds up to a positive multiplicative factor; combining this objective with the DSM loss restores identifiability and yields a consistent estimator of the log-densities. In the binary case $N = 2$, it can be further shown that $\mathcal{L}_{\mathrm{DiffCLF}}$ recovers the tSM objective in the continuous-time limit, which provides a natural bridge with time-score-matching approaches. Computationally, this objective only requires $N$ evaluations of $\mathbf{U}^\theta$ per sampled time and bypasses any backpropagation through higher-order derivatives, making it significantly cheaper than LFPE-based formulations.

---

[10]While He et al. (2026) propose to replace $q_{t|s}$, though tractable, by its Euler-Maruyama estimation, our implementation relies rather on its exact formulation to avoid bringing additional approximation error into the loss.

# B  Details on (de)noising diffusion processes

In this section, we consider a target probability distribution $\pi \in \mathcal{P}(\mathbb{R}^d)$ and a pair of time points $(s,t)$ satisfying $T \geq t > s \geq 0$. We present technical derivations related to the integration of (de)noising diffusion processes under a unified framework, covering the generic setting (Appendix B.1), the Variance-Preserving scheme (Appendix B.2), and the Variance-Exploding scheme (Appendix B.3). Throughout, the notation $\mathbf{s}_t(x)$ and $\mathbf{H}_t(x)$ denotes, respectively, exact or approximate evaluation of the score $\nabla \log p_t(x)$ and the Hessian $\nabla^2 \log p_t(x)$. This unified formulation allows our computations to encompass both idealized and practical regimes considered in this paper.

For diffusion-based deterministic maps obtained by integrating the noising or denoising ODE with step size $\delta > 0$, we use the standard numerical-analysis terminology: an integrator is called "1st order" if its integration error is $o(\delta)$, and "2nd order" if it is $o(\delta^2)$. This convention is unrelated to the terminology used in the main paper for Gaussian denoising kernels, where "1st order" refers to a mean-only parameterization, while "2nd order" refers to an additional covariance parameterization.

## B.1  General noising scheme

Here, we consider the most general form of SDE (2), where $f$ and $g$ both verify Assumption 2, and provide below the related results of ODE and SDE integration, respectively obtained via Euler and EM schemes.

**Lemma 8** (Exact noising SDE integration - General case). *The conditional distribution of $X_t$ given $X_s = x_s \in \mathbb{R}^d$ is defined by the Gaussian kernel*

$$q_{t|s}(\cdot|x_s) = \mathrm{N}\left(\alpha_{t|s}x_s, \sigma_{t|s}^2 \mathrm{I}_d\right) \ , \ \ with \ \alpha_{t|s} = S(t)/S(s) \ and \ \sigma_{t|s}^2 = S(t)^2\{\sigma^2(t) - \sigma^2(s)\} \ ,$$

*where $S(t) = \exp(\int_0^t f(u)\mathrm{d}u)$ and $\sigma^2(t) = \int_0^t g^2(u)/S(u)^2\mathrm{d}u$.*

*Proof.* This is an immediate corollary of Lemma 6. $\qquad\square$

**Lemma 9** (Approximate denoising SDE integration - General case). *Denote $\delta = t - s$. Then, the conditional distribution of $X_t$ given $X_s = x_s \in \mathbb{R}^d$ may be approximated by the Gaussian kernel*

$$q_{s|t}(\cdot|x_t) = \mathrm{N}\left((1 - f(t)\delta)x_t + g^2(t)\delta\,\mathbf{s}_t(x_t), g^2(t)\delta\,\mathrm{I}_d\right) \ ,$$

*Proof.* This result is a straightforward application of the Euler-Maruyama scheme applied to SDE (5). $\quad\square$

**Lemma 10** (Approximate noising ODE integration - General case). *Denote $\delta = t - s$. Then, the solution at time $t$ of the forward probability flow ODE (6) starting from $x_s \in \mathbb{R}^d$ at time $s$ may be approximated in two ways:*

$$\tilde{\mathrm{T}}_{t|s}(x_s) = x_s + \delta v(s, x_s) \qquad\qquad \textit{(Euler method: explicit, 1st order)}$$

$$\mathrm{T}_{t|s}(x_s) = x_s + \delta v\left(\frac{s+t}{2}, \frac{x_s + \mathrm{T}_{t|s}(x_s)}{2}\right) \qquad \textit{(Midpoint method : implicit, 2nd order)}$$

$$where \ v(u, x) = f(u)x - \frac{g^2(u)}{2}\mathbf{s}_u(x) \ .$$

**Lemma 11** (Approximate denoising ODE integration - General case). *Denote $\delta = t - s$. Then, the solution at time $s$ of the backward probability flow ODE (6) starting from $x_t \in \mathbb{R}^d$ at time $t$ may be approximated in two ways:*

$$\tilde{\mathrm{T}}_{s|t}(x_t) = x_t - \delta v(t, x_t) \qquad\qquad \textit{(Euler method : explicit, 1st order)}$$

$$\mathrm{T}_{s|t}(x_t) = x_t - \delta v\left(\frac{s+t}{2}, \frac{\mathrm{T}_{s|t}(x_t) + x_t}{2}\right) \qquad \textit{(Midpoint method: implicit, 2nd order)}$$

$$where \ v(u, x) = f(u)x - \frac{g^2(u)}{2}\mathbf{s}_u(x) \ .$$

**Remark on the mutual invertibility of the ODE integrators.** It is easy to verify that the noising and denoising implicit Midpoint integrators described above are mutual inversible maps, *i.e.*, we have $T_{s|t} \circ T_{t|s} = T_{t|s} \circ T_{s|t} = \text{Id}$. However, this is not the case for the Euler maps $\tilde{T}_{s|t}$ and $\tilde{T}_{t|s}$.

**Lemma 12** (Formula for the Jacobian of the Midpoint integrators). *Let $\delta > 0$, and let define the numerical constants $c_1(\delta)$, $c_2(\delta)$ and $c_3(\delta)$ as*

$$c_1(\delta) = \frac{1 + (\delta/2)f((s+t)/2)}{1 - (\delta/2)f((s+t)/2)} \ , \ c_2(\delta) = \frac{\delta}{4}\frac{g^2\left(\frac{s+t}{2}\right)}{1 - \frac{\delta}{2}f\left(\frac{s+t}{2}\right)} \ , \ c_3(\delta) = \frac{\delta}{4}\frac{g^2\left(\frac{s+t}{2}\right)}{1 + \frac{\delta}{2}f\left(\frac{s+t}{2}\right)} \ .$$

*Consider the same notation as in Lemma 10 and Lemma 11. Assume that there exists $L > 0$ such that $\mathbf{s}_{(s+t)/2}$ is L-Lipschitz. Then for any positive step-size $\delta = t - s$ such that $\max\left(|c_2(\delta)|, |c_3(\delta)|\right) < 1/L$, the Jacobians of Midpoint integration maps $T_{t|s}$ and $T_{s|t}$, respectively denoted by $J_{t|s}$ and $J_{s|t}$, verify for any inputs $x_s \in \mathbb{R}^d$ and $x_t \in \mathbb{R}^d$*

$$J_{t|s}(x_s) = c_1(\delta)\left(I_d + c_2(\delta)A(x_s)\right)^{-1}\left(I_d - c_3(\delta)A(x_s)\right) \ ,$$

$$J_{s|t}(x_t) = c_1(\delta)^{-1}\left(I_d - c_3(\delta)B(x_t)\right)^{-1}\left(I_d + c_2(\delta)B(x_t)\right) \ ,$$

*where $A(x_s) = \mathbf{H}_{(s+t)/2}\left(\frac{x_s + T_{t|s}(x_s)}{2}\right)$ and $B(x_t) = \mathbf{H}_{(s+t)/2}\left(\frac{x_t + T_{s|t}(x_t)}{2}\right)$.*

*Proof.* The result from Lemma 12 follows from the factorization of the following identities, inherited from the implicit expressions of $T_{t|s}$ and $T_{s|t}$,

$$J_{t|s}(x_s) = \left(\left(1 - \frac{\delta}{2}f\left(\frac{s+t}{2}\right)\right)I_d + \frac{\delta}{4}g^2\left(\frac{s+t}{2}\right)A(x_s)\right)^{-1}\left(\left(1 + \frac{\delta}{2}f\left(\frac{s+t}{2}\right)\right)I_d - \frac{\delta}{4}g^2\left(\frac{s+t}{2}\right)A(x_s)\right) \ ,$$

$$J_{s|t}(x_t) = \left(\left(1 + \frac{\delta}{2}f\left(\frac{s+t}{2}\right)\right)I_d - \frac{\delta}{4}g^2\left(\frac{s+t}{2}\right)B(x_t)\right)^{-1}\left(\left(1 - \frac{\delta}{2}f\left(\frac{s+t}{2}\right)\right)I_d + \frac{\delta}{4}g^2\left(\frac{s+t}{2}\right)B(x_t)\right) \ .$$

Here, the assumption on $\delta$ guarantees the invertibility of the matrices $I_d + c_2(\delta)A(x_s)$ and $I_d - c_3(\delta)B(x_t)$. $\square$

**Proposition 13** (Exact expression of the Jacobian log-determinants of the Midpoint integrators via power series). *Consider the same notation as in Lemma 12. Assume that there exists $L > 0$ such that $\mathbf{s}_{(s+t)/2}$ is L-Lipschitz. Then, for any positive step-size $\delta = t - s$ such that $\max\left(|c_2(\delta)|, |c_3(\delta)|\right) < 1/L$, for any inputs $x_s \in \mathbb{R}^d$ and $x_t \in \mathbb{R}^d$, we have*

$$\log\left|\det J_{t|s}(x_s)\right| = \sum_{i=0}^{\infty} a_i(s,t)\text{Tr}([A(x_s)]^i) \ ,$$

$$\log\left|\det J_{s|t}(x_t)\right| = \sum_{i=0}^{\infty} b_i(s,t)\text{Tr}([B(x_t)]^i) \ ,$$

*where $\{a_i(s,t), b_i(s,t)\}_{i=0}^{\infty}$ are numerical coefficients defined by*

$$a_0(s,t) = -b_0(s,t) = d\log\left[\left|\frac{1 + (\delta/2)f((s+t)/2)}{1 - (\delta/2)f((s+t)/2)}\right|\right] \ ,$$

$$a_i(s,t) = \frac{\delta^i}{4^i}g^{2i}\left(\frac{s+t}{2}\right)\frac{(-1)^i\left(1 + \frac{\delta}{2}f\left(\frac{s+t}{2}\right)\right)^i - \left(1 - \frac{\delta}{2}f\left(\frac{s+t}{2}\right)\right)^i}{i\left(1 - \frac{\delta^2}{4}f^2\left(\frac{s+t}{2}\right)\right)^i} \ \text{for any } i \geq 1 \ ,$$

$$b_i(s,t) = \frac{\delta^i}{4^i}g^{2i}\left(\frac{s+t}{2}\right)\frac{\left(1 - \frac{\delta}{2}f\left(\frac{s+t}{2}\right)\right)^i - (-1)^i\left(1 + \frac{\delta}{2}f\left(\frac{s+t}{2}\right)\right)^i}{i\left(1 - \frac{\delta^2}{4}f^2\left(\frac{s+t}{2}\right)\right)^i} \ \text{for any } i \geq 1 \ .$$

*Proof.* Consider the Jacobian matrices $J_{t|s}(x_s)$ and $J_{s|t}(x_t)$ introduced in Lemma 12. Note that we have $\|A(x_s)\| < \min\left(1/|c_2(\delta)|, 1/|c_3(\delta)|\right)$ and $\|B(x_t)\| < \min\left(1/|c_2(\delta)|, 1/|c_3(\delta)|\right)$ based on the assumptions on $\mathbf{s}_{(s+t)/2}$ and $\delta$. This allows us to apply Corollary 5 on $J_{t|s}(x_s)$ and $J_{s|t}(x_t)$, respectively with $M = A(x_s)$ and $M = B(x_t)$, to obtain their expansion series in a straightforward manner. $\square$

**Remark on the $\delta$-assumption in Lemma 12 and Proposition 13.** For any general noise schedule defined by coefficients $f$ and $g$, the assumption $\max\left(|c_2(\delta)|,|c_3(\delta)|\right) < 1/L$ can be rephrased into $\delta = O(1/L)$, by considering limit approximations of coefficients $c_2(\delta)$ and $c_3(\delta)$ in the asymptotic regime $\delta \to 0$. Below, we present a rigorous expression of this upper bound on $\delta$ for the noising schemes considered in this paper, that is the *Variance-Preserving* approach (see Appendix B.2) and the *Variance-Exploding* approach (see Appendix B.3).

## B.2 Variance-Preserving diffusion

Consider the noising SDE (2) where $f(t) = -g^2(t)/2$ and $g$ being such that $\int_0^T g^2(s)\mathrm{d}s \gg 1$, with arbitrary volatility coefficient $\sigma > 0$,

$$\mathrm{d}X_t = -\frac{g^2(t)X_t}{2}\mathrm{d}t + \sigma g(t)\mathrm{d}W_t \ , \ X_0 \sim \pi \ . \tag{27}$$

This noising scheme, known as the *Variance-Preserving* (VP) scheme (Song et al., 2021), is largely used in score-based generative models. In the following, we denote $\alpha_t = \int_0^t g^2(t)\mathrm{d}t$ for any $t \in [0,T]$. Below, we derive the related results of VP-based ODE and SDE integration, obtained by using the EI scheme.

**On the choice of the $g$-schedule.** Previous works have considered a linear schedule $g^2(t) = \beta_{\min}(1 - t/T) + \beta_{\max}(t/T)$ where $\beta_{\min} = 0.1$, $\beta_{\max} \in \{10, 20\}$ and $T = 1$, see e.g., Song et al. (2021) or cosine parameterization (Nichol & Dhariwal, 2021), which has been proved to perform better in generative modeling. In our sampling experiments, we did not observe any significant difference between these two settings. Hence, we fix the linear schedule to be the default setting for our numerics, and let $\sigma$ be arbitrarily chosen.

**Lemma 14** (Exact noising SDE integration - VP case). *The conditional distribution of $X_t$ given $X_s = x_s \in \mathbb{R}^d$ is defined by the Gaussian kernel*

$$q_{t|s}(\cdot|x_s) = \mathrm{N}\left(\alpha_{t|s}x_s, \sigma_{t|s}^2 \mathrm{I}_d\right) \ , \ with \ \alpha_{t|s} = \sqrt{1 - \lambda_{s,t}^f} \ and \ \sigma_{t|s}^2 = \sigma^2 \lambda_{s,t}^f \ ,$$

*where $\lambda_{s,t}^f = 1 - \exp(\alpha_s - \alpha_t)$. Since $p_T(x) = \int_{\mathbb{R}^d} p_{T|0}(x|x_0)\mathrm{d}\pi(x_0)$, it results that $p_T \approx \mathrm{N}(0, \sigma^2 \mathrm{I}_d)$.*

*Proof.* Lemma 6 applied on noising SDE (27). $\qquad\square$

Based on the previous lemma, the interpolation coefficients in Equation (3) are given by

$$S(t) = \exp(-\alpha_t/2) \ and \ \sigma(t) = \sigma\sqrt{1 - \exp(-\alpha_t)}.$$

In particular, $t \mapsto \sigma(t)$ is not explicitly invertible. Following Lemma 14, the VP scheme is an 'ergodic' noising scheme, converging exponentially fast to the Gaussian distribution $\mathrm{N}(0, \sigma^2 \mathrm{I}_d)$; therefore, we have $\pi^{\mathrm{base}} = \mathrm{N}(0, \sigma^2 \mathrm{I}_d)$ in this setting. Moreover, under mild assumptions on $\pi$, the denoising SDE (5) writes as

$$\mathrm{d}X_t = -\frac{g^2(t)}{2}\{X_t + 2\sigma^2 \nabla \log p_t(X_t)\}\mathrm{d}t + \sigma g(t)\mathrm{d}\tilde{B}_t, \ X_T \sim \pi^{\mathrm{base}} \ . \tag{28}$$

To integrate this SDE (or the equivalent probability flow ODE), one could turn to the formulas introduced in Appendix B.2, by replacing general coefficients with VP coefficients. Instead, we propose to rely on Exponential Integration (EI) formulas dispensed in Lemma 6 (SDE case) and Lemma 7 (ODE case), that make exact the integration of the linear part of the drift.

**Lemma 15** (Approximate denoising SDE EI-based integration - VP case). *The conditional distribution of $X_t$ given $X_s = x_s \in \mathbb{R}^d$ may be approximated by the Gaussian kernel*

$$q_{s|t}(\cdot|x_t) = \mathrm{N}\left(\sqrt{1 + \lambda_{s,t}^b}x_t + 2\sigma^2\left\{\sqrt{1 + \lambda_{s,t}^b} - 1\right\}\mathbf{s}_t\left(x_t\right), \sigma^2\lambda_{s,t}^b \mathrm{I}_d\right) \ ,$$

*with $\lambda_{s,t}^b = \exp(\alpha_t - \alpha_s) - 1$.*

*Proof.* Lemma 6 applied on denoising SDE (28). □

**Lemma 16** (Approximate noising ODE EI-based integration - VP case). *The solution at time $t$ of the forward probability flow ODE (6) starting from $x_s \in \mathbb{R}^d$ at time $s$ may be approximated in two ways:*

$$\tilde{\mathrm{T}}_{t|s}(x_s) = \sqrt{1 - \lambda_{s,t}^f}\, x_s + \sigma^2 \left\{ \sqrt{1 - \lambda_{s,t}^f} - 1 \right\} \mathbf{s}_s(x_s) \qquad \text{(Euler method : explicit)}$$

$$\mathrm{T}_{t|s}(x_s) = \sqrt{1 - \lambda_{s,t}^f}\, x_s + \sigma^2 \left\{ \sqrt{1 - \lambda_{s,t}^f} - 1 \right\} \mathbf{s}_{(s+t)/2} \left( \frac{x_s + \mathrm{T}_{t|s}(x_s)}{2} \right) \quad \text{(Midpoint method : implicit)}$$

*Proof.* Lemma 7 applied on forward time ODE (6). □

**Lemma 17** (Approximate denoising ODE EI-based integration - VP case). *The solution at time $s$ of the probability flow ODE (6) starting from $x_t \in \mathbb{R}^d$ at time $t$ may be approximated in two ways:*

$$\tilde{\mathrm{T}}_{s|t}(x_t) = \sqrt{1 + \lambda_{s,t}^b}\, x_t + \sigma^2 \left\{ \sqrt{1 + \lambda_{s,t}^b} - 1 \right\} \mathbf{s}_t(x_t) \qquad \text{(Euler method: explicit)}$$

$$\mathrm{T}_{s|t}(x_t) = \sqrt{1 + \lambda_{s,t}^b}\, x_t + \sigma^2 \left\{ \sqrt{1 + \lambda_{s,t}^b} - 1 \right\} \mathbf{s}_{(s+t)/2} \left( \frac{\mathrm{T}_{s|t}(x_t) + x_t}{2} \right) \quad \text{(Midpoint : implicit)}$$

*Proof.* Lemma 7 applied on backward time ODE (6). □

**Remark on the mutual invertibility of the ODE integrators.** The noising and denoising implicit Midpoint integrators described above are mutual inversible maps, *i.e.*, $\mathrm{T}_{s|t} \circ \mathrm{T}_{t|s} = \mathrm{T}_{t|s} \circ \mathrm{T}_{s|t} = \mathrm{Id}$. This is due to the identity $(1 + \lambda_{s,t}^b)^{-1} = 1 - \lambda_{s,t}^f$. This is not the case for the Euler maps $\tilde{\mathrm{T}}_{s|t}$ and $\tilde{\mathrm{T}}_{t|s}$.

**Simplification of $\delta$-assumption in Lemma 12 and Proposition 13.** Following the notation introduced in Lemma 12, we obtain simplifications of $c_2(\delta)$ and $c_3(\delta)$ in the specific VP case, for any positive step-size $\delta$, that are given by

$$c_2(\delta) = \sigma^2 \left( \frac{4}{\delta g^2 \left( (s+t)/2 \right)} + 1 \right)^{-1} , \quad c_3(\delta) = \sigma^2 \left( \frac{4}{\delta g^2 \left( (s+t)/2 \right)} - 1 \right)^{-1} .$$

Hence, for any given $L > 0$, if we have $\delta < 4/\{(\sigma^2 L + 1)g^2\left( (s+t)/2 \right)\}$, then it comes that $\max\left( |c_2(\delta)|, |c_3(\delta)| \right) < 1/L$. In particular, we may use this upper bound on $\delta$ as a more readable $\delta$-assumption in Lemma 12 and Proposition 13.

**Lemma 18** (Formula for the Jacobian of the Midpoint integrators - VP case). *Let $\delta > 0$, and let define the numerical constants $c_1(\delta)$, $c_2(\delta)$ and $c_3(\delta)$ as*

$$c_1(\delta) = \sqrt{1 - \lambda_{s,t}^f} , \; c_2(\delta) = \frac{\sigma^2}{2} \left\{ 1 - \exp\left( \frac{\alpha_s - \alpha_t}{2} \right) \right\} , \; c_3(\delta) = \frac{\sigma^2}{2} \left\{ \exp\left( \frac{\alpha_t - \alpha_s}{2} \right) - 1 \right\} .$$

*Consider the same notation as in Lemma 16 and Lemma 17. Assume that there exists $L > 0$ such that $\mathbf{s}_{(s+t)/2}$ is $L$-Lipschitz. If we further assume that $(\alpha_t - \alpha_s) < 2\log\left( 1 + 2/(L\sigma^2) \right)$, then the Jacobians of Midpoint integration maps $\mathrm{T}_{t|s}$ and $\mathrm{T}_{s|t}$, respectively denoted by $J_{t|s}$ and $J_{s|t}$, verify for any inputs $x_s \in \mathbb{R}^d$ and $x_t \in \mathbb{R}^d$*

$$J_{t|s}(x_s) = c_1(\delta) \left( \mathrm{I}_d + c_2(\delta) A(x_s) \right)^{-1} \left( \mathrm{I}_d - c_3(\delta) A(x_s) \right) ,$$

$$J_{s|t}(x_t) = c_1(\delta)^{-1} \left( \mathrm{I}_d - c_3(\delta) B(x_t) \right)^{-1} \left( \mathrm{I}_d + c_2(\delta) B(x_t) \right) ,$$

*where $A(x_s) = \mathbf{H}_{(s+t)/2}\left( \frac{x_s + \mathrm{T}_{t|s}(x_s)}{2} \right)$ and $B(x_t) = \mathbf{H}_{(s+t)/2}\left( \frac{x_t + \mathrm{T}_{s|t}(x_t)}{2} \right)$.*

*Proof.* The result from Lemma 18 follows from the factorization of the following identities, inherited from the implicit expressions of $T_{t|s}$ and $T_{s|t}$,

$$J_{t|s}(x_s) = \left(I_d - \frac{\sigma^2}{2}\left\{\sqrt{1 - \lambda_{s,t}^f} - 1\right\} A(x_s)\right)^{-1} \left(\sqrt{1 - \lambda_{s,t}^f}I_d + \frac{\sigma^2}{2}\left\{\sqrt{1 - \lambda_{s,t}^f} - 1\right\} A(x_s)\right) ,$$

$$J_{s|t}(x_t) = \left(I_d - \frac{\sigma^2}{2}\left\{\sqrt{1 + \lambda_{s,t}^b} - 1\right\} B(x_t)\right)^{-1} \left(\sqrt{1 + \lambda_{s,t}^b}I_d + \frac{\sigma^2}{2}\left\{\sqrt{1 + \lambda_{s,t}^b} - 1\right\} B(x_t)\right) .$$

Here, the additional assumption on the term $(\alpha_t - \alpha_s)$ may be seen as the EI-based analog to the assumption on the step size $\delta = t - s$ in Lemma 12. Indeed, if we have $(\alpha_t - \alpha_s) < 2\log\left(1 + 2/(L\sigma^2)\right)$, then it comes that $\max(|c_2(\delta)|, |c_3(\delta)|) < 1/L$, which thus guarantees the invertibility of the matrices $I_d + c_2(\delta)A(x_s)$ and $I_d - c_3(\delta)B(x_t)$. □

**Proposition 19** (Exact expression of the Jacobian log-determinants of the Midpoint integrators via power series - VP case)**.** *Consider the same notation as in Lemma 18. Assume that there exists $L > 0$ such that $\mathbf{s}_{(s+t)/2}$ is L-Lipschitz. If we further assume that $(\alpha_t - \alpha_s) < 2\log\left(1 + 2/(L\sigma^2)\right)$, then, for any inputs $x_s \in \mathbb{R}^d$ and $x_t \in \mathbb{R}^d$, we have*

$$\log\left|\det J_{t|s}(x_s)\right| = \sum_{i=0}^{\infty} a_i(s,t)\operatorname{Tr}([A(x_s)]^i) ,$$

$$\log\left|\det J_{s|t}(x_t)\right| = \sum_{i=0}^{\infty} b_i(s,t)\operatorname{Tr}([B(x_t)]^i) ,$$

*where $\{a_i(s,t), b_i(s,t)\}_{i=0}^{\infty}$ are numerical coefficients defined by*

$$a_0(s,t) = -b_0(s,t) = \frac{d}{2}(\alpha_s - \alpha_t) ,$$

$$a_i(s,t) = \frac{\sigma^{2i}}{2^i i}\left(\exp\left(\frac{\alpha_s - \alpha_t}{2}\right) - 1\right)^i \left(1 - (-1)^i \exp\left(-i\frac{\alpha_s - \alpha_t}{2}\right)\right) \text{ for any } i \geq 1 ,$$

$$b_i(s,t) = \frac{\sigma^{2i}}{2^i i}\left(\exp\left(-\frac{\alpha_s - \alpha_t}{2}\right) - 1\right)^i \left(1 - (-1)^i \exp\left(i\frac{\alpha_s - \alpha_t}{2}\right)\right) \text{ for any } i \geq 1 .$$

*Proof.* Similarly to the proof of Proposition 13, we combine the results of Lemma 18 and Corollary 5 to get the final result. Intermediary simplifications of the terms are omitted here to help the reading. □

### B.3  Variance-Exploding diffusion

Consider the case where $f(t) = 0$. Then, SDE (2) simply writes as

$$dX_t = g(t)dW_t, \ X_0 \sim \pi . \tag{29}$$

This noising scheme is known as the *Variance-Exploding* (VE) scheme (Song et al., 2021). Below, we derive the related results of VE-based ODE and SDE integration, obtained by using the EI scheme.

**On the choice of the $g$-schedule.**  Following the guidelines from (Karras et al., 2022), we consider the geometric schedule

$$g^2(t) = \sigma_{\min}^2 \left(\frac{\sigma_{\max}^2}{\sigma_{\min}^2}\right)^t \log\left(\frac{\sigma_{\max}^2}{\sigma_{\min}^2}\right) ,$$

where $\sigma_{\min} \approx 0$ and $\sigma_{\max} \gg 1$ can be arbitrarily chosen.

**Lemma 20** (Exact noising SDE integration - VE case). *The conditional distribution of $X_t$ given $X_s = x_s \in \mathbb{R}^d$ is defined by the Gaussian kernel*

$$q_{t|s}(\cdot|x_s) = \mathrm{N}\left(\alpha_{t|s}x_s, \sigma_{t|s}^2\,\mathrm{I}_d\right) \ , \ \textit{with } \alpha_{t|s} = 1 \textit{ and } \sigma_{t|s}^2 = \lambda_{s,t} = \sigma_{min}^2 \left(\frac{\sigma_{max}}{\sigma_{min}}\right)^{2s}\left(\left(\frac{\sigma_{max}}{\sigma_{min}}\right)^{2(t-s)} - 1\right) \ .$$

*Since $p_T(x) = \int_{\mathbb{R}^d} q_{T|0}(x|x_0)\mathrm{d}\pi(x_0)$, it results that $p_T \approx \mathrm{N}(0, \sigma_{max}^2\mathrm{I}_d)$.*

*Proof.* Lemma 6 applied on noising SDE (29). $\qquad\square$

Based on the previous lemma, the interpolation coefficients in (3) are given by

$$S(t) = 1 \text{ and } \sigma(t) = \sigma_{\min}\sqrt{\left(\frac{\sigma_{\max}}{\sigma_{\min}}\right)^{2t} - 1}.$$

In particular, $t \mapsto \sigma(t)$ is explicitly invertible, since we have

$$\sigma^{-1}(\sigma) = \frac{\log\left(\left(\frac{\sigma}{\sigma_{\min}}\right)^2 + 1\right)}{2\log\left(\frac{\sigma_{\max}}{\sigma_{\min}}\right)} \ .$$

Under mild assumptions on $\pi$, the denoising SDE (5) writes as

$$\mathrm{d}X_t = -g^2(t)\nabla\log p_t(X_t)\mathrm{d}t + g^2(t)\mathrm{d}\tilde{B}_t, \ X_T \sim \pi^{\mathrm{base}} \ . \tag{30}$$

Similarly to the VP case (see Appendix B.2), we present below approximate transition kernels and maps based on the Exponential Integration (EI). Since the linear drift term is 0 here, the EI strategy amounts to exactly integrate the time-dependent coefficient associated to the (unknown) score drift term.

**Lemma 21** (Approximate denoising SDE EI-based integration - VE case). *The conditional distribution of $X_t$ given $X_s = x_s \in \mathbb{R}^d$ may be approximated by the Gaussian kernel*

$$q_{s|t}(\cdot|x_t) = \mathrm{N}\left(x_t + \lambda_{s,t}\mathbf{s}_t\left(x_t\right), \lambda_{s,t}\,\mathrm{I}_d\right) \ .$$

*Proof.* Lemma 6 applied on denoising SDE (30). $\qquad\square$

**Lemma 22** (Approximate noising ODE EI-based integration - VE case). *The solution at time $t$ of the forward probability flow ODE (6) starting from $x_s \in \mathbb{R}^d$ at time $s$ may be approximated in two ways:*

$$\tilde{\mathrm{T}}_{t|s}(x_s) = x_s - \frac{\lambda_{s,t}}{2}\mathbf{s}_s\left(x_s\right) \qquad\qquad \textit{(Euler method : explicit)}$$

$$\mathrm{T}_{t|s}(x_s) = x_s - \frac{\lambda_{s,t}}{2}\mathbf{s}_{(s+t)/2}\left(\frac{x_s + \mathrm{T}_{t|s}(x_s)}{2}\right) \qquad\qquad \textit{(Midpoint method : implicit)}$$

*Proof.* Lemma 7 applied on forward time ODE (6). $\qquad\square$

**Lemma 23** (Approximate denoising ODE EI-based integration - VE case). *The solution at time $s$ of the backward probability flow ODE (6) starting from $x_t \in \mathbb{R}^d$ at time $t$ may be approximated in two ways:*

$$\tilde{\mathrm{T}}_{s|t}(x_t) = x_t + \frac{\lambda_{s,t}}{2}\mathbf{s}_t\left(x_t\right) \qquad\qquad \textit{(Euler method : explicit)}$$

$$\mathrm{T}_{s|t}(x_t) = x_t + \frac{\lambda_{s,t}}{2}\mathbf{s}_{(s+t)/2}\left(\frac{\mathrm{T}_{s|t}(x_t) + x_t}{2}\right) \qquad\qquad \textit{(Midpoint method : implicit)}$$

*Proof.* Lemma 7 applied on backward time ODE (6). $\qquad\square$

**Remark on the mutual invertibility of the ODE integrators.** The noising and denoising implicit Midpoint integrators described above are mutual inversible maps, *i.e.*, $T_{s|t} \circ T_{t|s} = T_{t|s} \circ T_{s|t} = \text{Id}$. This is not the case for the Euler maps $\tilde{T}_{s|t}$ and $\tilde{T}_{t|s}$.

**Simplification of $\delta$-assumption in Lemma 12 and Proposition 13.** Following the notation introduced in Lemma 12, we obtain simplifications of $c_2(\delta)$ and $c_3(\delta)$ in the specific VE case, for any positive step-size $\delta$, that are given by

$$c_2(\delta) = c_3(\delta) = \frac{\delta}{4} g^2 \left( \frac{s+t}{2} \right) .$$

Hence, for any given $L > 0$, if we have $\delta < 4/\{Lg^2((s+t)/2)\}$, then it comes that $\max(|c_2(\delta)|, |c_3(\delta)|) < 1/L$. In particular, we may use this upper bound on $\delta$ as a more readable $\delta$-assumption in Lemma 12 and Proposition 13.

**Lemma 24** (Formula for the Jacobian of the Midpoint integrators - VE case). *Let $\delta > 0$, and let define the numerical constants $c_1(\delta)$, $c_2(\delta)$ and $c_3(\delta)$ as*

$$c_1(\delta) = 1 , \ c_2(\delta) = c_3(\delta) = \frac{\lambda_{s,t}}{4} .$$

*Consider the same notation as in Lemma 22 and Lemma 23. Assume that there exists $L > 0$ such that $\mathbf{s}_{(s+t)/2}$ is L-Lipschitz. If we further assume that $\lambda_{s,t} < 4/L$, then the Jacobians of Midpoint integration maps $T_{t|s}$ and $T_{s|t}$, respectively denoted by $J_{t|s}$ and $J_{s|t}$, verify for any inputs $x_s \in \mathbb{R}^d$ and $x_t \in \mathbb{R}^d$*

$$J_{t|s}(x_s) = c_1(\delta) \left( I_d + c_2(\delta) A(x_s) \right)^{-1} \left( I_d - c_3(\delta) A(x_s) \right) ,$$
$$J_{s|t}(x_t) = c_1(\delta)^{-1} \left( I_d - c_3(\delta) B(x_t) \right)^{-1} \left( I_d + c_2(\delta) B(x_t) \right) ,$$

*where $A(x_s) = \mathbf{H}_{(s+t)/2} \left( \frac{x_s + T_{t|s}(x_s)}{2} \right)$ and $B(x_t) = \mathbf{H}_{(s+t)/2} \left( \frac{x_t + T_{s|t}(x_t)}{2} \right)$.*

*Proof.* The result from Lemma 24 follows from the factorization of the following identities, inherited from the implicit expressions of $T_{t|s}$ and $T_{s|t}$,

$$J_{t|s}(x_s) = \left( I_d + \frac{\lambda_{s,t}}{4} A(x_s) \right)^{-1} \left( I_d - \frac{\lambda_{s,t}}{4} A(x_s) \right) ,$$
$$J_{s|t}(x_t) = \left( I_d - \frac{\lambda_{s,t}}{4} B(x_t) \right)^{-1} \left( I_d + \frac{\lambda_{s,t}}{4} B(x_t) \right) .$$

Here, the additional assumption on the term $\lambda_{s,t}$ may be seen as the EI-based analog to the assumption on the step size $\delta = t - s$ in Lemma 12. Indeed, if we have $\lambda_{s,t} < 4/L$, then it comes that $\max(|c_2(\delta)|, |c_3(\delta)|) < 1/L$, which thus guarantees the invertibility of the matrices $I_d + c_2(\delta) A(x_s)$ and $I_d - c_3(\delta) B(x_t)$. $\square$

**Proposition 25** (Exact expression of the Jacobian log-determinants of the Midpoint integrators via power series - VE case). *Consider the same notation as in Lemma 24. Assume that there exists $L > 0$ such that $\mathbf{s}_{(s+t)/2}$ is L-Lipschitz. If we further assume that $\lambda_{s,t} < 4/L$, then, for any inputs $x_s \in \mathbb{R}^d$ and $x_t \in \mathbb{R}^d$, we have*

$$\log \left| \det J_{t|s}(x_s) \right| = \sum_{i=0}^{\infty} a_i(s,t) \operatorname{Tr}([A(x_s)]^i) ,$$
$$\log \left| \det J_{s|t}(x_t) \right| = \sum_{i=0}^{\infty} b_i(s,t) \operatorname{Tr}([B(x_t)]^i) ,$$

*where $\{a_i(s,t), b_i(s,t)\}_{i=0}^{\infty}$ are numerical coefficients defined by*

$$a_i(s,t) = b_i(s,t) = 0 \text{ for any even } i \in \mathbb{N} ,$$
$$a_i(s,t) = -b_i(s,t) = -2\frac{\lambda_{s,t}^i}{4^i} \text{ for any odd } i \in \mathbb{N} .$$

*Proof.* Similarly to the proof of Proposition 13, we combine the results of Lemma 24 and Corollary 5 to get the final result. Intermediary simplifications of the terms are omitted here to help the reading. □

### B.4 Discrete time setting for diffusion models

Following Karras et al. (2024); Grenioux et al. (2024), we define the time discretization $\{t_k\}_{k=0}^K \subset [0, T]$ from a uniform grid in log-SNR space. Recalling the interpolation coefficients $S(t)$ and $\sigma(t)$ from Equation (3), the log-SNR at time $t$ is given by

$$\mathrm{logSNR}(t) = 2 \log S(t) - \log \sigma(t)^2 \ .$$

Given fixed endpoints $t_{\mathrm{start}} \approx 0$ and $t_{\mathrm{end}} \approx T$, we discretize uniformly in log-SNR between $\lambda_0 = \mathrm{logSNR}(t_{\mathrm{start}})$ and $\lambda_K = \mathrm{logSNR}(t_{\mathrm{end}})$, and recover $t_k$ by inversion:

$$\lambda_k = \lambda_0 + \frac{k}{K}(\lambda_K - \lambda_0) \ , \ t_k = \mathrm{logSNR}^{-1}(\lambda_k) \ . \tag{31}$$

We have $p_{t_K} \approx \pi^{\mathrm{base}}$ and $p_{t_0} \approx \pi$, and denote $\delta_k = t_{k+1} - t_k$.

**VP case.** Since $S(t)^2 = \exp(-\alpha_t)$ and $\sigma(t)^2 = \sigma^2\{1 - \exp(-\alpha_t)\}$ (see Appendix B.2), the inversion $t_k = \mathrm{logSNR}^{-1}(\lambda_k)$ reduces to inverting $\alpha$:

$$t_k = \alpha^{-1}\big(\log\big(1 + \sigma^2 \mathrm{e}^{\lambda_k}\big)\big) \ .$$

When $g^2(t) = \beta_{\min}(1 - t/T) + \beta_{\max}(t/T)$, $\alpha_t$ is quadratic in $t$ and $\alpha^{-1}$ is available in closed form.

**VE case.** Since $S(t) = 1$, we have $\mathrm{logSNR}(t) = -\log \sigma(t)^2$, so a uniform grid in log-SNR is equivalent to a uniform grid in $\log \sigma^2(t)$. With $\sigma^2(t) = \sigma_{\min}^2\{(\sigma_{\max}/\sigma_{\min})^{2t} - 1\}$, the inversion is explicit and yields

$$t_k = \frac{1}{2\log(\sigma_{\max}/\sigma_{\min})} \log\left(1 + (\sigma_{\max}/\sigma_{\min})^{2k/K}\right) \ .$$

Following Karras et al. (2022), we directly parameterize the grid through $\sigma$ rather than through log-SNR. Specifically, we set $\sigma_k = \sigma_{\min}(\sigma_{\max}/\sigma_{\min})^{k/K}$ for $k \in \{0, \dots, K\}$, which is a geometric progression in $\sigma$ between $\sigma_{\min}$ and $\sigma_{\max}$, and recover $t_k = \sigma^{-1}(\sigma_k)$ via the closed-form expression above. This is equivalent to the log-SNR-uniform discretization since $\mathrm{logSNR}(t) = -\log \sigma(t)^2$ in the VE case, but is more natural to specify in practice as the endpoints $\sigma_{\min}$ and $\sigma_{\max}$ have a direct interpretation as noise levels.

**Considering the $\Lambda$-optimal time discretization for diffusion models.** To ensure a fair comparison with tempering-based samplers, we also consider an alternative time discretization for diffusion models based on the global-barrier criterion $\Lambda$ of Syed et al. (2021; 2022; 2025). We recall that, for a generic annealing path $(p_t)_{t \in [0,1]}$ traversed by an MCMC transition kernel, the local communication barriers for RE and SMC are respectively given by

$$\lambda_{\mathrm{RE}}(t) = \tfrac{1}{2}\,\mathbb{E}\left[|\partial_t \log p_t(X_t) - \partial_t \log p_t(X_t')|\right], \quad \lambda_{\mathrm{SMC}}(t) = \sqrt{\mathbb{V}\left[\partial_t \log p_t(X_t)\right]},$$

with $X_t, X_t' \sim p_t$ taken independent. The cumulative barrier $\Lambda(t) = \int_0^t \lambda(u)\,\mathrm{d}u$ provides the optimality criterion: an equal-mass discretization of $\Lambda$ on $[0, 1]$ asymptotically maximizes the RE round-trip time and the SMC log-normalizing-constant variance. In contrast to Syed et al. (2021; 2022; 2025), who estimate $\Lambda$ adaptively while running the sampler, we exploit the controlled nature of our experiments to compute $\lambda(t)$ *a priori* on a fine grid of 2048 levels and invert the resulting cumulative profile to obtain the equal-mass time grid for any target value of $K$.

The barriers above can in principle be sharpened to account for the actual stochastic kernels used by the algorithm via additional terms $\partial_s \log p_{t|s}(x_t|x_s)|_{s=t}$ and $\partial_s \log q_{s|t}(x_s|x_t)|_{s=t}$ involving the forward and backward transitions. We do not pursue this for two reasons. First, the partial derivatives of the transition log-densities diverge on the diagonal $s = t$, so the stochastic-kernel barriers cannot be evaluated a priori.

Second, the deterministic-kernel barriers, which replace those terms by $\nabla \log p_t(x)^\top v(t, x) + \nabla \cdot v(t, x)$ via $\partial_s T_{s|t}|_{s=t} = v(t, \cdot)$, are well defined and numerically tractable, but empirically yield values numerically indistinguishable from their classic MCMC-kernel counterparts and therefore induce essentially the same discretization. We consequently work with the classic barriers above throughout.

The expectations and variances defining $\lambda(t)$ are estimated with $N = 2048$ Monte Carlo samples, drawn as follows.

- *Tempering path.* At each $t$, the intermediate density admits no closed-form sampler. We run a few Newton-Raphson steps on $\log p_t$ initialized at the modes of the target mixture to obtain their tempered counterparts, build a Laplace Gaussian-mixture proposal centred at those modes with covariances given by the inverse of the negative Hessian of $\log p_t$, and produce $N$ approximate samples by sampling-importance-resampling from a pool of size $5N$. The derivative $\partial_t \log p_t$ is available in closed form.

- *Perfect diffusion path.* The intermediate marginals are Gaussian mixtures whose parameters are known analytically; we sample $X_0 \sim p_0$ and apply the analytic forward transition, and evaluate $\partial_t \log p_t$ in closed form.

- *Learned diffusion path.* The learned model is energy-based, so $\log p_t^\theta$ and $\partial_t \log p_t^\theta$ are directly accessible. We draw samples from the perfect diffusion path and use them as a proposal in self-normalized importance sampling against the learned marginal $p_t^\theta$, then evaluate the barriers on the reweighted samples.

In all three cases, the $\lambda$-curve is integrated by the trapezoidal rule on the 2048-level grid to produce $\Lambda$, which is then inverted by linear interpolation to yield the $\Lambda$-optimal discretization for each target $K$.

We report the comparison between $\Lambda$-optimal and log-SNR discretizations for diffusion-based samplers in idealized setting **(A)** in Appendix D.4. Overall, the $\Lambda$-optimal discretization brings no substantial improvement and can even lead to severe performance degradation. We also ran the full ablation in the learned setting and observed the same pattern, typically more pronounced: $\Lambda$-optimal discretization degraded performance more frequently and more severely than in the perfect-density case. Since even the idealized comparison fails to favour it, we omit the learned-setting numbers and retain log-SNR as the default discretization for all DM-based aMC-BGs throughout the paper.

## C  Proofs of Section 4

In the main paper, we present our methodology to design diffusion-based aMC-BGs via deterministic transitions between noise levels, by relying on the general noising framework presented in Appendix B.1 to maintain a certain generality. We highlight that these results still hold within the specific EI-based framework of VP noising (see Appendix B.2) and VE noising (see Appendix B.3), based on the formulas introduced in the respective sections. We leave the proof for the reader.

*Proof of Proposition 1.* This is a restatement of the results of Lemma 10 and Lemma 11 in the case where $s = t_k$, $t = t_{k+1}$ and $\mathbf{s}_t = \nabla \log p_t$. The mutual invertibility property is immediate. □

We give below a formal version of Assumption 1.

**Assumption 3** (Score smoothness & discretization error - Formal version of Assumption 1)**.** *(a) There exists $L_k > 0$ such that $\nabla \log p_{t_{k+1/2}}$ is $L_k$-Lipschitz and (b) the step-size $\delta_k$ verifies*

$$\max \left( L_k \left| c_2(\delta_k) \right|, L_k \left| c_3(\delta_k) \right|, c_4(\delta_k, L_k) \right) < 1 \,,$$

*where $c_2$ and $c_3$ are given in Lemma 12, and $c_4(\delta, L) = \frac{\delta}{2} \left( \left| f(t_{k+1/2}) \right| + L g^2(t_{k+1/2})/2 \right)$.*

*Proof of Proposition 2.* Assume Assumption 3. Fix current states $x_k$ and $x_{k+1}$. Respectively, denote the sequences $\{T_{k+1|k}^{(n)}(x_k)\}_{n\in\mathbb{N}}$ and $\{T_{k|k+1}^{(n)}(x_{k+1})\}_{n\in\mathbb{N}}$ by $\{y_n\}_{n\in\mathbb{N}}$ and $\{z_n\}_{n\in\mathbb{N}}$. Respectively define the *forward* map $\Psi_{k+1|k} : \mathbb{R}^d \to \mathbb{R}^d$ and the *backward* map $\Psi_{k|k+1} : \mathbb{R}^d \to \mathbb{R}^d$ by

$$\Psi_{k+1|k}(y) = x_k + \delta_k v\left(t_{k+1/2}, \frac{x_k + y}{2}\right) \ , \ \Psi_{k|k+1}(z) = x_{k+1} - \delta_k v\left(t_{k+1/2}, \frac{z + x_{k+1}}{2}\right) \ ,$$

such that $y_{n+1} = \Psi_{k+1|k}(y_n)$ and $z_{n+1} = \Psi_{k|k+1}(z_n)$ for any $n \in \mathbb{N}$. By combining Assumption 3-(a) and $c_4(\delta_k, L_k) < 1$ from Assumption 3-(b), it is easy to see that both maps $\Psi_{k+1|k}$ and $\Psi_{k|k+1}$ are contractive Lipschitz mappings. We directly obtain the result by application of Banach fixed-point theorem. $\square$

**Lemma 26** (Formula for the Jacobian of the IM integrator). *Following the same notation as in Proposition 1, under Assumption 1, the Jacobians of $T_{k|k+1}$ and $T_{k+1|k}$ verify*

$$J_{T_{k+1|k}}(x_k) = c_1 \left(I_d + c_2 A(x_k)\right)^{-1} \left(I_d - c_3 A(x_k)\right) \ ,$$
$$J_{T_{k|k+1}}(x_{k+1}) = c_1^{-1} \left(I_d - c_3 B(x_{k+1})\right)^{-1} \left(I_d + c_2 B(x_{k+1})\right) \ ,$$

*where*

$$A(x_k) = \mathbf{H}_{t_{k+1/2}}\left(\frac{x_k + T_{k+1|k}(x_k)}{2}\right) \ , \ B(x_{k+1}) = \mathbf{H}_{t_{k+1/2}}\left(\frac{x_{k+1} + T_{k|k+1}(x_{k+1})}{2}\right) \ ,$$

$\mathbf{H}_{t_{k+1/2}}$ *is the Hessian of* $\log p_{t_{k+1/2}}$, *and* $c_1$, $c_2$, $c_3$ *are the numerical constants given in Lemma 12.*

*Proof of Lemma 26.* This is a restatement of Lemma 12 in the case where $s = t_k$, $t = t_{k+1}$, with exact score and Hessian functions ($\nabla \log p_{t_{k+1/2}}$ and $\mathbf{H}_{t_{k+1/2}}$) used for $\mathbf{s}$ and $\mathbf{H}$. In particular, the assumptions that are needed for this result are verified by Assumption 3. $\square$

Note that similar constants to those introduced in Proposition 1 are derived for VP, respectively VE, noising scheme combined with exponential integration in Lemma 18, respectively Lemma 24, under small change in the assumption.

*Proof of Proposition 3.* Assume Assumption 3. Consider a prescribed fixed-point range $M \geq 1$ satisfying (25). We first consider the approximations of $A(x_k)$ and $B(x_{k+1})$, introduced in Lemma 26, obtained by replacing the intractable implicit map evaluations with their fixed-point estimations, see (23) and (24). This leads to the following expressions

$$A^{(M)}(x_k) = \mathbf{H}_{t_{k+1/2}}\left(\frac{x_k + T_{k+1|k}^{(M)}(x_k)}{2}\right) \ , \ B^{(M)}(x_{k+1}) = \mathbf{H}_{t_{k+1/2}}\left(\frac{x_{k+1} + T_{k|k+1}^{(M)}(x_{k+1})}{2}\right) \ .$$

By Assumption 3(a)-(b) we verify that we have $\|A(x_k)\| < \min(1/|c_2(\delta_k)|, 1/|c_3(\delta_k)|)$ and $\|B(x_{k+1})\| < \min(1/|c_2(\delta_k)|, 1/|c_3(\delta_k)|)$. Therefore, we encounter the same theoretical requirements as in the proof of Proposition 13 where $s = t_k$, $t = t_{k+1}$, with exact score and Hessian functions ($\nabla \log p_{t_{k+1/2}}$ and $\mathbf{H}_{t_{k+1/2}}$) used for $\mathbf{s}$ and $\mathbf{H}$, which allows us to define the *exact* expression of the Jacobian log-determinants

$$\log\left|\det J_{T_{k+1|k}}(x_k)\right| = \sum_{i=0}^{\infty} a_i(t_k, t_{k+1}) \operatorname{Tr}([A(x_k)]^i) \ , \tag{32}$$

$$\log\left|\det J_{T_{k+1|k}}(x_k)\right| = \sum_{i=0}^{\infty} b_i(t_k, t_{k+1}) \operatorname{Tr}([B(x_{k+1})]^i) \ , \tag{33}$$

using the numerical coefficients introduced in Proposition 13. On the other hand, we also have that $\|A^{(M)}(x_k)\| < \min(1/|c_2(\delta_k)|, 1/|c_3(\delta_k)|)$ and $\|B^{(M)}(x_{k+1})\| < \min(1/|c_2(\delta_k)|, 1/|c_3(\delta_k)|)$, which allows us to *exactly* define the following expansion series based on replacing Jacobian terms $A^{(M)}(x_k)$ and $B^{(M)}(x_{k+1})$

$$\sum_{i=0}^{\infty} a_i(t_k, t_{k+1}) \operatorname{Tr}([A^{(M)}(x_k)]^i) \text{ and } \sum_{i=0}^{\infty} b_i(t_k, t_{k+1}) \operatorname{Tr}([B^{(M)}(x_{k+1})]^i) \ . \tag{34}$$

Since we expect to have $A(x_k) \approx A^{(M)}(x_k)$ and $B(x_{k+1}) \approx B^{(M)}(x_{k+1})$, we may substitute the trace terms in (32) and (33) by those in (34) to approximate $\log \left| \det J_{\mathrm{T}_{k+1|k}}(x_k) \right|$ and $\log \left| \det J_{\mathrm{T}_{k|k+1}}(x_{k+1}) \right|$. Finally, we obtain the result from Proposition 3 under additional approximation induced by the truncation of the power series at a given order $I \geq 1$, letting $a_{k,i} = a_i(t_k, t_{k+1})$ and $b_{k,i} = b_i(t_k, t_{k+1})$. $\square$

In our experiments based on the VP noising scheme combined with exponential integration, we adapt the result from Proposition 3 by using the coefficients introduced in Proposition 19. Similarly, one could use the coefficients from Proposition 25 for the VE noising scheme combined with exponential integration.

## C.1 The penalty correction

The content of this subsection is a textbook recipe in the statistical-physics literature that, to our knowledge, has rarely been used in machine learning; we recap it here as a reminder. The correction dates back to Zwanzig (1954) in the context of free-energy perturbation. We adopt the more refined take of Ceperley & Dewing (1999), which (i) sharpens the bias correction to an *exact* transform and (ii) proves its applicability to MCMC schemes. In the context of Boltzmann Generators, this recipe was leveraged only very recently in the concurrent work of Hoffmann et al. (2026).

To disambiguate notation: throughout this subsection, T denotes a generic $C^1$-diffeomorphism on $\mathbb{R}^d$, distinct the DM maps $\{\mathrm{T}_{k+1|k}, \mathrm{T}_{k|k+1}\}_k$ of the main text. The role of T also differs between the IS and MH cases below: in the IS case T may be any diffeomorphism; in the MH case T must additionally be an involution.

**Working assumption.** We assume access to an unbiased stochastic estimator $\widehat{\ell}(x)$ of $\log |\det J_{\mathrm{T}}(x)|$ with finite variance. Given $N$ i.i.d. copies, we form the empirical mean and the empirical variance of the mean

$$\widehat{L}(x) = \tfrac{1}{N} \sum_{n=1}^N \widehat{\ell}^{\,n}(x) , \quad \widehat{V}(x) = \tfrac{1}{N(N-1)} \sum_{n=1}^N \left( \widehat{\ell}^{\,n}(x) - \widehat{L}(x) \right)^2 ,$$

where $\widehat{V}(x)$ is an unbiased estimator of the variance $\nu^2(x)$ of $\widehat{L}(x)$. Following the classical derivation of Ceperley & Dewing (1999), what follows treats $\widehat{\ell}(x)$ as Gaussian; under this idealization, Cochran's theorem then ensures jointly $\widehat{L}(x) \sim \mathrm{N}(\log |\det J_{\mathrm{T}}(x)| , \nu^2(x))$, $(N-1)\widehat{V}(x)/\nu^2(x) \sim \chi^2_{N-1}$, and $\widehat{L}(x) \perp \widehat{V}(x)$.

Strictly speaking, exact Gaussianity does not hold in our setting: $\widehat{\ell}(x)$ is a quadratic form in Gaussian probes against the midpoint Hessian via the Hutchinson identity (see below), and as such follows a generalized $\chi^2$ distribution rather than a Gaussian. However, by applying the Central Limit Theorem across the $d^{\mathrm{eff}}$ eigendirections of the Hessian sensed by the probe (with $d^{\mathrm{eff}} = d$ for the standard Hutchinson estimator and typically smaller for Hutch++), this quadratic form is well-approximated by a Gaussian as soon as $d^{\mathrm{eff}}$ is moderately large. In practice, $d^{\mathrm{eff}} \geq 10$ is more than sufficient, a regime comfortably met by our multi-modal targets ($d \geq 16$) and our Hutch++ design ($d^{\mathrm{eff}} = 13$). Appendix D.4 empirically confirms that the resulting penalty estimators behave as expected.

**(a) The importance-sampling case.** Whenever $\log |\det J_{\mathrm{T}}|$ appears additively inside the log-importance weight of (21), the naive plug-in $\widehat{\ell} \leftarrow \log |\det J_{\mathrm{T}}|$ is biased by a multiplicative factor $\exp(\nu^2(x)/2)$ on the weight itself, by Jensen's inequality and the Gaussian MGF. With $\nu^2$ known, subtracting $\nu^2(x)/2$ from the log-weight removes the bias exactly. With $\nu^2$ unknown, plugging in $\widehat{V}/2$ leaves an order $\nu^4/(N-1)$ residual via the $\chi^2_{N-1}$ MGF. The exact fix of Ceperley & Dewing (1999) replaces $\widehat{V}/2$ by the *penalty* $u_B$ defined as

$$u_B(\widehat{V}(x), N) = -\log h_B\left( \widehat{V}(x), \mu \right) , \quad h_B(\gamma^2, \mu) := \sum_{k=0}^\infty \frac{(-1)^k}{2^k k!} \frac{\mu^k \Gamma(\mu)}{\Gamma(\mu+k)} (\gamma^2)^k , \quad \mu = \tfrac{N-1}{2} ,$$

which has the closed form $h_B(\gamma^2, \mu) = \Gamma(\mu)(2/(\mu\gamma^2))^{(\mu-1)/2} \mathcal{J}_{\mu-1}(\sqrt{2\mu\gamma^2})$, with $\mathcal{J}_\nu$ the Bessel function of the first kind. By construction, $\mathbb{E}[h_B(\widehat{V}(x), \mu)] = \exp(-\nu^2(x)/2)$, so substituting $\log |\det J_{\mathrm{T}}(x)|$ by $\widehat{L}(x) - u_B(\widehat{V}(x), N)$ in the log IS weight yields an estimator that is *exactly* unbiased on the weight itself.

Bessel functions being numerically unstable, we approximate $u_B$ by truncating its power series, the first terms reading

$$u_B(\gamma^2, N) = \frac{\gamma^2}{N} + \frac{\gamma^4}{4(N+1)} + \frac{\gamma^6}{3(N+1)(N+3)} + O(\gamma^8) . \tag{35}$$

This re-introduces a residual bias of strictly higher order than $\nu^4/(N-1)$, shown to be empirically negligible in Appendix D.4. This entire correction extends verbatim to AIS and SMC weights, applied at each level of the deterministic ladder.

**(b) The Metropolis–Hastings case.** Assume now T is an involution $(\text{T} \circ \text{T} = \text{I}_d)$. With a noisy log-Jacobian, define the random acceptance probability

$$\widehat{\alpha}(x) = \min\left(1, \tfrac{\pi(\text{T}(x))}{\pi(x)} \exp\left(\widehat{L}(x) - u_B(\widehat{V}(x), N)\right)\right) .$$

A direct Gaussian-shift computation (Ceperley & Dewing, 1999, Section II) shows that $\mathbb{E}[\widehat{\alpha}(x)]$ satisfies the detailed-balance criterion w.r.t. $\pi$, provided that $\nu^2(\text{T}(x)) = \nu^2(x)$ (and, in fact, $\widehat{V}(\text{T}(x)) = \widehat{V}(x)$ deterministically). This recipe extends verbatim to RE swaps, by viewing the joint swap $\bar{\text{T}}_k$ on the extended space as the relevant involution.

**Constructing the noisy log-Jacobian estimator.** It remains to provide a concrete $\widehat{\ell}(x)$ satisfying the working assumption. In our setting, the matrix logarithm of Lemma 4 gives an exact power-series expansion

$$\log|\det J_{\text{T}}(x)| = \text{Tr}\log J_{\text{T}}(x) = \sum_{i=1}^{\infty} a_i \,\text{Tr}(M(x)^i) ,$$

where $M(x)$ is the symmetric matrix derived from the midpoint Hessian (cf. Lemma 26 for the Euler-IM case, Lemmas 18 and 24 for the EI counterparts) and $\{a_i\}_{i \geq 1}$ are explicit scalar coefficients. Truncating at order $I$ introduces a deterministic error that is empirically negligible (cf. Appendix D.4). We then form $\widehat{\ell}(x)$ as a stochastic estimate of the truncated sum, *sharing the same random probes across all powers $i = 1, \dots, I$ within a single draw*; only the draws across $n = 1, \dots, N$ are independent.

*(i) Hutchinson estimator.* With $v \sim \text{N}(0, \text{I}_d)$, set

$$\widehat{\ell}(x) = \sum_{i=1}^{I} a_i \, v^\top M(x)^i v .$$

Each $v^\top M^i v$ requires $i$ successive Hessian-vector products against the same probe $v$. Averaging $N$ i.i.d. probes $\{v_n\}_{n=1}^{N}$ yields $\widehat{L}$, $\widehat{V}$.

*(ii) Hutch++ estimator.* With $S \sim \text{N}(0, \text{I}_d \otimes \text{I}_r)$, $Q = \text{QR}(M(x)S)$, $P = \text{I}_d - QQ^\top$ and $g \sim \text{N}(0, \text{I}_d)$, set

$$\widehat{\ell}(x) = \sum_{i=1}^{I} a_i \left\{ \text{Tr}(Q^\top M(x)^i Q) + (Pg)^\top M(x)^i (Pg) \right\} .$$

Sharing the same sketch $Q$ across all powers is principled because $M(x)$ is symmetric, so $M(x)$ and all its powers share eigenspaces: a sketch capturing the top eigenspace of $M$ is informative for every $M^i$. We treat $S$ as fixed (its randomness is absorbed in the deterministic prefactor) and average over i.i.d. probes $\{g_n\}_{n=1}^{N}$, again shared across powers, yielding $\widehat{L}$, $\widehat{V}$.

In the RE case, the joint log-Jacobian of the swap $\bar{\text{T}}_k$ decouples as a sum of forward and backward midpoint contributions, see (22) and Proposition 3. The working assumption only requires an unbiased Gaussian estimator of the *sum*, so we directly form a single $\widehat{\ell}(x_k, x_{k+1})$ by adding the Hutchinson (resp. Hutch++) estimators of the two contributions, sharing the same probes $v_n$ (resp. sketch $S$ and probes $g_n$) between them within each draw, and resampling across draws. This shared-probe construction is what enforces $\widehat{V}(\bar{\text{T}}_k(x)) = \widehat{V}(x)$ deterministically and thus preserves detailed balance under the MH correction of (b).

**Limitations of the proposed statistical estimation.** The penalty correction of Ceperley & Dewing (1999) relies on a Gaussianity assumption on the log-determinant estimator, which is justified empirically in the considered dimensional settings. This approximation may however deteriorate in very low-dimensional settings ($d \leq 10$), which fall outside the scope of our empirical study. A natural direction for future work would be to tailor the correction more precisely to the distribution of our Hutchinson estimators, for instance by replacing the chi-squared assumption on the variance estimate with a more accurate generalized chi-squared model, potentially yielding tighter corrections in such edge cases.

# D   Experimental details

## D.1   Target details

**Definition of the *TwoModes* target distribution.**   For our target $\pi$, we first consider the Gaussian mixture introduced in Grenioux et al. (2025), whose density is defined over $\mathbb{R}^d$ as

$$\gamma(x) = \tfrac{2}{3}\mathrm{N}(x; -a\mathbf{1}_d, \Sigma_1) + \tfrac{1}{3}\mathrm{N}(x; a\mathbf{1}_d, \Sigma_2) \,,$$

where $\Sigma_1, \Sigma_2 \in \mathbb{R}^{d\times d}$ are diagonal covariance matrices. The diagonal entries of $\Sigma_1$ are given by $(\Sigma_1)_{i,i} = \frac{i}{d}\sigma_{\max}^2 + \frac{d-i}{d}\sigma_{\min}^2$, and those of $\Sigma_2$ are the reverse of $\Sigma_1$: $(\Sigma_2)_{i,i} = (\Sigma_1)_{d-i,d-i}$, with $\sigma_{\max}^2 = 0.2$ and $\sigma_{\min}^2 = 0.01$ (hence, the conditioning number of each covariance matrix is 20). We consider three main hyperparameter settings throughout the paper: $(a, d) = (1.0, 128)$, denoted by "Low distance – High dimension", $(a, d) = (10.0, 16)$, denoted by "High distance – Low dimension", and $(a, d) = (5.0, 64)$, denoted by "Medium distance – Medium dimension". In our experiments, we rather consider the "standardized" version of $\gamma$ (*i.e.*, with zero mean and unit covariance), given by the unnormalized density $x \mapsto \gamma(\Sigma_\pi^{1/2}x + \mathbf{m}_\pi)$, where $\mathbf{m}_\pi$ is the exact mean of $\pi$, and $\Sigma_\pi$ is a diagonal covariance matrix whose entries correspond to the exact marginal variances of $\pi$ along each coordinate. For this target, the mode-weight metric evaluates the Monte Carlo estimate, computed from generated samples, of the largest mode weight (*i.e.*, 66.67%); see (Grenioux et al., 2025, Section 3.1) for details. In our experiments, we display boxplots of this estimate to assess both its bias and variance, following the methodology Grenioux et al. (2025).

**Definition of the *ManyModes* target distribution.**   We also consider the $d$-dimensional Gaussian mixture with $L > 2$ components introduced in (Noble et al., 2025, Appendix H.1) defined for any $x \in \mathbb{R}^d$ by its density $\gamma(x) = \sum_{\ell=1}^{L} w_\ell \mathrm{N}(x; \mathbf{m}_\ell, 0.5\mathrm{I}_d)$, where the means $\{\mathbf{m}_\ell\}_{\ell=1}^{L}$ are sampled independently from $\mathrm{U}([-L, L]^d)$, and the weights $\{w_\ell\}_{\ell=1}^{L}$ form a strictly increasing geometric sequence such that $w_L/w_1 = 3$. We will consider $L \in \{4, 16, 64\}$ with fixed dimension $d = 32$. Moreover, we apply the same standardization procedure as for the *TwoModes* targets. To evaluate how well mode weights are recovered, we compute the Total Variation (TV) distance between the true mode weight histogram and its Monte Carlo estimate.

**Definition of the *ManyWell* target distribution.**   To move beyond synthetic Gaussian-mixture targets, we additionally consider the $d$-dimensional *ManyWell* distribution, defined for even $d$ as the product of $d/2$ independent copies of the two-dimensional Double-Well distribution (Noé et al., 2019), whose density is

$$\gamma_{1,2}(x_1, x_2) = \frac{1}{Z_{1,2}} \exp\left( \frac{x_1}{2} + 6x_1^2 - x_1^4 - \frac{x_2^2}{2} \right) \,,$$

where $Z_{1,2}$ denotes its tractable normalizing constant. Following the construction of Midgley et al. (2023b), each Double-Well factor is evaluated on a distinct pair of coordinates of the $d$-dimensional input. The resulting density therefore factorizes as

$$\gamma^{(d)}(x) = \prod_{i=1}^{d/2} \gamma_{1,2}\left( x_{2i-1}, x_{2i} \right) \,,$$

and its log-normalizing constant is given by $\log Z^{(d)} = (d/2)\log Z_{1,2}$. Whereas Midgley et al. (2023b) considered only the case $d = 32$, we evaluate the target at dimensions $d \in \{16, 32, 64\}$. The resulting distributions are highly multimodal, with $2^{d/2}$ modes corresponding to all possible combinations of the two modes of each Double-Well factor. Consequently, the number of modes grows exponentially with the dimension, making the higher-dimensional instances increasingly challenging for sampling methods. To generate exact reference samples, we follow the procedure described in (Midgley et al., 2023b, Appendix E.1). Before applying the density-learning and sampling methods, we also use the same standardization procedure as for the Gaussian-mixture targets. For this target, directly assessing recovery of all mode weights is impractical because of the high number of modes. Instead, we exploit the factorization of the target into $d/2$ independent Double-Well pairs, whose two mode weights depend solely on the first coordinate and are tractable. For each pair, we estimate the mode probabilities from the generated samples and compute their TV distance from the true weights as done for the *ManyModes* target. We then average this quantity across all pairs to obtain a global mode-weight-like recovery metric coined "averaged TV" (aTV).

Table 2: Hyperparameter grid for each DSM-regularized objective on *TwoModes* and *ManyModes*. The initialization scheme is fixed per objective family (warm-start: 1000-epoch DSM pretrain + 1000-epoch training with the target loss; scratch: 2000 epochs from random initialization).

| Objective | $\lambda_{\text{reg}}$ | Other hyperparameters | Initialization |
|---|---|---|---|
| DSM | — | — | warm-start |
| TSM+DSM | — | — | warm-start |
| tSM+DSM | — | — | scratch |
| LFPE+DSM | $\{10^{-3}, 10^{-2}, 10^{-1}, 1, 10\}$ | — | scratch |
| aLFPE+DSM | $\{10^{-3}, 10^{-2}, 10^{-1}, 1, 10\}$ | — | scratch |
| RNE+DSM | $\{10^{-3}, 10^{-2}, 10^{-1}, 1, 10\}$ | $\delta \in \{10^{-5}, 10^{-4}, 10^{-3}, 10^{-2}, 10^{-1}\}$ | warm-start |
| DiffCLF+DSM | $\{10^{-2}, 10^{-1}, 1, 10\}$ | $n_{\text{clf}} \in \{4, 6, 8\}$ | warm-start |

Table 3: Hyperparameter grid for each DSM-regularized objective on *ManyWell*. The initialization scheme is fixed per objective family (warm-start: 1000-epoch DSM pretrain + 1000-epoch training with the target loss; scratch: 2000 epochs from random initialization).

| Objective | $\lambda_{\text{reg}}$ | Other hyperparameters | Initialization |
|---|---|---|---|
| DSM | — | — | warm-start |
| TSM+DSM | — | — | warm-start |
| tSM+DSM | — | — | scratch |
| LFPE+DSM | $\{10^{-3}, 10^{-2}\}$ | — | scratch |
| aLFPE+DSM | $\{10^{-3}, 10^{-2}\}$ | — | scratch |
| RNE+DSM | $\{10^{-3}, 1, 10\}$ | $\delta \in \{10^{-4}, 10^{-2}\}$ | warm-start |
| DiffCLF+DSM | $\{10^{-2}, 1, 10\}$ | $n_{\text{clf}} \in \{4, 8\}$ | warm-start |

## D.2 Training and sampling parameters

**Diffusion model training details.** As explained in Section 6.1, we consider two types of architectures $\mathcal{E}^\theta$ to learn the log-densities of DMs: (i) a pinned architecture, ensuring exact recovery of the target distribution $\pi$ at $t_0$ and (ii) an hardcoded architecture, without any boundary condition fixed at training stage. For both of these models, we rely on an enhanced version of the score-like architecture advocated by Richter et al. (2023), denoted by $\mathbf{s}^\theta : (t, x) \in \mathbb{R}^{d+1} \to \mathbf{s}_t^\theta(x) \in \mathbb{R}^d$, which is a 4-layer 128-width fully connected network with GeLU activations, position-input preconditioning (based on target mean and scalar variance), time-input preconditioning (based on Fourier embedding) and time-input skip-connections at every layer. Our models are the following: (a) *Pinned*: given (26), we set $g_t^\theta(x) = \frac{1}{2} \left\| \mathbf{s}_t^\theta(x) \right\|_2^2$ and $f^\theta$ to be a scalar-to-scalar 4-layer 64-width fully connected network with GeLU activations and the same time-input preconditioning as in $\mathbf{s}^\theta$; (b) *Hardcoded*: we adopt the network preconditioning strategy proposed by Thornton et al. (2025) on $\mathbf{s}^\theta$.

For each *TwoModes*, *ManyModes*, *ManyWell* setting varying in dimension, mode separation, and/or number of modes, we train both network architectures with the seven objectives described in Appendix A.3. We use the parameterizations $\mathbf{U}_t^\theta = -\mathcal{E}_t^\theta$, $\mathbf{s}_t^\theta = -\nabla_x \mathcal{E}_t^\theta$, and $\mathbf{u}_t^\theta = -\partial_t \mathcal{E}_t^\theta$. In the case of DiffCLF, the log-normalizing constant $\log \mathcal{Z}_t(\theta)$ is modeled by a scalar-input scalar-output neural network. For all targets, diffusion models are trained on datasets of size $60,000$, with batch size $1024$.

Following the energy-matching literature, we consider the DSM-regularized objectives $\mathcal{L}_{\text{TSM+DSM}} = \mathcal{L}_{\text{TSM}} + \mathcal{L}_{\text{DSM}}$, $\mathcal{L}_{\text{tSM+DSM}} = \mathcal{L}_{\text{tSM}} + \mathcal{L}_{\text{DSM}}$, and $\mathcal{L}_{X+\text{DSM}} = \lambda_{\text{reg}} \mathcal{L}_X + \mathcal{L}_{\text{DSM}}$ for $X \in \{\text{LFPE}, \text{aLFPE}, \text{RNE}, \text{DiffCLF}\}$, with $\lambda_{\text{reg}} > 0$ being a tunable hyperparameter. Score-matching losses (DSM, TSM, tSM) are rescaled by $d^{-1}$, and energy-matching losses (LFPE, aLFPE, RNE) by $d^{-2}$; time steps are sampled uniformly in log-SNR space (Kingma et al., 2021).

Table 4: Selected hyperparameters per (target, architecture). Entries show $\lambda_{\mathrm{reg}}$ for LFPE+DSM and aLFPE+DSM, $(\lambda_{\mathrm{reg}}, \delta)$ for RNE+DSM, and $(\lambda_{\mathrm{reg}}, n_{\mathrm{clf}})$ for DiffCLF+DSM. DSM, TSM+DSM and tSM+DSM have no tunable hyperparameter and are omitted.

| Target | Architecture | LFPE+DSM | aLFPE+DSM | RNE+DSM | DiffCLF+DSM |
|---|---|---|---|---|---|
| TwoModes ($d{=}16, a{=}10$) | precond | $10^{-3}$ | $10^{-2}$ | $(1, 10^{-3})$ | $(10^{-2}, 4)$ |
| | pinned | $10^{-1}$ | $10^{-3}$ | $(10^{-1}, 10^{-4})$ | $(10^{-2}, 8)$ |
| TwoModes ($d{=}64, a{=}5$) | precond | $10^{-3}$ | $10^{-3}$ | $(1, 10^{-3})$ | $(10^{-2}, 6)$ |
| | pinned | $10^{-2}$ | $10^{-2}$ | $(10, 10^{-3})$ | $(10^{-1}, 4)$ |
| TwoModes ($d{=}128, a{=}1$) | precond | $10^{-3}$ | $10^{-3}$ | $(1, 10^{-2})$ | $(10^{-2}, 8)$ |
| | pinned | $10^{-1}$ | $10$ | $(10^{-1}, 10^{-2})$ | $(10, 8)$ |
| ManyModes ($d{=}32, L{=}4$) | precond | $10^{-3}$ | $10^{-3}$ | $(1, 10^{-3})$ | $(10^{-1}, 4)$ |
| | pinned | $10^{-1}$ | $10$ | $(10^{-2}, 10^{-5})$ | $(10, 8)$ |
| ManyModes ($d{=}32, L{=}16$) | precond | $10^{-3}$ | $10^{-2}$ | $(10^{-1}, 10^{-5})$ | $(10^{-2}, 6)$ |
| | pinned | $10^{-2}$ | $10^{-2}$ | $(10^{-1}, 10^{-4})$ | $(10, 8)$ |
| ManyModes ($d{=}32, L{=}64$) | precond | $10^{-1}$ | $10^{-1}$ | $(10^{-3}, 10^{-2})$ | $(10^{-2}, 6)$ |
| | pinned | $10$ | $10^{-2}$ | $(10^{-1}, 10^{-4})$ | $(10^{-1}, 8)$ |
| ManyWell ($d{=}16$) | precond | $10^{-3}$ | $10^{-3}$ | $(10^1, 10^{-2})$ | $(1, 8)$ |
| | pinned | $10^{-3}$ | $10^{-2}$ | $(10^1, 10^{-2})$ | $(10^1, 8)$ |
| ManyWell ($d{=}32$) | precond | $10^{-3}$ | $10^{-3}$ | $(10^1, 10^{-2})$ | $(10^{-2}, 8)$ |
| | pinned | $10^{-3}$ | $10^{-2}$ | $(10^1, 10^{-2})$ | $(10^1, 8)$ |
| ManyWell ($d{=}64$) | precond | $10^{-2}$ | $10^{-2}$ | $(10^1, 10^{-4})$ | $(10^{-2}, 4)$ |
| | pinned | $10^{-3}$ | $10^{-2}$ | $(10^1, 10^{-2})$ | $(10^1, 8)$ |

The initialization scheme is fixed per objective family: TSM+DSM, RNE+DSM, and DiffCLF+DSM are warm-started from a 1000-epoch DSM pretrain and trained for 1000 further epochs with their target loss, while tSM+DSM, LFPE+DSM, and aLFPE+DSM are trained from random initialization for 2000 epochs (for the latter methods, DSM warmstart led to degraded performance); the standalone DSM baseline likewise uses the 1000-epoch pretrain followed by 1000 further training epochs, so the total compute budget is 2000 epochs in every case. The full hyperparameter grid is summarized in Tables 2 and 3. All trainings use AdamW (Loshchilov & Hutter, 2019) with default hyperparameters and learning rate $10^{-4}$.

Assessing log-density learning is notoriously difficult and remains an active research topic; we rank trained models lexicographically by three criteria of decreasing priority: (a) the effective sample size between learned and ground-truth marginals (to maximize); (b) the Fisher divergence between the same marginals (to minimize); and (c) the global classification loss of OuYang et al. (2026) (to minimize). Criteria (a) and (b) are averaged over $t \in [0, T]$ and computed from exact marginal samples. For each of the twelve main configurations (two architectures $\times$ six targets) and each DSM-regularized objective, this rule selects $\lambda_{\mathrm{reg}}$ and the objective-specific hyperparameter; the resulting choices are listed in Table 4. This produces a single neural network per (target, architecture, objective) triple, reused by all aMC samplers to ensure a fair comparison.

**General remarks on annealed sampling methods.** Since the considered targets are systematically standardized, we set the base distribution as their Gaussian approximation $\pi^{\mathrm{base}} = \mathrm{N}(0, \mathrm{I}_d)$, for both tempering and diffusion-based approaches. We recall that, when using second-order approaches, *i.e.*, methods that require access to the Hessians of the bridging log-densities, we only exploit the diagonal of these Hessians to ensure a good compromise between accuracy and computational efficiency in high dimensional scenarios.

All SMC variants (also including diffusion-enhanced SMC samplers), as well as the standard AIS sampler, apply 160 MCMC steps (including 128 warm-up steps) for local exploration at each level $k \in \{0, \ldots, K\}$, using *Metropolis-Adjusted Langevin Algorithm* (MALA) (Roberts & Tweedie, 1996). Following Chopin & Papaspiliopoulos (2020), we do not perform resampling systematically, but instead apply it adaptively based on the current IS weights, using an effective sample size threshold of 30% with systematic resampling scheme.

For RE-based sampling methods, we perform a total of 24,576 MCMC steps (including 8,192 warm-up steps), with local exploration made via MALA and swaps occurring every 8 steps, thereby *defining the computational budget of RE (with or without transition kernels) to be comparable to the footprint of the SMC setting with the largest number of levels* (*i.e.*, $K = 256$ where SMC performs the best), see the last row of Figure 3. For RE, we consider two intermediate-level initializations: a *score-informed* one, where each level is populated by simulating the denoising SDE (5) from $\pi^{\mathrm{base}}$, and a *base* one, sampling each level independently from $\pi^{\mathrm{base}}$ (as in tempering). We use the base initialization by default. The ablation in Appendix D.4 shows this choice is essentially neutral for deterministic and second-order stochastic backbones, but *degrades* performance with first-order stochastic kernels which is consistent with Section 3.3, where these kernels were already found to be uninformative for between-level transitions.

For all variants of annealed samplers based on deterministic transitions, we use by default $M = 4$ fixed-point iterations, truncate the power series at order $I = 3$, and use 39 samples in the Hutch++ estimator for the first-order variant. In Appendix D.4, we provide a precise ablation study of these three hyperparameters to evaluate their individual effect.

Finally, all local MALA steps are performed with an initial step size of 0.01; then, its is geometrically adapted during both warm-up and effective sampling based on local MH acceptance rates, targeting 70% acceptance.

**Inference and sampling details.** For diffusion-based methods, whether the path is learned or fixed, we adopt by default the SNR-adapted discretization from Appendix B.4 to establish the annealing levels : when combined with a learned path, this ensures consistency between learning and inference stages. For tempering paths, the sequence of densities defined by (18) is employed with the $\Lambda$-optimal schedule (computed for each value of $K$), proposed by Syed et al. (2025) in the case of AIS and SMC, and Syed et al. (2021; 2022) in the case of RE. For all AIS/SMC samplers, we use 8,192 particles, and keep, for each particle, when it is available, the last 32 MCMC samples generated at the last level (properly reweighted using the associated importance weights) to compute the metrics. For all RE methods, we use 4 parallel RE chains; once the fixed global number of MCMC steps is reached, each of these chains is subsampled by retaining only the last local MCMC state before each swap. For all annealed samplers, we repeat the sampling run 8 times to produce averaged results in the plots.

**Estimation of** $\log \mathcal{Z}$**.** We summarize how each scheme of Section 2.2 estimates $\log \mathcal{Z}$ and the bias each estimator carries. Throughout, $\rho$ is a proposal density and $\{p_k\}_{k=0}^{K}$ is an *arbitrary* density path with $p_0 = \pi$, $p_K = \pi^{\mathrm{base}}$, and per-level normalising constants $\mathcal{Z}_k$ (so $\mathcal{Z}_0 = \mathcal{Z}$ and $\mathcal{Z}_K = 1$).

**IS.** With $X^i \overset{\mathrm{iid}}{\sim} \rho$, the estimator $\widehat{\mathcal{Z}}_{\mathrm{IS}} = N^{-1} \sum_{i=1}^{N} \tilde{\pi}(X^i)/\rho(X^i)$ is *unbiased* and a.s. consistent for $\mathcal{Z}$; by Jensen, $\log \widehat{\mathcal{Z}}_{\mathrm{IS}}$ is *negatively biased* but consistent.

**SNIS.** When the proposal is normalised ($\mathcal{Z}_\rho = 1$), the unnormalised weights $w^i = \tilde{\pi}(X^i)/\rho(X^i)$ satisfy $\mathbb{E}_\rho[w] = \mathcal{Z}$, so for the normalising constant itself SNIS collapses to IS: $N^{-1} \sum_i w^i$ is unbiased and consistent for $\mathcal{Z}$ and $\log \widehat{\mathcal{Z}}$ is Jensen-biased.

**AIS.** For any forward/backward kernels $q_{k+1|k}, q_{k|k+1}$ defining $\pi_{0:K}, \rho_{0:K}$ as in (14), drawing $x_{0:K}^i \sim \rho_{0:K}$ and averaging $w^{\mathrm{AIS}}(x_{0:K}^i)$ from (15) yields $\widehat{\mathcal{Z}}_{\mathrm{AIS}} = N^{-1} \sum_i w^{\mathrm{AIS}}(x_{0:K}^i)$, which is *unbiased* and consistent for $\mathcal{Z}$; $\log \widehat{\mathcal{Z}}_{\mathrm{AIS}}$ is negatively biased and consistent. AIS reduces to IS at $K = 0$ or in the deterministic setting.

**SMC.** With the same incremental weights as AIS plus intermediate resampling, the following product-of-averages estimator remains *unbiased* for $\mathcal{Z}$ (Del Moral et al., 2006)

$$\widehat{\mathcal{Z}}_{\mathrm{SMC}} \;=\; \prod_{k=0}^{K-1} \left( \tfrac{1}{N} \sum_{i=1}^{N} w_k^i \right) .$$

Therefore, $\log \widehat{\mathcal{Z}}_{\mathrm{SMC}}$ is negatively biased and consistent. Resampling cuts weight-degeneracy variance without breaking unbiasedness.

**RE (classic swap).** For *any* path $\{p_k\}_{k=0}^K$, RE targets $\bar{\pi}_{0:K}$ and supplies, at stationarity, $T$ samples $\{X_t^k\}_{t=1}^T \sim p_k$ on every chain. The log-normaliser is then recovered *post hoc* by the telescoping identity

$$\log \mathcal{Z} \;=\; \sum_{k=0}^{K-1} \log \frac{\mathcal{Z}_k}{\mathcal{Z}_{k+1}}, \qquad \frac{\mathcal{Z}_k}{\mathcal{Z}_{k+1}} \;=\; \mathbb{E}_{p_{k+1}}\left[\frac{\tilde{p}_k(X)}{\tilde{p}_{k+1}(X)}\right] \;,$$

estimated layer-wise by free-energy perturbation (FEP) Zwanzig (1954) or, when samples from both adjacent chains are used, by the Bennett acceptance ratio (BAR) (Bennett, 1976). Each layer-ratio estimator is unbiased *given exact samples*; with finite-$T$ MCMC samples and chains coupled through swaps the product is biased and $\log \widehat{\mathcal{Z}}$ carries an additional Jensen bias at every layer, both vanishing as $T \to \infty$.

**Generalised RE.** Following Zhang et al. (2026), the classic deterministic swap of (17) between adjacent levels $k$ and $k+1$ is replaced by stochastic refinements issued from the same forward/backward kernels $q_{k+1|k}, q_{k|k+1}$ already used in AIS (14): given $(x_k, x_{k+1})$, draw $y_{k+1} \sim q_{k+1|k}(\cdot \mid x_k)$ and $y_k \sim q_{k|k+1}(\cdot \mid x_{k+1})$ and accept the swap $(x_k, x_{k+1}) \mapsto (y_k, y_{k+1})$ via the corresponding MH correction extending (13). The same telescoping identity governs $\log \mathcal{Z}$, with each layer ratio now estimated from the path-weights collected at every swap attempt:

$$\frac{\mathcal{Z}_{k+1}}{\mathcal{Z}_k} \;=\; \mathbb{E}\left[w_{k+1|k}(x_k, y_{k+1})\right] \;, \qquad w_{k+1|k}(x_k, y_{k+1}) \;:=\; \frac{\tilde{p}_{k+1}(y_{k+1})\, q_{k|k+1}(x_k|y_{k+1})}{\tilde{p}_k(x_k)\, q_{k+1|k}(y_{k+1}|x_k)} \;,$$

with $x_k \sim p_k$, $y_{k+1} \mid x_k \sim q_{k+1|k}$, and a symmetric backward weight $w_{k|k+1}(x_{k+1}, y_k)$ yielding $\mathcal{Z}_k/\mathcal{Z}_{k+1}$. Averaging the two directions yields the geometric-mean estimator, which admits a BAR refinement (Bennett, 1976). Bias/consistency match classical RE (layer-unbiased given exact samples, $\log \widehat{\mathcal{Z}}$ Jensen-biased, consistent as $T \to \infty$); and classical RE is recovered when both kernels collapse to Dirac masses (identity or deterministic swap).

## D.3 Additional metrics and results

This section presents extended experiments that complement the main findings by considering additional configurations and evaluation metrics. We first verify in Figure 8 that the number of samples used for metrics computation ($N = 8192$) is sufficient to obtain reliable estimates, by measuring the Monte Carlo variance of sliced $W_2$ and mode weights on ground-truth samples across both target families.

**$\Lambda$-optimal tempering path vs log-SNR diffusion path.** As a complement to the Sliced $W_2$ metric of Figure 1, Figure 9 reports log-normalization constant estimates in the same setting and with the same visualization convention. Tempering paths yield systematically biased estimates across all aMC samplers (especially AIS), whereas diffusion paths perform substantially better, with estimates improving as $K$ grows and SMC/RE already uniformly accurate at $K \geq 64$. This further confirms the advantage of diffusion over tempering paths on multi-modal targets.

**Complementary results in the idealized setting (A).** To complement Figure 3, we report mode-weight (Figure 10) and log-normalization constant (Figure 11) estimates on the same idealized multi-modal experiments.

- *Mode weights.* Within each aMC class, samplers split into two groups. The first, composed of zeroth and first order stochastic samplers (standard aMC baseline in red, prior diffusion-based methods in blue), performs uniformly poorly, except for SMC on *TwoModes*. The second, composed of second-order stochastic kernels (green), deterministic with Hessian (pink) and deterministic with Hutchinson (yellow), performs substantially better, with fairly uniform results within each class. As in the main paper, our first-order deterministic method matches its second-order counterpart for $K \geq 64$ in AIS/SMC and across all $K$ in RE.

- *Log-normalization constant.* Among stochastic kernels, only second-order variants give accurate AIS estimates; for SMC and RE, first-order methods are low-bias but high-variance, while second-order methods are both low-bias and low-variance. Deterministic variants display a noticeable bias on AIS and SMC but only a limited one on RE, where they also achieve substantially lower variance than their stochastic counterparts.

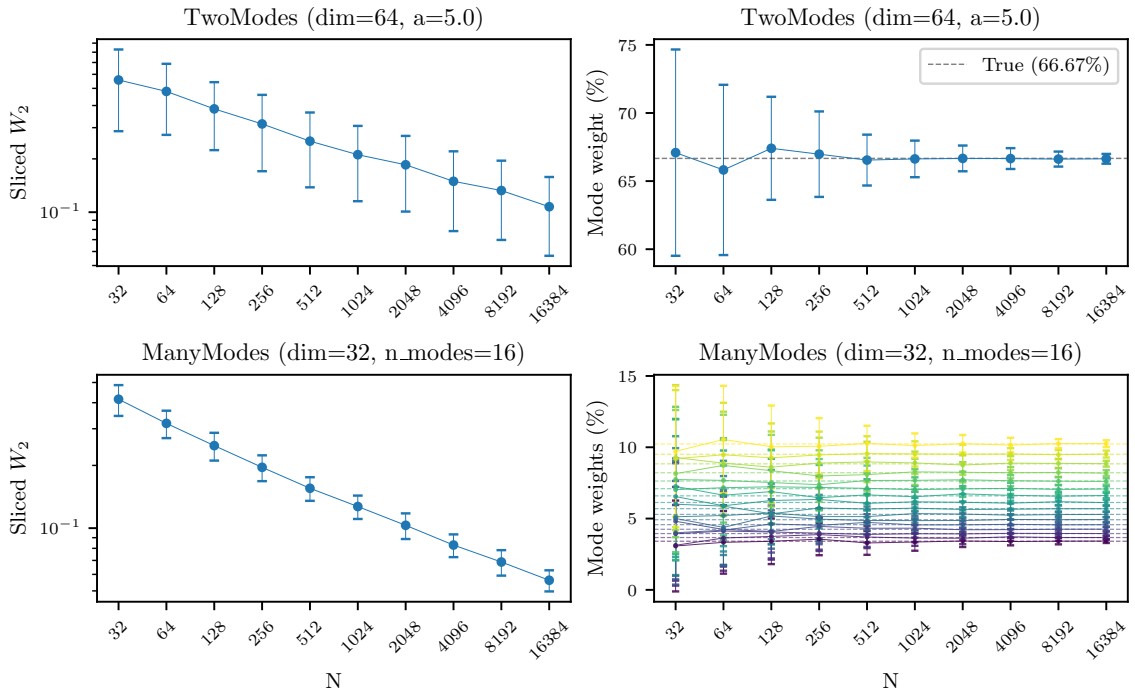

Figure 8: **Monte Carlo variance of standard metrics on ground-truth samples.** For each target, $N$ samples are drawn directly from the target distribution and passed to the standard evaluation pipeline; the procedure is repeated 128 times independently to estimate mean ± standard deviation. **(Left column)** Sliced $W_2$ distance for *TwoModes* (Medium distance – Medium dimension) and *ManyModes* (16 modes) as a function of $N$ (log scale). **(Right column)** Estimated mode weights as a function of $N$: for *TwoModes*, the weight of the dominant mode is shown together with its ground-truth value (dashed line); for *ManyModes*, each of the 16 mode weights is displayed in a distinct color, with the corresponding ground-truth value shown as a matching dashed line. At the sample size used throughout our experiments ($N = 8{,}192$), the standard deviation of sliced $W_2$ is below 0.06 on both targets, and all estimated mode weights lie within ±0.5% of their ground-truth value, confirming that the reported metrics are not dominated by Monte Carlo noise.

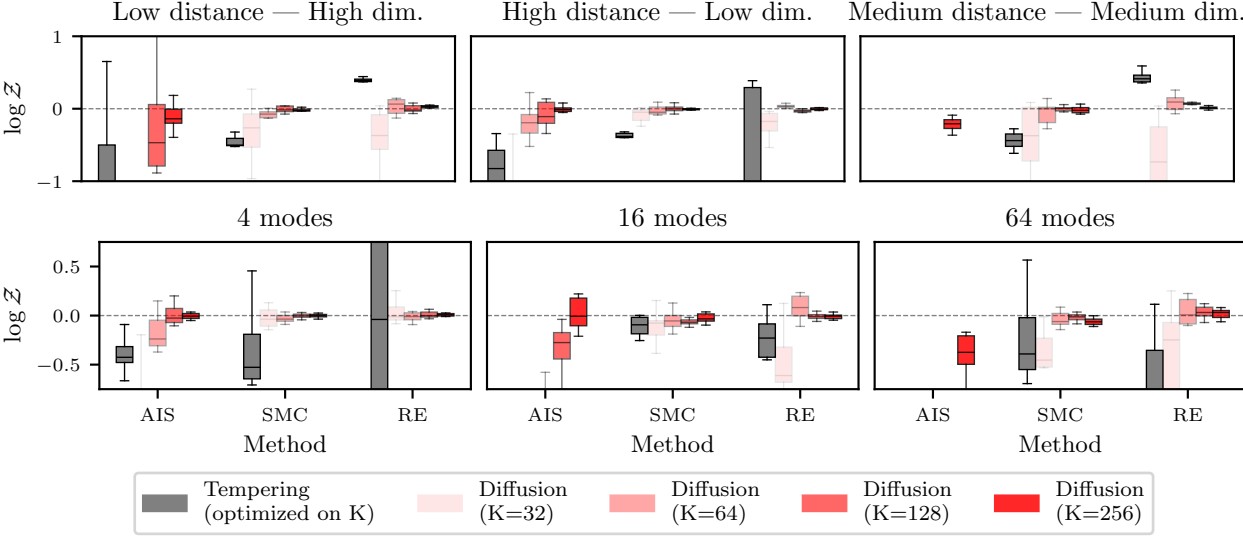

Figure 9: *Log-normalization constant estimation* **results for classic annealed samplers with diffusion (red) and tempering (grey) density paths**, when targeting *TwoModes* **(Top)** and *ManyModes* **(Bottom)** in idealized setting **(A)**. This is complementary to Figure 1.

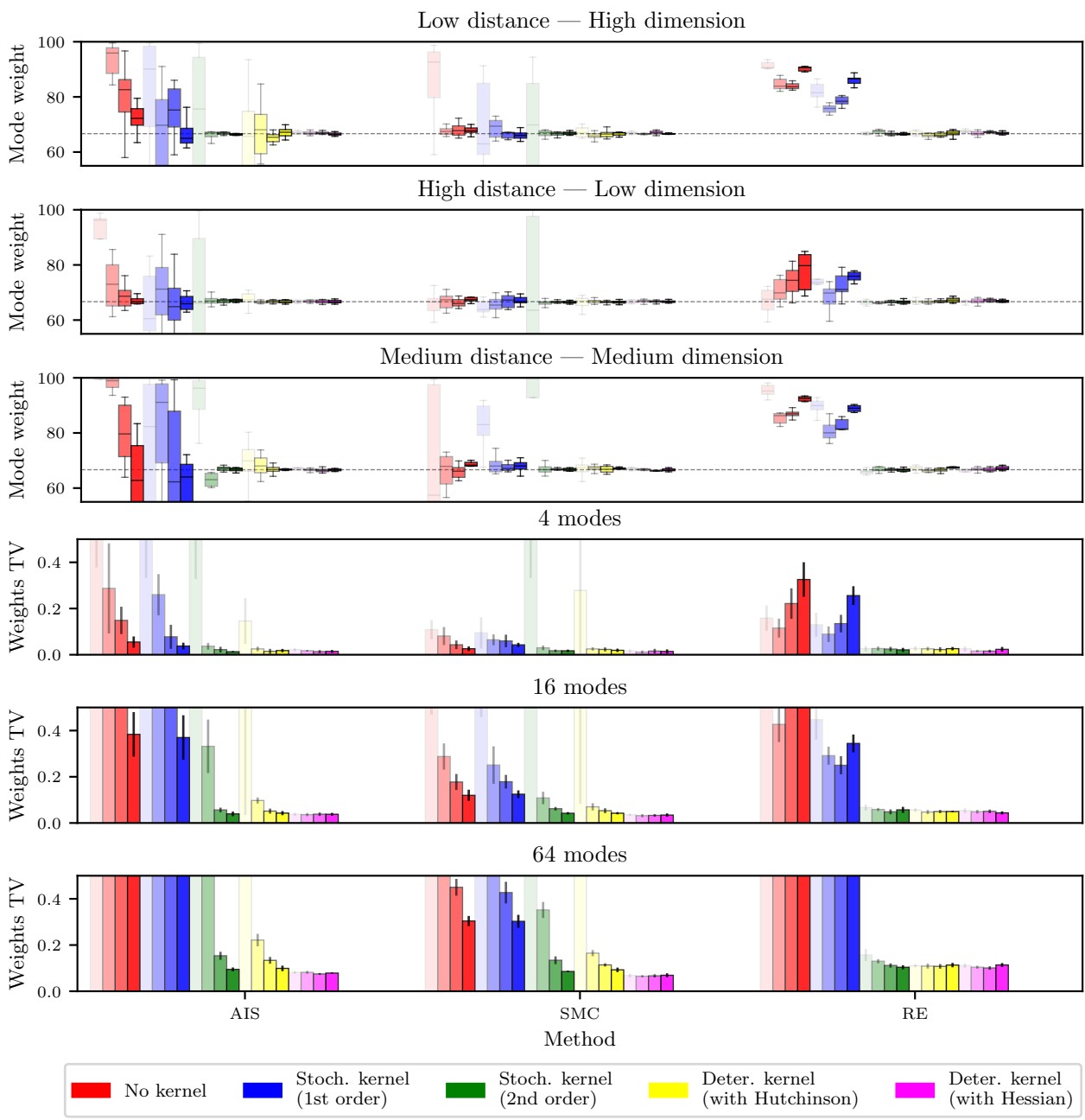

Figure 10: ***Mode weight estimation* results of DM-based aMC-BGs using different mechanisms**, when targeting *TwoModes* (**Top three rows**) and *ManyModes* (**Bottom three rows**) distributions in idealized setting **(A)**. This is complementary to the Sliced $W_2$ results displayed Figure 3, with the same visualization convention and experimental design.

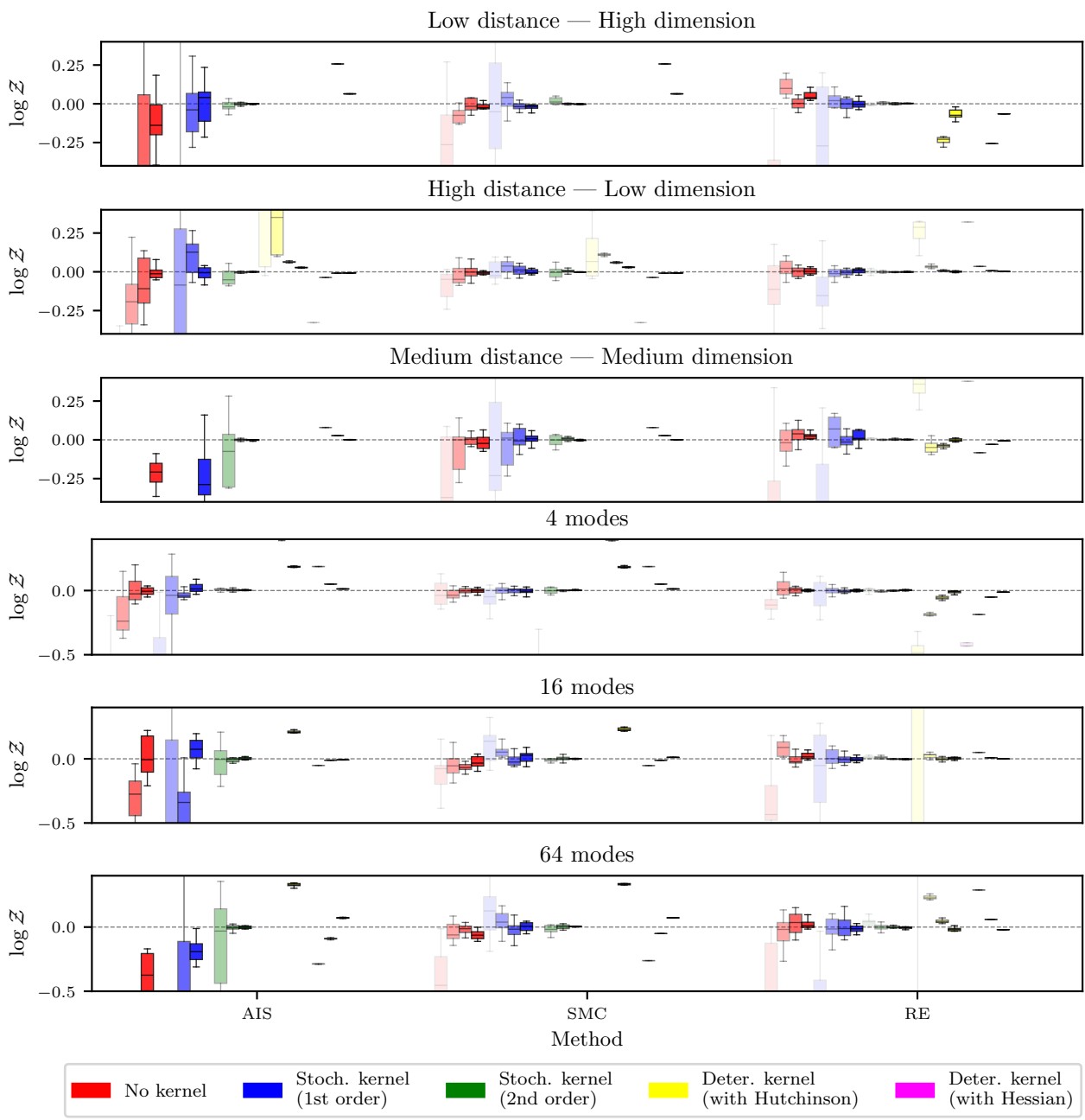

Figure 11: ***Log-normalization constant estimation* results of DM-based aMC-BGs using different mechanisms**, when targeting *TwoModes* (**Top three rows**) and *ManyModes* (**Bottom three rows**) distributions in idealized setting **(A)**. This is complementary to the Sliced $W_2$ results displayed Figure 3, with the same visualization convention and experimental design.

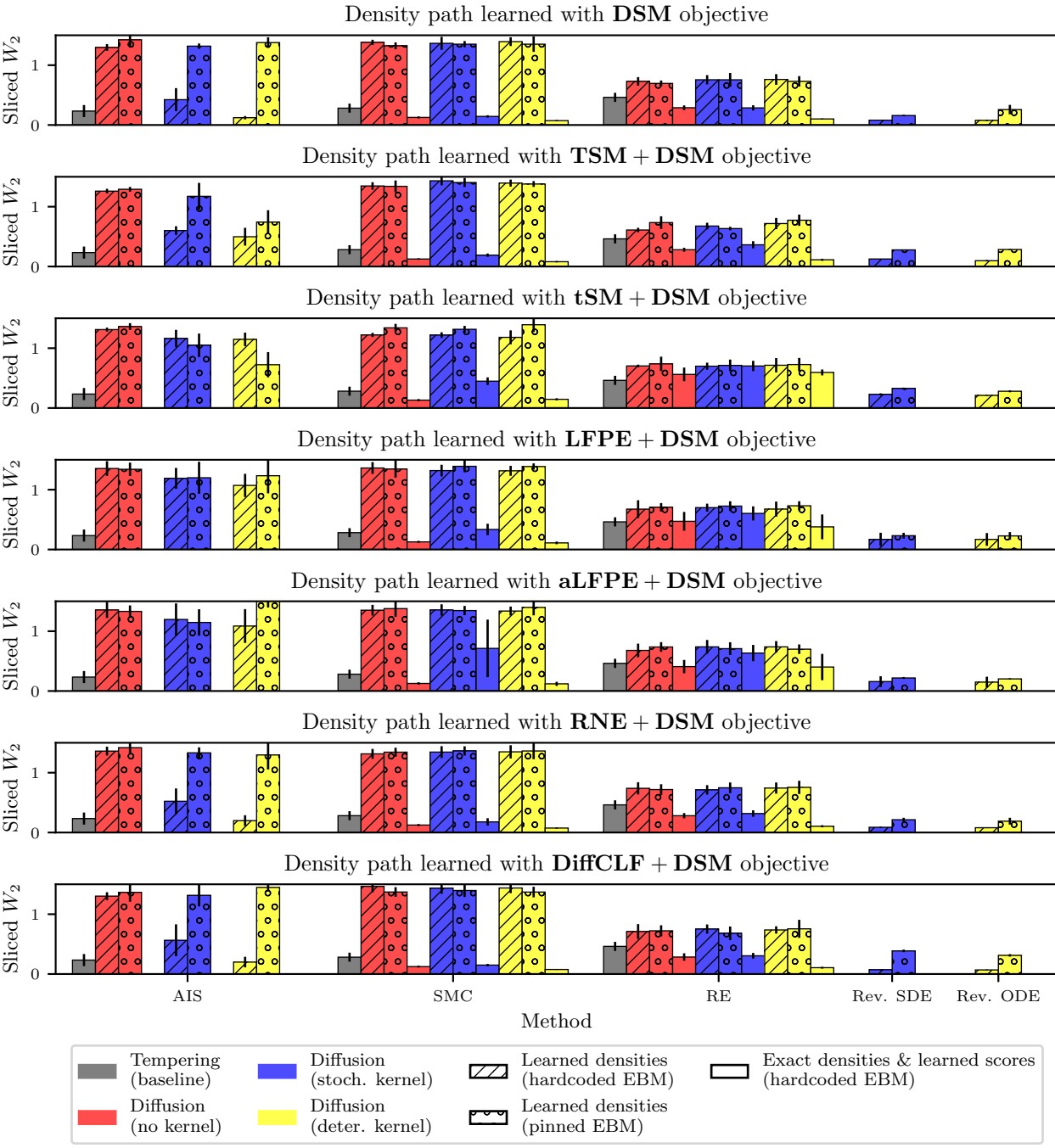

Figure 12: **_Sliced_ $W_2$ results of DM-based aMC-BG**, when targeting _ManyModes_ distribution with **16 modes** in setting **(B)** : **(From top to bottom)** the DM is trained via DSM, TSM+DSM, tSM+DSM, LFPE+DSM, aLFPE+DSM, RNE+DSM or DiffCLF+DSM objective with identical computational budget. We use the same visualization convention and experimental design as in Figure 4.

**Complementary results in realistic setting (B).** We provide the _ManyModes_ (16 modes) counterpart of Figure 4 in Figure 12, reporting the Sliced $W_2$ metric of DM-based aMC-BGs across all training objectives of Section 6.1. We further provide a per-loss zoom of Figure 7 in Figures 15 and 16. All of these results are fully consistent with the conclusions of Section 6. When carrying out this realistic experiment on the remaining target distributions, we observe the same results, across all considered metrics; we omit them to avoid overloading the manuscript. For the _ManyWell_ target, we additionally report the mode-weight estimation (aTV) and log-normalization constant estimation results in Figures 13 and 14, complementing the Sliced $W_2$ results of Figure 5; these results are consistent with the conclusions drawn in Section 6.

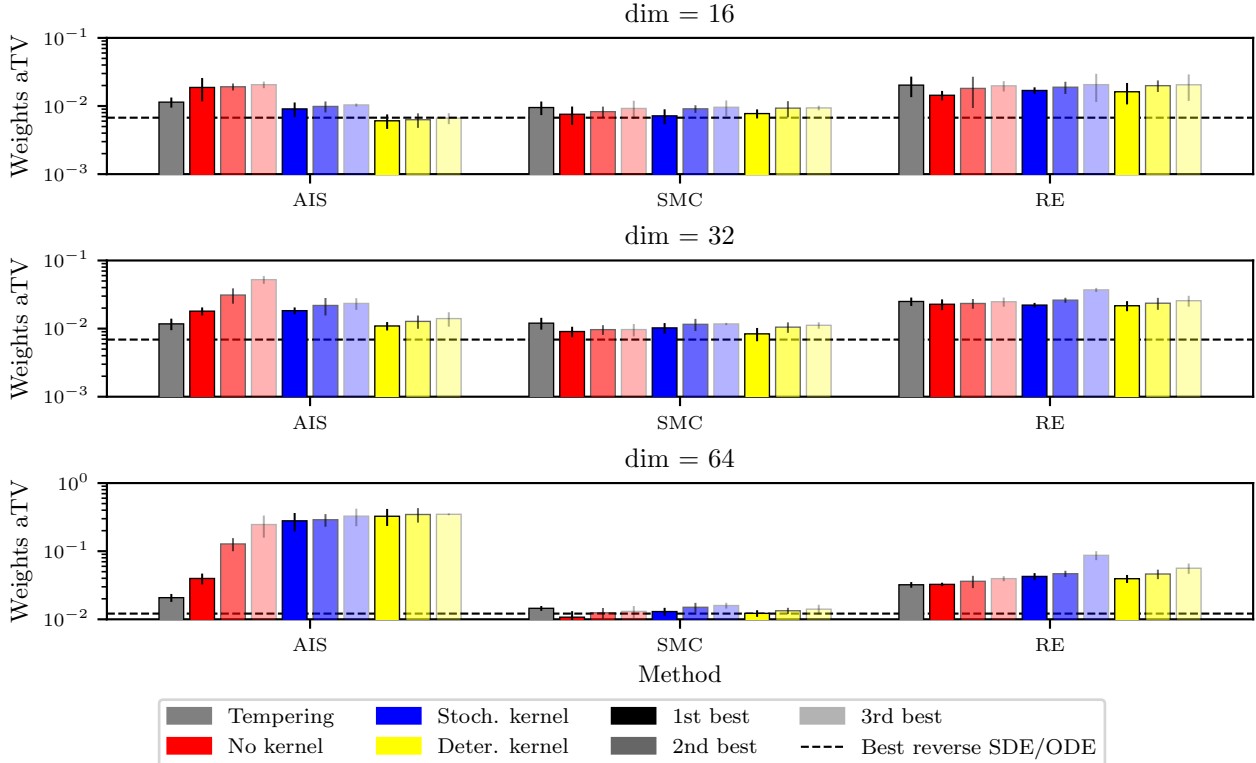

Figure 13: *Mode-weight estimation* (**aTV**) results of DM-based aMC-BGs, when targeting the *ManyWell* distribution in the practical setting **(B)**, for $d = 16$ **(top)**, $d = 32$ **(middle)**, and $d = 64$ **(bottom)**. This is complementary to Figure 5, with the same visualization convention and experimental design.

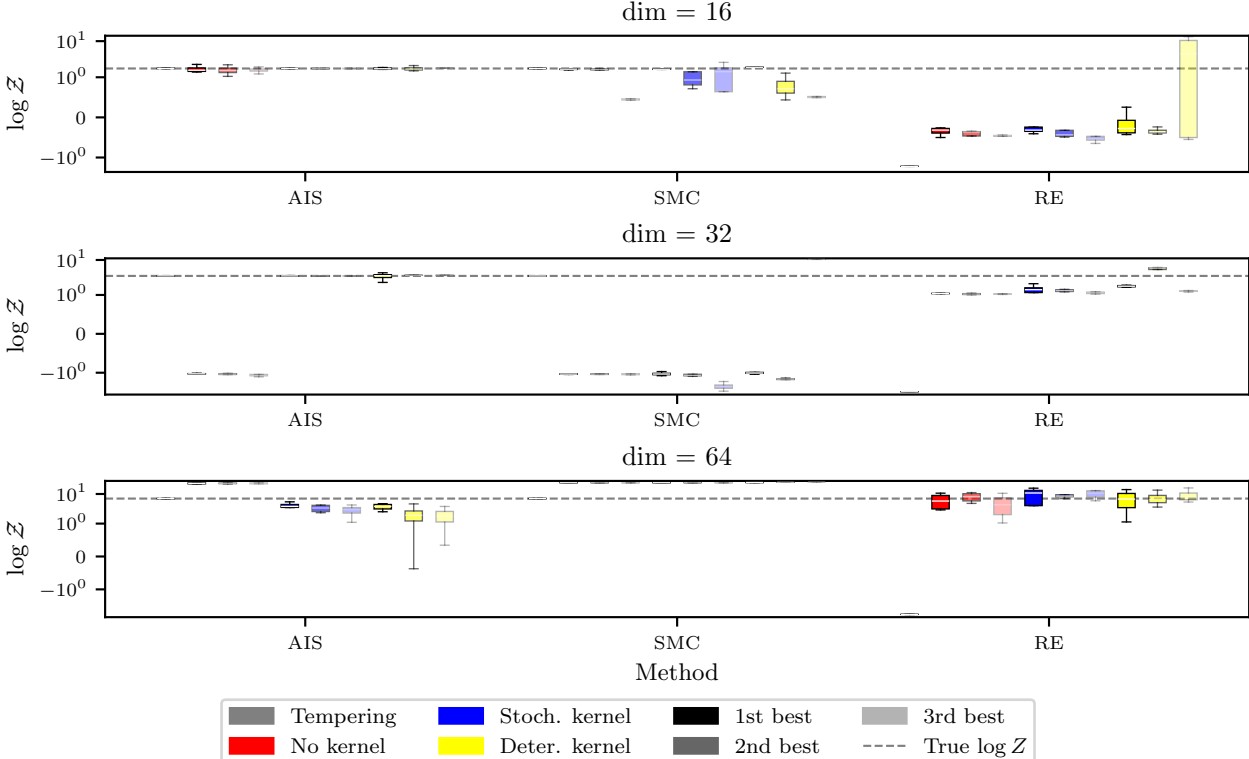

Figure 14: *Log-normalization constant estimation* results of DM-based aMC-BGs, when targeting the *ManyWell* distribution in the practical setting **(B)**, for $d = 16$ **(top)**, $d = 32$ **(middle)**, and $d = 64$ **(bottom)**. This is complementary to Figure 5, with the same visualization convention and experimental design. The dashed line indicates the true log-normalization constant.

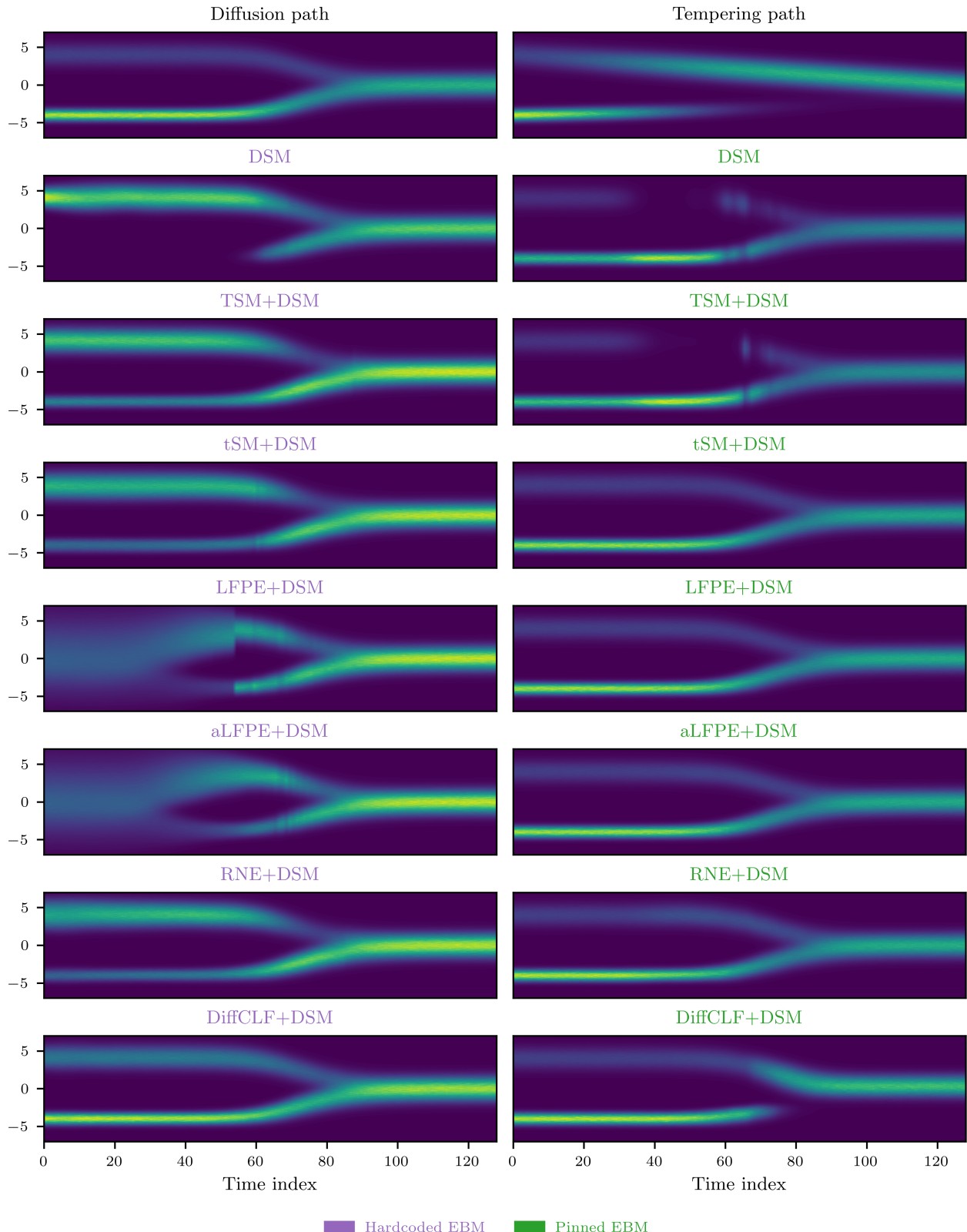

Figure 15: **Diffusion density paths bridging $\pi^{\mathbf{base}}$ (last time index) to the same *TwoModes* target as in Figure 6 (first time index). (First row)** the exact diffusion path is displayed on the left, the exact tempering path on the right, **(From second to last row)** we display the learned density path when using the DSM, TSM+DSM, tSM+DSM, LFPE+DSM, aLFPE+DSM, RNE+DSM or DiffCLF+DSM objective, with identical computational budget, **(Left)** use of hardcoded EBM, **(Right)** use of pinned EBM. This is complementary to Figure 7. Zoom in to get more details.

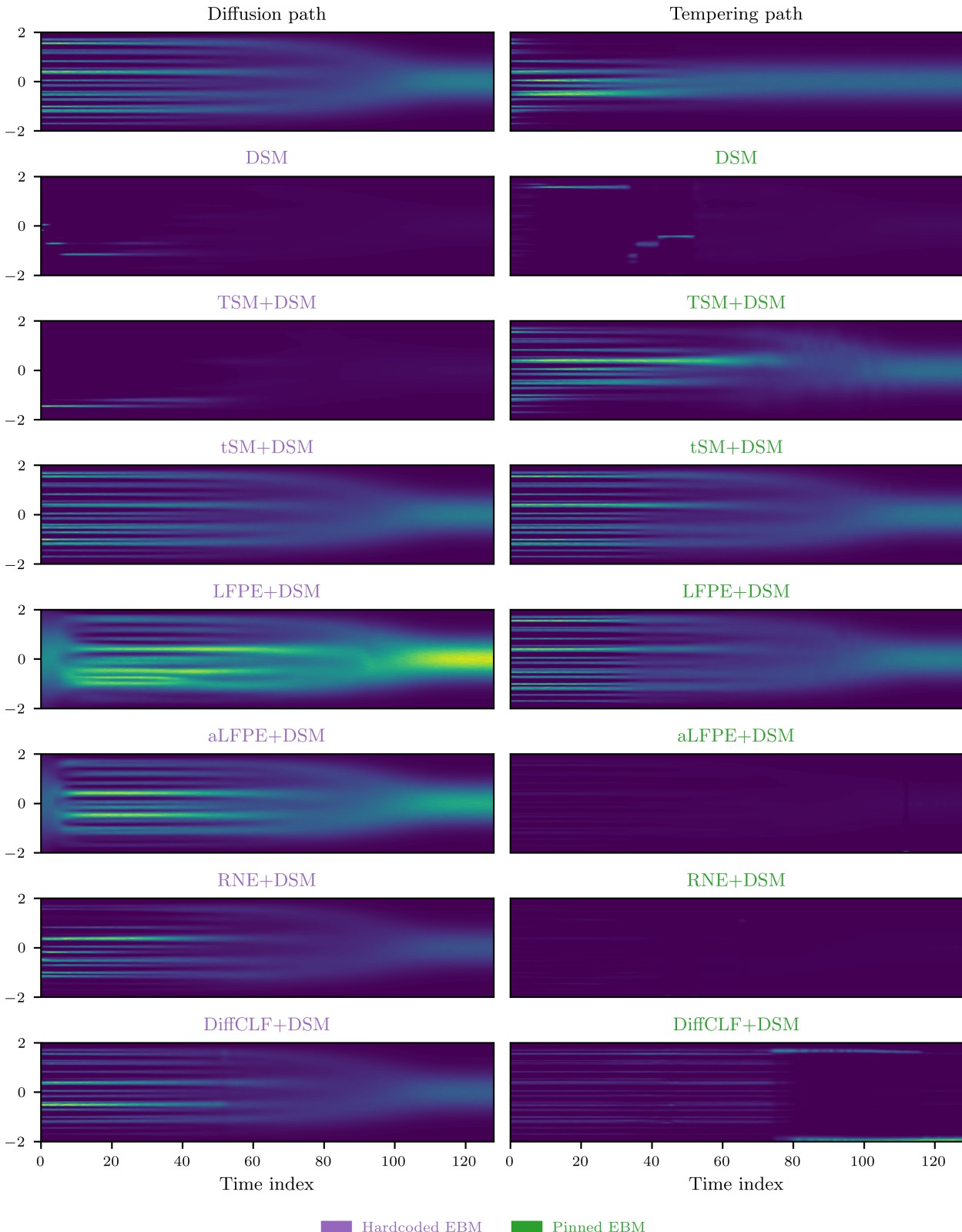

Figure 16: **Diffusion density paths bridging $\pi^{\mathbf{base}}$ (last time index) to the same *ManyModes* target as in Figure 6 (first time index). (First row)** the exact diffusion path is displayed on the left, the exact tempering path on the right, **(From second to last row)** we display the learned density path when using the DSM, TSM+DSM, tSM+DSM, LFPE+DSM, aLFPE+DSM, RNE+DSM or DiffCLF+DSM objective, with identical computational budget, **(Left)** use of hardcoded EBM, **(Right)** use of pinned EBM. This is complementary to Figure 7. Zoom in to get more details.

### D.4 Ablation studies on DM-based aMC methods

For clarity in the given ablation studies, we report sampling performance solely using the Sliced $W_2$ metric.

**$\Lambda$-optimal vs log-SNR discretization.** In idealized setting **(A)**, we compare the default log-SNR discretization with the $\Lambda$-optimal schedule originally developed for tempering paths, which we can pre-compute thanks to the tractability of our continuous-time diffusion path (see Appendix B.4). Figure 17 shows that the $\Lambda$-optimal schedule yields slight gains for the standard aMC baseline and first-order stochastic methods, but is comparable or worse on the methods identified by Figure 3 as the most effective under log-SNR with clear failures for all remaining AIS variants and second-order stochastic SMC, and similar performance in RE. The main conclusions therefore stand: deterministic methods and second-order stochastic kernels remain the most effective designs. Being target-independent and easy to compute, we conjecture log-SNR to be the most practical choice for general targets.

**EI vs DDPM parameterization for stochastic first-order kernels.** Previous diffusion-based BGs with first-order stochastic transition kernels relied on EI or EM discretizations (Phillips et al., 2024; Zhang et al., 2026); see Lemma 9 for an arbitrary noising schedule, Lemma 15 for the VP case and Lemma 21 for the VE case. In contrast, we use the DDPM kernel (9) in our implementation. This choice is motivated by the idealized experiments of Figure 18, where DDPM yields substantially better performance than EI across all aMC samplers. To our knowledge, none of the prior diffusion-based BG works rely on this kernel; we hope our results encourage its broader use.

**Base vs score-informed RE initialization.** Unlike sequential AIS/SMC, RE samplers using diffusion paths can warm-start each annealing level by simulating the reverse SDE from $\pi^{\mathrm{base}}$. Figure 19 ablates this *score-informed initialization* against the *base initialization* (independent samples from $\pi^{\mathrm{base}}$, used by default in the main experiments for fair comparison) on all *TwoModes* and *ManyModes* targets. Score-informed initialization does not improve over the base one for most variants; for first-order stochastic RE, the effect is inconsistent (beneficial on *TwoModes* but detrimental on *ManyModes*). This is consistent with the conclusions of Section 3.3: first-order stochastic transitions are not informative enough, a weakness visible for any initialization.

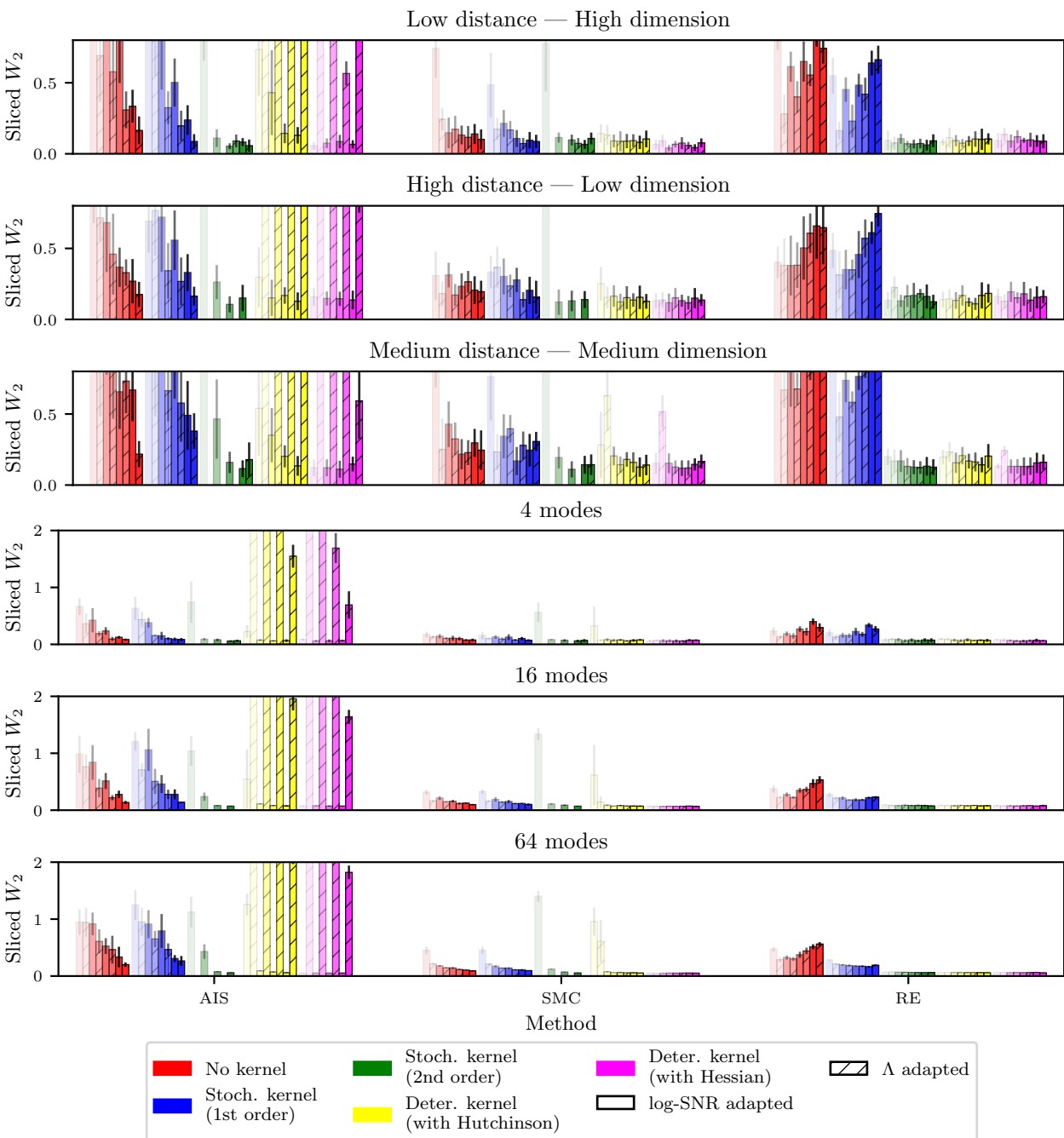

Figure 17: **DM-based aMC-BG results with annealed samplers based on log-SNR and Λ-optimal scheduling, using different mechanisms**, when targeting *TwoModes* **(Top three rows)** and *ManyModes* **(Bottom three rows)** distributions in idealized setting **(A)**. This figure follows the same visualization convention as Figure 3: each group of bars with the same color corresponds to a given aMC method. We additionally distinguish samplers using the Λ-optimal diffusion time discretization, shown with bar hatching, from those using the default log-SNR discretization, shown without hatching. The latter correspond exactly to the bars reported in Figure 3.

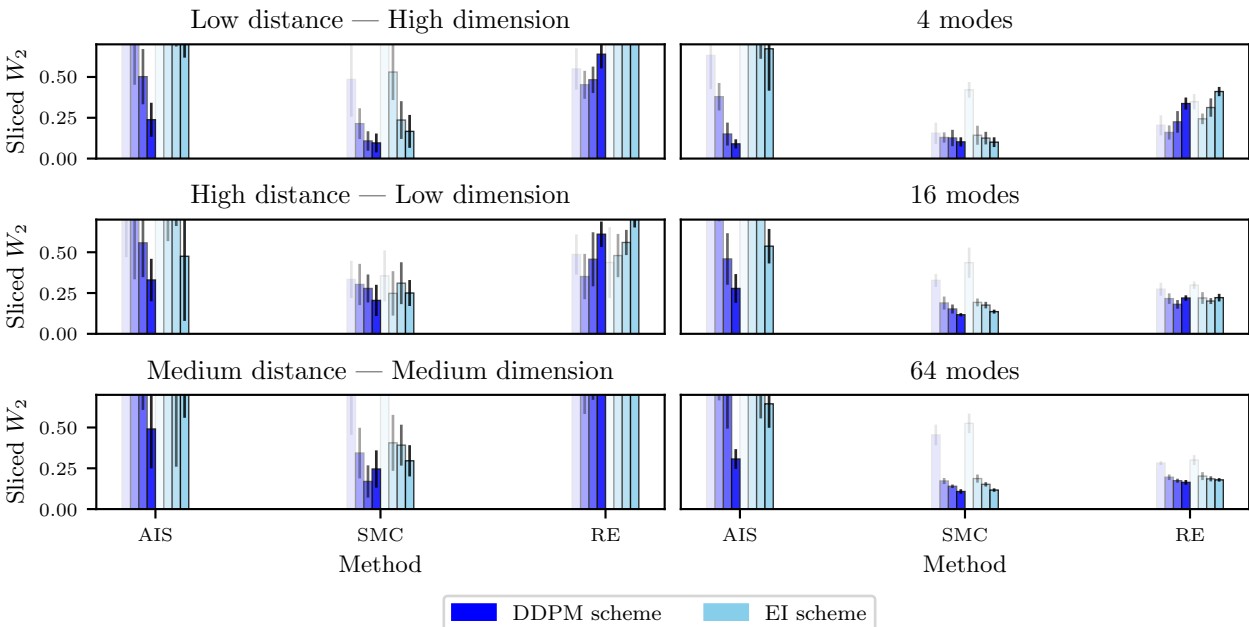

Figure 18: **Impact of the first-order stochastic denoising transition kernel in DM-based aMC-BGs**, when targeting *TwoModes* distributions **(left)** and *ManyModes* distributions **(right)** in idealized setting **(A)**. We compare two variants: one based on the DDPM scheme (9), used in our main experiments (blue bars, identical to those in Figure 3), and one based on the EI scheme (cyan), as proposed in prior work. Our results show that, for all aMC samplers and all values of $K$, the DDPM scheme substantially improves sampling performance over EI, especially for AIS and SMC.

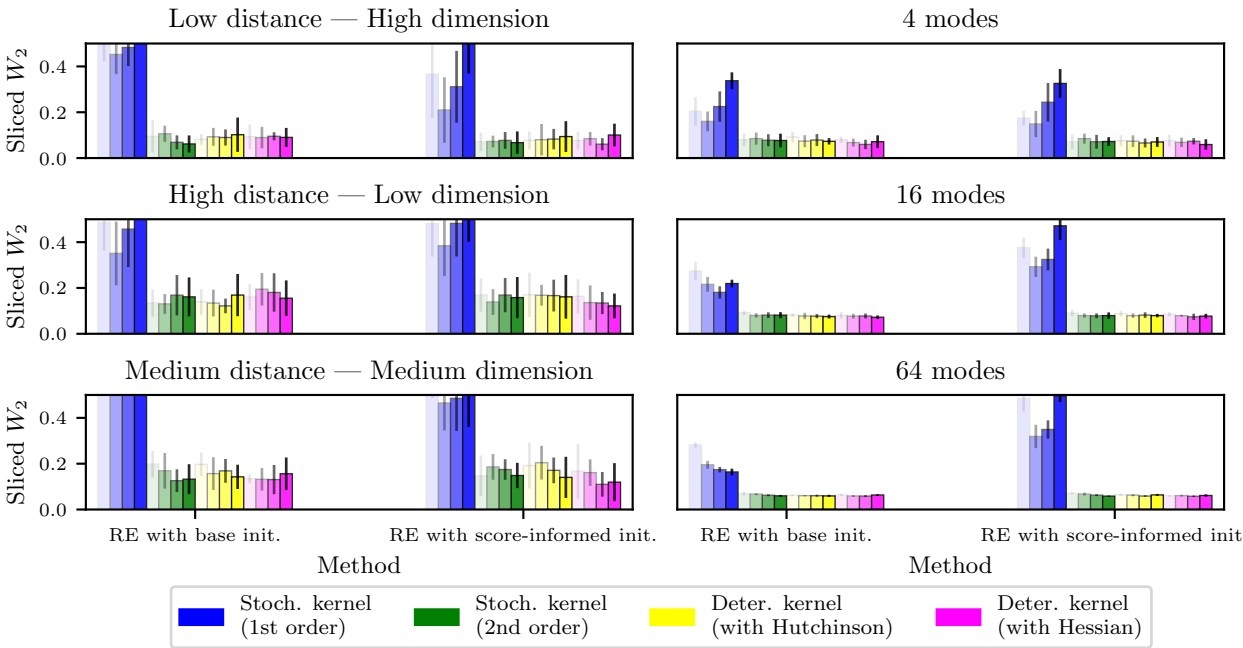

Figure 19: **Sensitivity of diffusion-based RE-BGs to per-level initialization**, when targeting *TwoModes* distributions **(left)** and *ManyModes* distributions **(right)** in idealized setting **(A)**. For each target distribution, we compare two RE initializations: "RE with score-informed init.", where each level is initialized by simulating the reverse SDE associated with the DM for the given value of $K$; and "RE with base init", where each level is initialized independently from the base distribution, as in tempering approaches. The latter is our default setting in the main experiments, and the corresponding bars coincide with the RE bars in Figure 3. Overall, the initialization choice has little effect on sampling performance, except for first-order stochastic transition kernels, where score-informed initialization degrades performance for challenging targets. This further highlights the limitations of this specific RE variant.

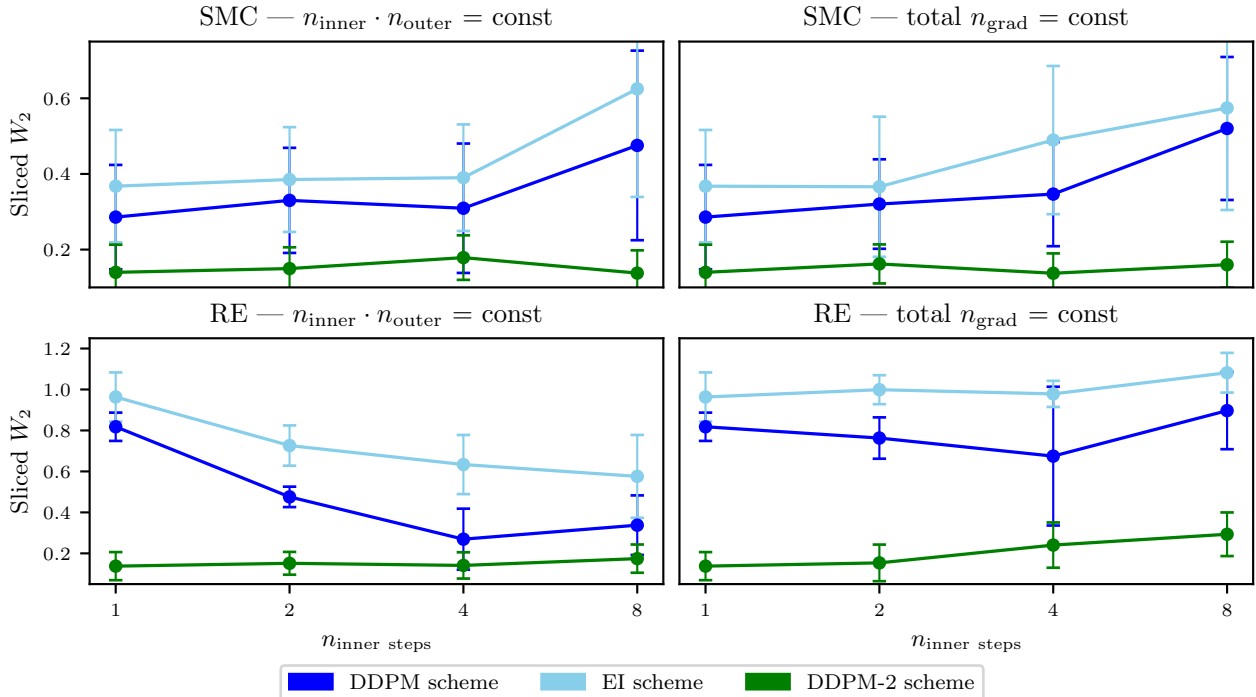

Figure 20: **Impact of multi-step stochastic transition kernels in DM-based SMC (top two rows) and RE (bottom two rows)**, on the intermediate *TwoModes* target in idealized setting **(A)**. The reported metric is the Sliced $W_2$ distance, with the same visualization convention as in Figures 3 and 18. We compare first-order methods based on the DDPM scheme (blue) and on the EI scheme (cyan), as well as second-order methods that additionally use the log-density Hessians (green). See the main text for the experimental setup ($n_{\text{inner}}$, $n_{\text{outer}}$, two budget regimes) and the conclusions.

**Effect of multi-step transition kernels in SMC and RE.** Following the diffusion-based RE design of Zhang et al. (2026), DM-based SMC and RE can be equipped with *multi-step* stochastic transition kernels, obtained by chaining $n_{\text{inner}} \geq 1$ single-step noising or denoising kernels into a single between-level transition. This construction is natural for DMs and applies in both the first- and second-order cases; the rest of the SMC and RE procedures is unchanged relative to our main implementation (which corresponds to $n_{\text{inner}} = 1$), up to the corresponding adaptations of the importance weights and Metropolis-Hastings acceptance probabilities. Each multi-step transition is, however, $n_{\text{inner}}$ times more expensive than a single-step one, so its net benefit is not obvious a priori.

In Figure 20, we study this effect on the intermediate *TwoModes* target in idealized setting **(A)**, for $n_{\text{inner}} \in \{1, 2, 4, 8\}$. To preserve the same underlying time grid as in our main experiments ($K = 128$ timesteps with $n_{\text{inner}} = 1$), we use $n_{\text{outer}} = 128/n_{\text{inner}}$ annealing levels in all cases, so that each multi-step transition simply spans $n_{\text{inner}}$ steps of that 128-step grid. We then consider two budget regimes:

• **(Left)** The per-level MCMC step counts (between adjacent levels in SMC, between swaps in RE) are kept fixed across $n_{\text{inner}}$. Since the number of annealing levels $n_{\text{outer}}$ shrinks as $n_{\text{inner}}$ grows, the total compute is not held constant.

• **(Right)** The per-level MCMC counts are adapted so that the total number of score evaluations stays constant across $n_{\text{inner}}$.

Increasing $n_{\text{inner}}$ leaves SMC unchanged or slightly degraded in both regimes, and helps only first-order RE in the Left regime. Once the comparison is rebalanced to equal budget (Right), even this benefit on first-order RE disappears, with multi-step transitions consistently degrading performance across all settings. Overall, these results support our default choice $n_{\text{inner}} = 1$, which additionally avoids the need to tune this hyperparameter. That said, we acknowledge that our Hutchinson-based deterministic transitions also require multiple gradient calls per transition, blurring the line between that regime and the multi-step setting studied here. Reassuringly, however, deterministic transitions uniformly improve all aMC samplers whereas multi-step transitions do not, suggesting that the gain does not stem from the extra compute alone.

**Hyperparameter sensitivity of deterministic approaches.** We assess the robustness of the deterministic diffusion-based aMC framework of Section 4 with respect to its three hyperparameters: the number of fixed-point iterations $n_{\text{iter}}$ for the Implicit Midpoint integrator (Proposition 2), the truncation order $n_{\text{trunc}}$ of the Jacobian log-determinant power series (Proposition 3), and the number of Hutchinson auxiliary variables $n_{\text{hutch}}$ used in the first-order variant (Appendix C.1). For each, we measure the error introduced in the relevant deterministic-aMC component, rather than its effect on final sampling performance, in order to isolate the approximation.

*(i) Fixed-point convergence ($n_{iter}$).* We numerically verify the geometric convergence guaranteed by Proposition 2 on all *TwoModes* and *ManyModes* targets, by measuring the error in the mutual invertibility condition (20) across all timestep pairs $(s, t)$ at $K = 128$, for $n_{\text{iter}} \in [2, 32]$. Across all targets and both VP and VE schedules (Figures 21 and 22), the error reaches numerical precision ($\sim 10^{-8}$) for $n_{\text{iter}} \geq 6$, supporting our default $n_{\text{iter}} = 4$.

*(ii) Sanity check on the penalty correction ($n_{trunc}$, $n_{hutch}$).* For AIS/SMC, the penalty correction (Appendix C.1) is designed to produce an unbiased estimator of the IS weight, so reporting its bias and variance against the deterministic ground truth is a meaningful check. For RE, however, the correction is designed to preserve $\bar{\pi}$-invariance of the MH kernel rather than to make the acceptance probability itself unbiased. The small gap to the deterministic acceptance is therefore not a defect to chase to zero, but simply a numerical witness that the stochastic kernel stays close to its deterministic counterpart. With this caveat in mind, we compare the first-order penalty-corrected estimator (Hutchinson + Bessel) to the second-order deterministic value (using the exact Hessian) on neighboring annealing levels at $K = 128$ in the VP setting. The two agree to within a few percent on both IS weights and MH acceptance probabilities, well within the precision relevant for sampling. Varying $n_{\text{trunc}}$ has virtually no effect (Figures 23 and 24), supporting our default $n_{\text{trunc}} = 3$, while $n_{\text{hutch}}$ reduces both bias and variance only gradually (Figures 25 and 26); since Jacobian–vector products are memory-bound, we set $n_{\text{hutch}} = 39$ as the largest feasible value in our setups.

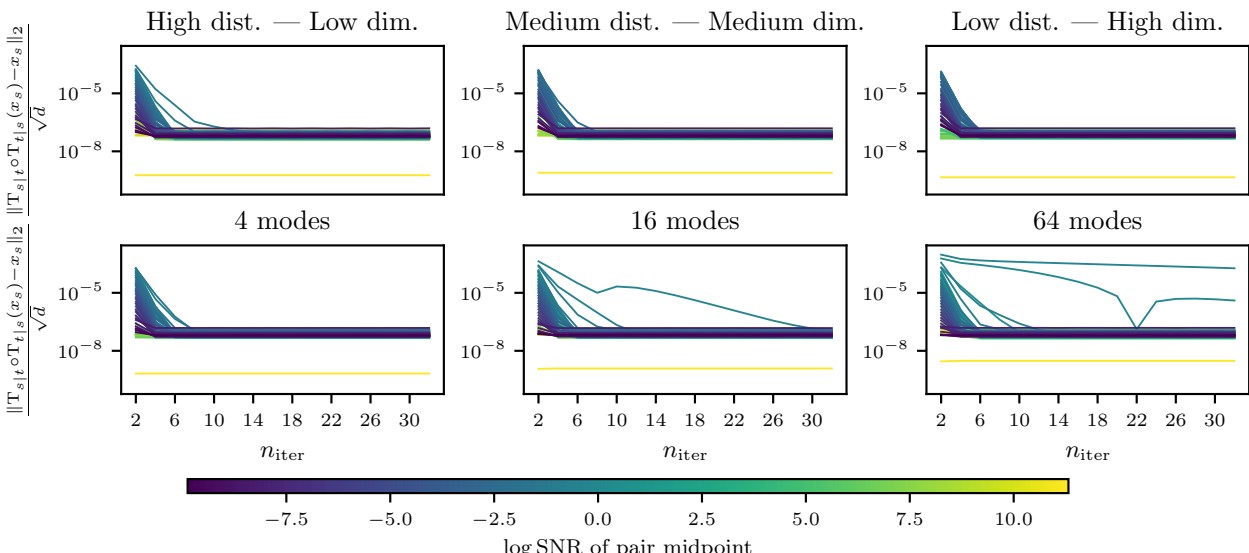

Figure 21: **Validity of the mutual invertibility condition for the Implicit Midpoint transition maps defined by the VP noising schedule**, evaluated on all *TwoModes* and *ManyModes* targets with $K = 128$. For each consecutive pair of discretization times $(s, t)$, with $s < t$, we report the average mutual invertibility error $\mathbb{E}[\|T_{s|t} \circ T_{t|s}(X_s) - X_s\|]$, rescaled by $d^{-1/2}$ to enable comparison across dimensions. Curves are colored according to the midpoint time $(s + t)/2$ in log-SNR space. Here, $T_{s|t}$ denotes the backward IM map and $T_{t|s}$ the forward IM map. We vary the number of fixed-point iterations $n_{\text{iter}}$ from 2 to 32, with $n_{\text{iter}} = 4$ used in our main experiments. Across all midpoint times and targets, the error decreases rapidly with $n_{\text{iter}}$ and shows little to no further improvement beyond $n_{\text{iter}} \geq 6$.

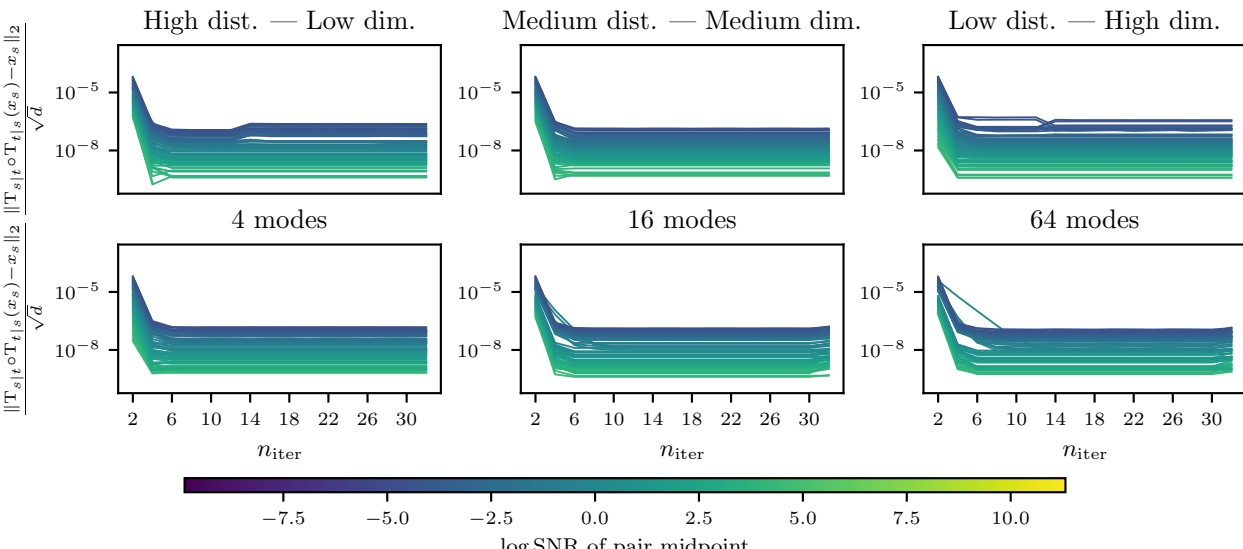

Figure 22: **Validity of the mutual invertibility condition for the Implicit Midpoint transition maps defined by the VE noising schedule**, evaluated on all *TwoModes* and *ManyModes* targets with $K = 128$. The experimental setting and visualization convention is the same as in Figure 21, with the same conclusions.

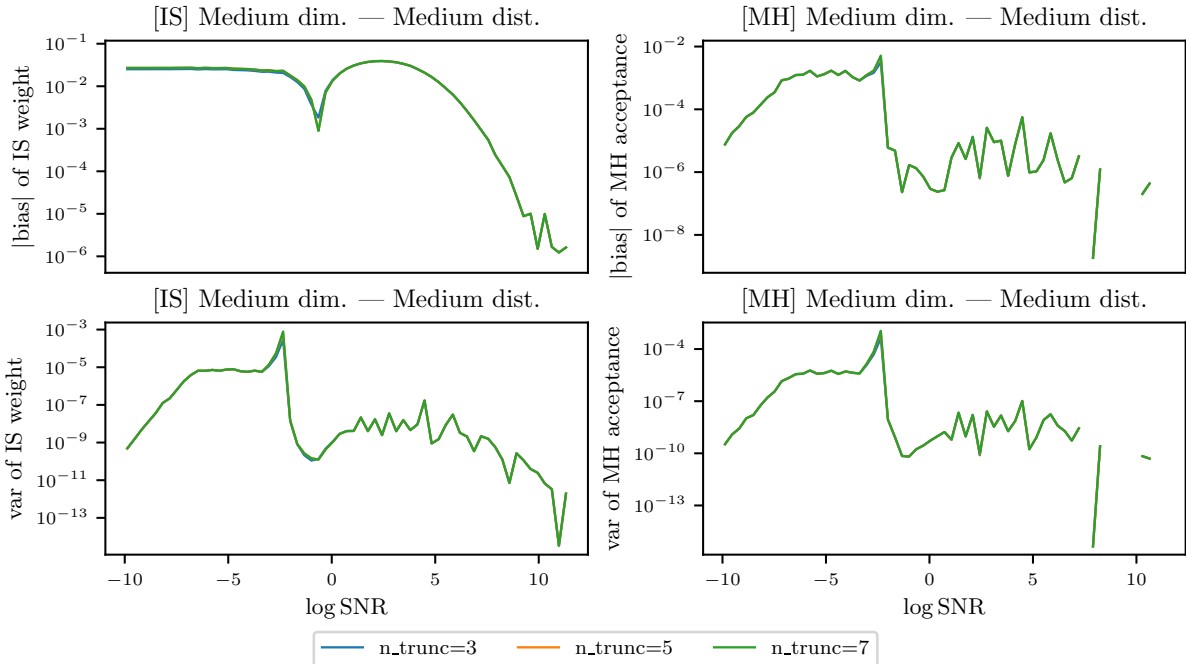

Figure 23: **Accuracy of the Hutchinson-based statistical estimates with respect to the truncation order $n_{\mathbf{trunc}}$** along the VP density path, evaluated on the intermediate *TwoModes* target with $K = 128$. **(Left):** IS weight estimation. **(Right)**: MH rate estimation. For each consecutive timestep pair $(s,t)$ induced by the discretization, displayed in log-SNR space, we report both the bias **(top)** and variance **(bottom)** of the estimator. We fix the number of Hutchinson auxiliary variables to 32 and the number of fixed-point iterations to 4, and vary the truncation order $n_{\mathrm{trunc}} \in \{3, 5, 7\}$, with $n_{\mathrm{trunc}} = 3$ used in our main experiments. Increasing $n_{\mathrm{trunc}}$ does not noticeably affect performance.

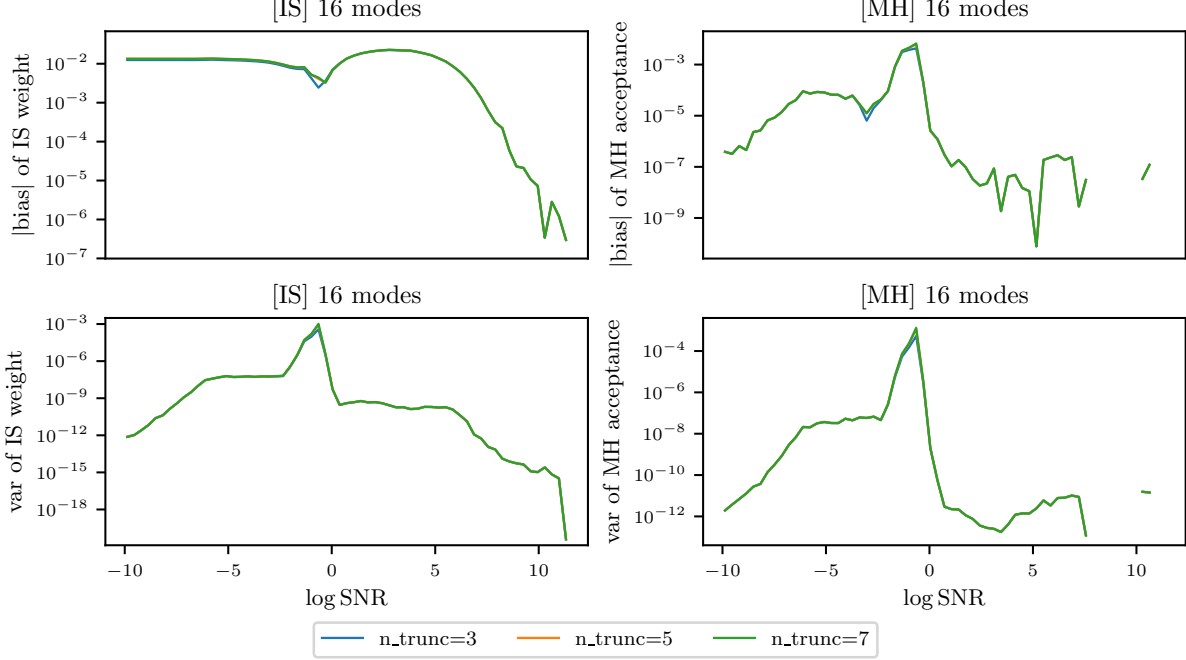

Figure 24: **Accuracy of the Hutchinson-based statistical estimates with respect to the truncation order $n_{\mathbf{trunc}}$** along the VP density path, evaluated on the intermediate *ManyModes* target with $K = 128$. The experimental setting and visualization convention is the same as in Figure 23, with the same conclusions.

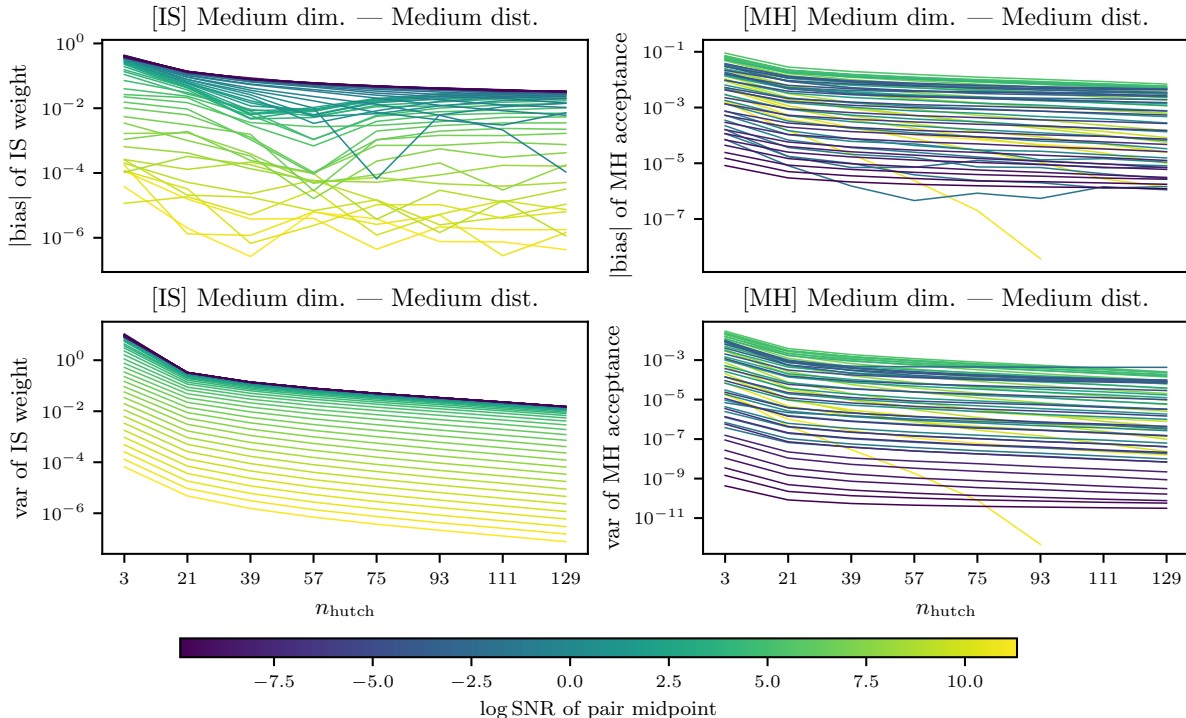

Figure 25: **Accuracy of the Hutchinson-based statistical estimates with respect to the number of auxiliary variables** $n_{\text{hutch}}$ along the VP density path, evaluated on the intermediate *TwoModes* target with $K = 128$. **(Left):** IS weight estimation. **(Right):** MH rate estimation. For each consecutive pair of discretization times $(s, t)$, with $s < t$, we report the estimator bias **(top)** and variance **(bottom)**. Curves are colored according to the midpoint time $(s + t)/2$ in log-SNR space. We fix $n_{\text{iter}} = 4$ and $n_{\text{trunc}} = 3$, and vary $n_{\text{hutch}}$ from 3 to 129 (multipliers of 3 due to our Hutch++-based formulation), with $n_{\text{hutch}} = 39$ used in our main experiments.

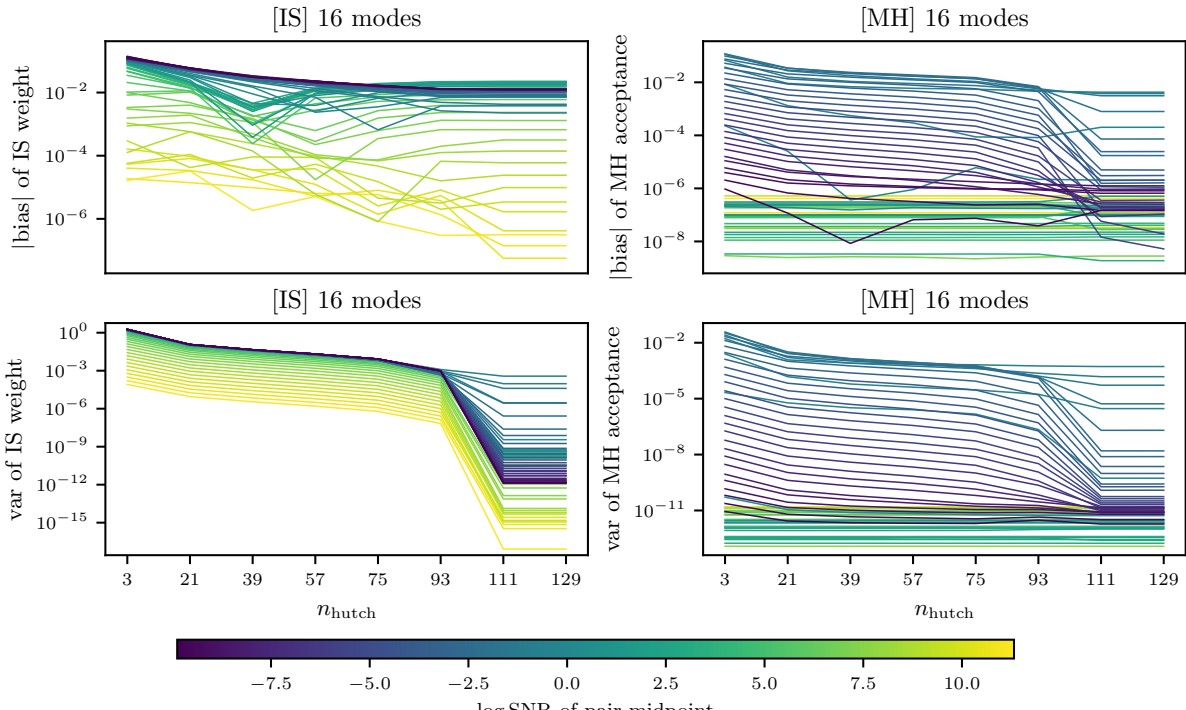

Figure 26: **Accuracy of the Hutchinson-based statistical estimates with respect to the number of auxiliary variables** $n_{\text{hutch}}$ along the VP density path, evaluated on the intermediate *ManyModes* target with $K = 128$. The experimental setting and visualization convention is the same as in Figure 25, with the same conclusions.

