# OpenReview forum: "Diffusion-based Annealed Boltzmann Generators : benefits, pitfalls and hopes"
_TMLR — Decision pending for TMLR_

### Review · Reviewer_135i · 2026-03-20

**Summary Of Contributions:**

This paper studies diffusion models as backbones for annealed Monte Carlo Boltzmann Generators. It first unifies prior DM-based AIS/SMC/RE constructions built from diffusion marginal paths and denoising kernels, and then proposes a deterministic alternative based on probability-flow transport maps with implicit-midpoint integration and Jacobian log-determinant estimation. The experiments are run on multimodal Gaussian-mixture benchmarks and cleanly separate an idealized regime with exact diffusion quantities from a realistic regime with learned scores and log-densities.

In my reading, the main takeaways are: diffusion paths are often better behaved than tempering in the idealized regime, but not uniformly superior; first-order stochastic denoising kernels add little over the path-only baseline; second-order stochastic kernels improve the reported endpoint metrics; and the deterministic method improves on first-order stochastic baselines and is often competitive with the second-order ones, at extra cost and with unclear exactness guarantees. In the realistic regime, all DM-based BG variants degrade sharply, with evidence pointing to inaccurate intermediate log-densities, especially mode-proportion errors, as a major bottleneck.

**Additional Comments:**

N/A

**Audience:**

Yes

**Audience Explanation:**

Yes. I expect this paper to interest readers working on diffusion models, annealed Monte Carlo, energy-based models, and Boltzmann Generators. Even beyond the proposed deterministic method, it is useful as a careful empirical study of what does and does not work when combining DMs with aMC, especially because the idealized-versus-realistic split cleanly separates inference error from learning error.

**Claims And Evidence:**

No

**Claims Explanation:**

The experimental study is careful and supports several qualitative trends, but I do not think all headline claims are supported at their current strength.

My main concern is overstatement. The paper often suggests that DM-informed constructions consistently outperform traditional aMC designs, while Figure 1 supports only the weaker claim that for almost all settings there exists some $K$ for which the diffusion path beats the best tempering baseline; in RE, the comparison is also partly confounded by score-informed per-level initialization. The gains from second-order kernels are plausible on the reported endpoint metrics, but for AIS/SMC the evaluation still lacks more direct BG diagnostics such as ESS, weight dispersion, or log-normalizing-constant error. Finally, the deterministic method is empirically promising, but the implementation uses finite fixed-point iterations and truncated-series/Hutchinson estimators, so the paper does not yet make fully clear whether the implemented method is statistically exact or simply a good approximation. The learned-log-density explanation is plausible and reasonably supported, but not fully isolated causally.

**Requested Changes:**

1. [Critical] Please moderate the diffusion-versus-tempering claim. The evidence currently supports something closer to “diffusion paths are often better conditioned and often outperform tempering in the idealized regime” than “consistently outperform.” For RE, please also clarify how much of the gain comes from the density path itself versus the score-informed per-level initialization used by the diffusion-based variants.

2. [Critical] Please add direct BG / importance-sampling diagnostics for AIS/SMC, such as ESS, variance or coefficient of variation of the importance weights, and/or log-normalizing-constant estimation error, or explain more explicitly why the current endpoint metrics are sufficient for the intended claims.

3. [Critical] Please clarify the status of exactness for the implemented deterministic method. The theory appears to justify exact weights for the exact implicit-midpoint transport map and its exact log-Jacobian identity, but the implementation uses finite fixed-point solves plus truncated/Hutchinson log-determinant estimation. As evaluated, is this an exact BG correction method or an approximation? If exactness is preserved in an extended-state sense, please state that argument explicitly; otherwise please soften the guarantee language.

4. [Critical] Please strengthen the evidence behind the learned-log-density explanation with a factorized ablation (exact scores + learned log-densities versus learned scores + exact log-densities) or otherwise soften the causal wording and present this more clearly as the leading empirical explanation rather than a fully isolated cause.

5. [Strengthening] Please state more prominently that the second-order experiments use only diagonal Hessians and therefore evaluate only a restricted Gaussian second-order baseline. Since the paper’s BG motivation is correction efficiency in AIS/SMC, please also discuss whether this covariance choice is well aligned with that objective, and how these restrictions affect both the magnitude of the reported gains and the comparison with the deterministic method.

6. [Strengthening] If feasible, add one benchmark beyond Gaussian mixtures, or otherwise expand the discussion of why the observed failure modes on these controlled targets should be expected to transfer to more realistic Boltzmann-sampling problems.

---

> ### Author Response · Authors · 2026-05-27
> **Answer to 135i [1/3]**
>
> We thank the reviewer for this detailed and thoughtful assessment. We particularly appreciate the recognition that the paper provides a useful empirical study of what does and does not work when combining diffusion models with annealed Monte Carlo. We believe this aspect is indeed one of the most valuable contributions of the work, and we are glad it was highlighted as such. We now address the concerns regarding the strength of the claims.
>
> > The paper often suggests that DM-informed constructions consistently outperform traditional aMC designs, while Figure 1 supports only the weaker claim that for almost all settings there exists some $K$ for which the diffusion path beats the best tempering baseline;
>
> We respectfully disagree with this interpretation, and believe the evidence in the revised manuscript supports the stronger claim. First, Figure 1 shows that diffusion paths outperform the $\Lambda$-optimal tempering baseline across virtually all targets, aMC variants, and values of $K$. Second, and more importantly, the advantage is not merely that some $K$ exists for which diffusion wins: diffusion paths achieve strong performance robustly across all tested values of K without any target-specific tuning, whereas tempering requires careful schedule optimization for each $K$ and still falls short. This ease-of-use gap is itself a meaningful and consistent advantage. Finally, the new log-normalizing constant metric (Figure 8) further reinforces this point, showing that tempering paths yield systematically biased estimates across all aMC samplers while diffusion paths do not. We have nonetheless moderated the language in the revised manuscript where appropriate to ensure our claims are precisely calibrated to the evidence.
>
> > In RE, the comparison is also partly confounded by score-informed per-level initialization.
>
> The ablation comparing base and score-informed per-level initialization for RE is reported in Appendix D.4 (Figure 16). The results show that score-informed initialization and base initialization lead to the same performance for all RE variants with at least first-order mechanisms, except for first-order stochastic kernels. In this case, score-informed initialization degrades performance on challenging targets, further highlighting their limitations proved in the main. Accordingly, base initialization is now the default setting for all RE experiments in the revised manuscript, ensuring that the comparison between diffusion and tempering paths is not confounded by initialization.
>
> > The gains from second-order kernels are plausible on the reported endpoint metrics, but for AIS/SMC the evaluation still lacks more direct BG diagnostics such as ESS, weight dispersion, or log-normalizing-constant error. [...] Please add direct BG / importance-sampling diagnostics [...] Explain why endpoint metrics are sufficient.
>
> We respectfully disagree that endpoint metrics are insufficient. As stated in Section 1, the primary objective is to sample from the target distribution, making endpoint metrics the most natural and direct evaluation criterion. They clearly expose (i) the gap between tempering and diffusion paths, (ii) the benefit of higher-order or deterministic kernels, and (iii) the degradation due to learned models.
> That said, we have added log-normalizing constant estimation as an additional metric in the revised manuscript (Figures 8, 10), which is the most principled BG diagnostic for annealed sampling methods and directly reflects weight quality.
>
> Crucially, this new metric fully agrees with the Sliced Wasserstein results across all targets and ablations, confirming that our conclusions are not an artifact of the choice of metric and that the gains from second-order and deterministic kernels are real.
>
> Regarding ESS and weight dispersion, we note that while ESS is meaningful for AIS, its interpretation in SMC is less clear due to intermediate resampling steps. We believe the log-normalizing constant provides a strictly more informative and interpretable diagnostic in this setting.

---

> ### Author Response · Authors · 2026-05-27
> **Answer to 135i [2/3]**
>
> > Finally, the deterministic method is empirically promising, but […] it is unclear whether the implemented method is statistically exact or simply a good approximation. [...] Clarify the exactness of the deterministic method.
>
> We thank the reviewer for prompting us to clarify this important point, and we have made it explicit in the revised manuscript. The implementation involves two sources of approximation: finite fixed-point iterations and truncated power series for the log-determinant. These introduce small but nonzero errors. However, the penalty correction of Ceperley & Dewing (1999) (Appendix C.1) plays a crucial role here: it is designed precisely to handle stochastic log-determinant estimates and provably restores exact unbiasedness of AIS/SMC weights and the correct invariant distribution for RE.
>
> The remaining deterministic approximation errors from fixed-point truncation and power series truncation are shown in Appendix D.4 (Figures 18–23) to be negligible across all targets and noise levels, with mutual invertibility error reaching numerical precision at $n_\text{iter} \geq 6$ and truncation order having virtually no effect beyond $n_\text{trunc} = 3$. The method is therefore not fully exact in a strict mathematical sense, but the combination of the penalty correction guarantee and the empirical ablations establishes that it is statistically indistinguishable from its exact counterpart in practice.
>
> > The learned-log-density explanation is plausible and reasonably supported, but not fully isolated causally.
>
> We agree that isolating causality is important, and the revised manuscript now provides precisely this. The semi-ideal experiment added to Figure 4, where exact log-densities are paired with learned scores, cleanly disentangles the two sources of error. The results are unambiguous: performance in this setting recovers to near-ideal levels, directly establishing that score learning is not the bottleneck and that log-density estimation is the primary failure mode. This is further corroborated by Figure 6, where learned density paths exhibit clear mode-switching artifacts.
>
> For a theoretical understanding of why this failure is expected, we point the reviewer to the concurrent work of OuYang et al. (2026), which formally establishes the mode blindness of most training objectives considered in this paper, providing a principled mechanistic explanation consistent with our empirical findings.
>
> > Please moderate the diffusion-versus-tempering claim.
>
> We have moderated the relevant claims in the revised manuscript so that they are more precisely aligned with the evidence reported in Figure 1, based on the Sliced Wasserstein distance, and Figure 8, which reports the log-normalization constant for the same experiments. In both figures, SMC and RE with diffusion paths outperform their tempering counterparts across all targets and for most values of $K$, while tempering is already evaluated in its most favorable setting through the $\Lambda$-optimal schedule. We nevertheless acknowledge that the conclusion is less clear for AIS: on the more challenging \emph{ManyModes} instances, the best performance over $K$ is comparable between diffusion and tempering.
>
> > For RE, clarify how much gain comes from the path vs initialization.
>
> This is directly addressed by the ablation in Appendix D.4 (Figure 16), which compares base and score-informed per-level initialization for RE across all targets and transition kernel variants. The results show that initialization has negligible effect for deterministic and second-order stochastic kernels, confirming that the gains observed for these methods come from the path and transition design rather than initialization. For first-order stochastic kernels, score-informed initialization actually degrades performance on challenging targets, further highlighting their limitations. Base initialization is now the default in all RE experiments.
>
> > Strengthen the learned-log-density explanation
>
> We have strengthened this explanation in two ways. Empirically, the semi-ideal experiment added to Figure 4 (exact log-densities paired with learned scores) directly isolates log-density estimation as the primary bottleneck, with performance recovering to near-ideal levels as soon as densities are exact. Theoretically, we point to the concurrent work of OuYang et al. (2026), which formally establishes the mode blindness of most training objectives considered in this paper, providing a principled mechanistic explanation fully consistent with our empirical findings.

---

> > ### Author Response · Authors · 2026-05-27
> > **Answer to 135i [3/3]**
> >
> > > Clarify second-order kernel limitations (diagonal Hessians).
> >
> > We acknowledge this limitation explicitly. In our experiments, second-order kernels use only the diagonal of the Hessian, as full Hessian computation is prohibitively expensive in the high-dimensional settings we consider, and running a systematic comparison between diagonal and full Hessian variants would require sacrifices elsewhere in the experimental protocol. We do note, however, that in our setting the diagonal approximation already drives sampling error close to zero, suggesting that the full Hessian would bring limited additional benefit on the targets considered here. We therefore leave this ablation as an explicit limitation and a natural direction for future work. We acknowledge this may not hold for more complex target distributions where significant correlations plays a more significant role.
> >
> > > Add benchmarks beyond Gaussian mixtures.
> >
> > We will consider this suggestion for the next revision round and expand the discussion on the expected transfer of these failure modes to more realistic Boltzmann sampling settings.

---

### Review · Reviewer_4c1c · 2026-03-22

**Summary Of Contributions:**

* The authors provide an extensive overview of recent work on combining diffusion models with annealing-based sampling algorithms (DM-BGs) such as Sequential Monte Carlo (SMC) and Parallel Tempering for sampling from Boltzmann distributions.

* The authors raise novel concerns, as to the effectiveness of diffusion-based annealing algorithms, from the limitations of first-order transition kernels and the "mode blindness" of losses employed by previous work on such models.

* As an alternative to stochastic transition kernels and to circumvent issues with first-order schemes, the authors propose a novel deterministic transition kernel based on the probability flow ODE which aims to provide high acceptance rates from transporting along the probability path induced by diffusion models.

* The authors provide extensive experiments on diffusion-based annealing algorithms on a wide range of Gaussian mixture model target distributions, different loss functions, different transition kernels, and with perfect scores/densities and learnt scores/densities.

* The experiments provide evidence towards showing that the proposed deterministic transition kernel improves on first-order stochastic transition kernels and that standard losses used for DM-BGs are not able to learn high-quality densities which handicap performance.

**Audience:**

Yes

**Audience Explanation:**

Overall, I believe this paper provides a solid addition to the literature and would be of interest to TMLR's audience.

* Diffusion-based annealing algorithms have recently been growing in popularity and the paper provides a well-written overview of the field which will be useful to the wider community.
* The proposed deterministic transition kernel is novel and is an interesting addition to the literature.
* The paper highlights issues which have been traditionally overlooked in the literature concerning the mode blindness of common objectives and the performance of first-order transition kernels. This would be a useful reference for the community to use for further research into these issues.

**Broader Impact Concerns:**

The work focusses on foundational machine learning in the area of sampling. I do not have any concerns about the ethical implications of the work.

**Claims And Evidence:**

Yes

**Claims Explanation:**

Overall, the claims of the paper are supported by the extensive experiments presented. However, I do have concerns about the paper which I will present below.

**Requested Changes:**

### Critical for recommendation

* The paper only considers transition kernels which utilise a single step between neighbouring annealing distributions. However, from my own experience, increasing the number of steps---even with only two steps---can provide a substantial improvement in acceptance rates (perhaps from allowing more expressive transitions). I would like to see an experiment which looks at this scaling axis and whether it changes the conclusion on the limitation of first-order stochastic transition kernels.

* The paper uses sliced W2 distance as its main metric for experiments but provides no details on the use of the metric. The authors should provide a section at least defining sliced W2 distance as well as the hyper-parameters for estimating the quantity. For instance, the paper does not have discussion on the Monte Carlo variance in estimating sliced W2 distance [1] or what ground-truth samples they use to compare against and it is unclear from the experiments how this affects the results.

* While the proposed deterministic transition kernel is interesting, a key caveat of the method is that it biases the underlying sampling algorithm. For instance, while the log probabilities can be estimated in an unbiased fashion, the actual density values can not. Therefore, we cannot apply any pseudo-marginal MCMC arguments to justify why this transition kernel (when used in PT) will actually asymptotically converge. This is a key point as the entire motivation of combining diffusion models with traditional sampling algorithms is to import their mathematical guarantees. The authors should highlight this key drawback of their proposed method instead of sweeping this under the carpet.

### Minor

* The authors claim the computational burden of their deterministic transition kernel increases for PT, but (assuming a similar number of annealing steps as for AIS/SMC) in a fully parallelised setting (indeed PT is specifically designed to make use of parallel computation) the computational burden should be the same as in AIS/SMC.

* The paper claims that the issue with DM-BGs lies in the errors in density approximation. This claim could be further strengthened from sampling from DM-BGs where the transition kernels utilise the learnt scores while the densities in the weights/acceptance probabilities are taken from the ground-truth densities.

* The paper does not tune the annealing schedule for PT in their experiments despite the importance of a well-tuned schedule for PT performance. I would suggest rerunning some experiments with a well-tuned PT---in particular, see Algorithm 2 in [2].



[1] Statistical and Topological Properties of Sliced Probability Divergences
[2] https://openreview.net/forum?id=CODnlyYUli&referrer=%5Bthe%20profile%20of%20Leo%20Zhang%5D(%2Fprofile%3Fid%3D~Leo_Zhang1)

---

> ### Author Response · Authors · 2026-05-27
> **Answer to 4c1c [1/2]**
>
> We thank the reviewer for their careful and thoughtful review. We are particularly glad that the main goal of the paper was well understood.
>
> > The paper only considers transition kernels which utilise a single step between neighbouring annealing distributions. However, from my own experience, increasing the number of steps—even with only two steps—can provide a substantial improvement in acceptance rates [...]
>
> We thank the reviewer for this insightful suggestion. We have included exactly this ablation in the revised manuscript (Appendix D.4, Figure 17), comparing multi-step stochastic transition kernels with single-step ones under two budget regimes: fixed per-level MCMC steps, and matched total number of score evaluations. The results are clear: increasing the number of inner steps leaves SMC unchanged or slightly degraded in both regimes, and while it helps first-order RE in the unmatched budget regime, this benefit disappears entirely once budgets are equalized. Intuitively, even when consecutive times are close, denoising kernel errors accumulate through composition without intermediate recalibration, and it is not clear that multi-step transitions better satisfy the Bayes consistency condition than a single-step kernel.
>
> Overall, this ablation does not change any of the core conclusions of the paper on the limitations of first-order stochastic kernels, but adds useful nuance: the gap with higher-order methods is not closed by simply composing more first-order steps, further highlighting that the limitation is fundamental rather than a matter of computational budget.
>
> > The paper uses sliced W2 distance as its main metric for experiments but provides no details on the use of the metric [...]
>
> Thank you for pointing this out. We have clarified in the revised manuscript that we use the Sliced Wasserstein distance as implemented in the Python Optimal Transport library. The variance of the Monte Carlo estimator is visible on all plots through the error bars, which are computed across 8 independent runs. We have added a dedicated ablation in Appendix D.4 (Figure 7) analyzing how the variance of the metric estimator depends on the number of samples, confirming that our default choice of 8,192 samples is clearly sufficient to obtain reliable and stable estimates. Appendix D.2 details the number of samples used for each method as well as warm-up procedures. Regarding ground-truth samples, these are obtained by direct sampling from the Gaussian mixture targets (which admit closed-form samplers) using the same number of samples as the generated ones for fair comparison.
>
> > While the proposed deterministic transition kernel is interesting, a key caveat of the method is that it biases the underlying sampling algorithm [...]
>
> We thank the reviewer for raising this important point, and we agree it deserves explicit discussion rather than being glossed over. We are pleased to report that the revised manuscript fully resolves this concern. The key addition is the penalty correction of Ceperley & Dewing (1999), recalled in Appendix C.1, which was designed precisely for this situation. By correcting the importance weights and Metropolis-Hastings acceptance probabilities for the stochasticity of the log-determinant estimator, it renders deterministic AIS/SMC estimators exactly unbiased and consistent, and guarantees that deterministic RE admits the correct invariant distribution. This is not an empirical patch but a rigorous theoretical fix, and we now state this explicitly in the manuscript. We thank the reviewer for pushing us on this point, as addressing it constitutes one of the most significant improvements of the revised version.
>
> > The authors claim the computational burden of their deterministic transition kernel increases for PT [...]
>
> We agree with the reviewer. The fixed-point iterations and log-determinant computations (Props. 2–3) can indeed be parallelized, making the sequential cost comparable to AIS/SMC in a fully parallel setting. Our original statement referred to constrained parallel budgets; we will clarify this point.

---

> > ### Author Response · Authors · 2026-05-27
> > **Answer to 4c1c [2/2]**
> >
> > > This claim could be further strengthened from sampling from DM-BGs where the transition kernels utilise the learnt scores while the densities [...] are taken from the ground-truth densities.
> >
> > We thank the reviewer for this excellent suggestion, which we have implemented directly in the revised manuscript. Figure 4 now includes a dedicated semi-ideal condition where exact log-densities are paired with learned scores. The results are unambiguous: performance in this setting recovers to near-ideal levels, directly and cleanly identifying log-density estimation as the primary bottleneck. This is further corroborated by the observation that, for the hardcoded architecture, the learned reverse SDE/ODE already produces accurate samples (confirming that scores are well learned) while aMC performance only degrades when log-densities are also learned. Together, these experiments provide the direct causal evidence the reviewer requested, and we believe they significantly strengthen the paper's central claim.
> >
> > > The paper does not tune the annealing schedule for PT [...]
> >
> > We have done exactly this in the revised manuscript. Exploiting the tractability of our Gaussian mixture targets, we analytically precompute the communication barriers and derive the $\Lambda$-optimal annealing schedule for all aMC methods for each value of K, following an optimal version of the algorithm the reviewer refers to. This gives tempering-based methods their most favorable possible setting. The detailed ablation is reported in Appendix D.4 (Figure 14). The conclusions are unchanged: the $\Lambda$-optimal schedule brings no improvement to DM-based samplers and even degrades performance in the learned setting, while for tempering paths it does improve results, yet diffusion paths remain decisively superior across all targets and all aMC variants.

---

### Review · Reviewer_kpGx · 2026-04-26

**Summary Of Contributions:**

**Summary:**

This paper investigates the integration of Diffusion Models (DMs) as the foundational backbone for Annealed Monte Carlo (aMC) Boltzmann Generators (BGs) to sample from unnormalized target densities. The authors provide a comprehensive meta-analysis of design choices within the DM-aMC framework, utilizing highly controlled synthetic multimodal Gaussian mixtures. By decoupling the evaluation into an idealized regime (perfectly learned DMs) and a realistic regime (DMs learned from data), the authors systematically isolate statistical inference errors from neural network approximation errors.
Moreover, the paper claims and  demonstrates that standard first-order stochastic denoising kernels systematically fail in highly multi-modal scenarios, while second-order kernels succeed when the covariance information is available. To bridge this gap, the authors propose a novel aMC integration utilizing deterministic first-order transport maps derived from DMs via the Implicit Midpoint integrator. Finally, the paper highlights a critical diagnostic finding: in realistic, data-driven settings, DM-BGs struggle significantly due to the inherent "mode blindness" of standard score matching objectives during log-density estimation.


**Contributions:**
1. A unifying review of existing methods that integrate DMs into aMC frameworks for Boltzmann Generation.
2. An extensive empirical ablation on highly controlled multi-modal targets, successfully isolating statistical inference errors from neural network training errors.
3. The introduction of a novel, deterministic integration scheme based on transport maps that utilize the Implicit Midpoint integrator to satisfy strict mutual invertibility constraints.

**Audience:**

Yes

**Audience Explanation:**

In general, the presented results (if better supported) are relevant and interesting to the sampling community.

**Broader Impact Concerns:**

Not related

**Claims And Evidence:**

No

**Claims Explanation:**

I think that some of the claims need to be more supported by experimental results. Please, see the Strengths and Weaknesses section below.

**Requested Changes:**

**Strengths:**

1. The theoretical introduction is very clear and unifies the existing literature. Moreover, the theoretical foundations are presented in a very didactic way. The mathematical formulations are technically sound, and correct.
2. The decision to decouple the statistical inference error from the neural network learning error using highly controllable synthetic benchmarks is the good choice.
3. The proposed deterministic transport maps, particularly when combined with the Hutchinson trace estimator to bypass explicit Hessian computation, represent a novel and practical contribution.

**Weaknesses:**
1. While the controlled Gaussian mixtures are excellent for isolating specific failure modes, the experimental setting is too modest to fully support claims regarding physical systems. The benchmark lacks standard, well-established multimodal densities heavily utilized in the sampling community (e.g., DoubleWell, ManyWell, or funnel distributions). Furthermore, the evaluation completely bypasses realistic molecular systems with complex correlations, such as Lennard-Jones potentials or alanine dipeptide. The information about the mentioned targets can be found in the following papers, e.g., [1], [2], [3], and [4].
2. The empirical study relies only on the Sliced Wasserstein Distance. While important, this metric does not convey the full picture of the sampling dynamics. The evaluation lacks metrics such as the Evidence Lower Bound (ELBO), Empirical Upper Bound (EUBO), and Maximum Mean Discrepancy (MMD), which are necessary for diagnosing whether a method exhibits mode-seeking or mode-covering tendencies.
3. The manuscript provides a theoretical introduction to Variance Exploding SDE samplers (VE-SDE) in the appendix but completely omits them from the empirical comparisons. Because the paper is largely experimental, contrasting VE with Variance Preserving (VP) paths is strictly necessary to support general claims about diffusion-based samplers.
4. The manuscript proposes the Hutchinson estimator to maintain computational efficiency but lacks a theoretical or empirical discussion on how the unbounded variance of this estimator might artificially depress the Effective Sample Size (ESS) in Sequential Monte Carlo (SMC). Additionally, the deterministic transport maps appear to inflict a severe computational overhead when deployed within the Replica Exchange (RE) framework.

**Additional questions:**

Please look at the Weaknesses part. In addition, I have a few questions:

1. Could you provide a discussion or empirical insight into the fixed-point convergence guarantees (Proposition 2) when the neural network score estimates exhibit exploding gradients or violate Lipschitz constraints near the boundaries of isolated modes?
2. Please clarify the variance-bias tradeoff of using the Hutchinson estimator, specifically addressing how its inherent variance impacts the scaling of the aMC algorithms in higher dimensions.

**References:**

[1] Blessing, Denis, et al. "Beyond ELBOs: A Large-Scale Evaluation of Variational Methods for Sampling." International Conference on Machine Learning. PMLR, 2024.

[2] Sendera, Marcin, et al. "Improved off-policy training of diffusion samplers." Advances in Neural Information Processing Systems 37 (2024): 81016-81045.

[3] Akhound-Sadegh, Tara, et al. "Iterated Denoising Energy Matching for Sampling from Boltzmann Densities." International Conference on Machine Learning. PMLR, 2024.

[4] Rissanen, Severi, et al. "Progressive Tempering Sampler with Diffusion." International Conference on Machine Learning. PMLR, 2025.

---

> ### Author Response · Authors · 2026-05-27
> **Answer to kpGx [1/2]**
>
> We thank the reviewer for their careful reading and constructive feedback. We particularly appreciate the detailed summary of the paper and the recognition that the work provides a comprehensive and controlled empirical study of DM-based annealed Monte Carlo methods. We agree that this type of controlled analysis is a valuable contribution to the sampling community, especially given that many of these limitations have remained underexplored.
>
> We now address the concerns regarding the level of experimental support.
>
> > While the controlled Gaussian mixtures are excellent for isolating specific failure modes, the experimental setting is too modest to fully support claims regarding physical systems […]
>
> We thank the reviewer for this remark. Regarding DoubleWell/ManyWell distributions, the main limitation is that in high dimension we do not have access to ground-truth samples (most works rely on rejection sampling), nor to target-specific evaluation metrics. In contrast, Gaussian mixtures can be made arbitrarily complex while still providing exact samples and interpretable diagnostics (e.g., mode weights), which we report extensively. Regarding the Funnel distribution, this is not multimodal and therefore does not stress the failure modes we study.
>
> More broadly, we agree that our setting is simpler than realistic molecular systems. However, as explicitly discussed in the “Conclusion & Limitations”, our goal is precisely to expose fundamental limitations in a controlled setting. Failing on these Gaussian mixtures strongly suggests failure on more complex molecular systems. This work is intended as a first step: identifying bottlenecks that must be addressed before scaling to realistic applications.
>
> > Furthermore, the evaluation completely bypasses realistic molecular systems […]
>
> We fully agree that molecular systems are ultimately the target applications. However, our objective here is diagnostic rather than competitive benchmarking. As emphasized throughout the paper (especially Sec. 6), we do not claim that current DM-BG methods work well in practice; rather, we show that they already fail in simple controlled settings, and we identify why.
> Overall, we believe our work is a first step to identify promising directions and undermine limiting directions for future large-scale molecular sampling benchmarks.
>
> > The empirical study relies only on the Sliced Wasserstein Distance…
>
> We respectfully disagree with this assessment. The Sliced Wasserstein distance provides a comprehensive view of sample quality and is widely used as a reliable metric for multi-modal distributions. It is at least as informative as MMD, which is itself highly sensitive to kernel bandwidth selection. We note that we did compute MMD across all experiments and deliberately chose Sliced Wasserstein as our primary metric precisely because it proved more reliable and consistent in our setting.
>
> Regarding ELBO and EUBO, we believe these metrics are not relevant here: they are primarily designed for the variational inference setting and cannot even be computed for several of the methods we consider. We do, however, now report log-normalizing constant estimates across all experiments and ablations, a metric that is conceptually related and directly meaningful for annealed sampling methods. As detailed in the global response, this new metric fully agrees with the Sliced Wasserstein distance across all settings, further confirming that our conclusions are not an artifact of the choice of evaluation metric.
>
> > The manuscript provides a theoretical introduction to VE-SDE but omits them from experiments [...]
> > Because the paper is largely experimental, contrasting VE with Variance Preserving (VP) paths is strictly necessary to support general claims about diffusion-based samplers.
>
> We also respectfully disagree that this affects our conclusions. The observed failure modes (in particular mode blindness of learned log-densities) are a property of the training objectives themselves and are largely agnostic to the choice of noise schedule. We did run experiments with both VP and VE schedules and observed no meaningful differences in any of the reported metrics. Because the results were essentially unchanged, we retained VP as the default noising schedule, in line with the broader diffusion-based sampling literature and to avoid overloading the manuscript. To guarantee theoretical completeness, we nonetheless provide all VE-specific formulas in Appendix B.3, making it straightforward to reproduce our results under this schedule.

---

> > ### Author Response · Authors · 2026-05-27
> > **Answer to kpGx [2/2]**
> >
> > > The manuscript proposes the Hutchinson estimator [...] but lacks discussion on its unbounded variance
> >
> > We thank the reviewer for raising this point. In the revised manuscript, we have addressed this concern in a much more principled way. All deterministic methods are now embedded within the penalty correction of Ceperley & Dewing (1999), recalled in Appendix C.1. This is a rigorous statistical recipe that provably corrects for the stochasticity of the Hutchinson estimator: deterministic AIS/SMC weights are rendered exactly unbiased and consistent, and deterministic RE is guaranteed to admit the correct invariant distribution, regardless of the variance of the log-determinant estimator. The companion ablations in Appendix D.4 further confirm that the penalty-corrected estimators behave as expected across all targets, with bias and variance well within the precision relevant for sampling. We therefore believe this concern is now completely resolved.
> >
> > > Additionally, the deterministic transport maps appear to inflict a severe computational overhead in RE [...]
> >
> > Regarding the computational overhead of deterministic transport maps in RE, we acknowledge that they do introduce additional cost relative to the standard swap. However, as also noted by reviewer 4c1c, in a fully parallel setting this overhead largely vanishes, since the fixed-point iterations and log-determinant computations can be parallelized across chains. Our original statement referred to a constrained parallel setting, and we have clarified this in the revised manuscript. Moreover, as shown in Figure 3, deterministic RE achieves strong performance at small $K$ where the per-swap cost is lower, unlike stochastic methods which require large $K$ to be competitive, keeping the overall computational budget comparable across methods.
> >
> > > Could you provide insight into fixed-point convergence guarantees when score estimates violate Lipschitz constraints [...]
> >
> > Neural networks are Lipschitz functions (with finite constants), so the assumptions underlying Proposition 2 are satisfied in practice. Furthermore, we point the reviewer to the ablation in Appendix D.4 (Figures 18 and 19), where we directly measure the mutual invertibility error of the Implicit Midpoint maps across all targets, noise levels, and both VP and VE schedules. The error decreases rapidly with the number of fixed-point iterations and reaches numerical precision for $n_\text{iter} \geq 6$ across all settings. This provides strong empirical evidence that the convergence guarantees of Proposition 2 are robust in practice.
> >
> > > Please clarify the variance-bias tradeoff of the Hutchinson estimator [...]
> >
> > This tradeoff is now addressed both theoretically and empirically. On the theoretical side, the penalty correction of Ceperley & Dewing (1999) (Appendix C.1) provably absorbs the variance of the Hutchinson estimator, ensuring unbiased and consistent estimators regardless of dimension. On the empirical side, the ablations in Appendix D.4 (Figures 22 and 23) show that even modest values of $n_\text{hutch}$ yield bias and variance well within the precision relevant for sampling in high-dimension. Together, these provide strong reassurance that the estimator's variance does not hinder scalability in practice.

---

### Author Response · Authors · 2026-05-27
**Global answer**

We thank reviewers kpGx, 4c1c and 135i for their careful, thoughtful, and constructive feedback, as well as for their overall positive assessment of the paper.

We are particularly grateful for the detailed reading and the insightful comments, which helped us clarify the scope of our claims and strengthen the empirical section. We note that several key contributions were recognized across reviews: the paper's value as a careful empirical study with a clean idealized vs. realistic split, the novelty and interest of the deterministic transition kernel, and the usefulness of the work as a reference for the community. We address all concerns raised in detail below and summarize here the main changes made to the manuscript in response.

**Streamlined experimental protocol.** We removed redundant experimental configurations and focused on 3 carefully selected representative cases for each of TwoModes and ManyModes. This allowed us to run significantly more experiments and ablations within the same computational budget, while also substantially cleaning up the appendix. The revised manuscript is 10 pages shorter than the original despite containing considerably more content.

**Pushing first-order kernels further.** We investigated additional unexplored first-order kernels and now default to the DDPM scheme, shown in Appendix D.4 to outperform the EI scheme used in prior work. This slightly closes the gap with higher-order methods but does not affect any of the paper's conclusions on their fundamental limitations.

**Annealing schedule optimization.** Following reviewer 4c1c's recommendation, we precompute the $\Lambda$-optimal schedule using the tractability of our Gaussian mixture targets and provide a detailed ablation. This does not improve DM-based samplers and degrades performance in the learned setting. We do adopt it for tempering, giving it its most favorable setting, yet diffusion paths remain decisively superior.

**Log-normalizing constant estimation.** Following reviewer 135i's request, we now report log $\mathcal{Z}$ across all targets and ablations as an additional metric. It consistently conveys the same messages as the existing ones, and most strikingly provides undeniable quantitative evidence of the superiority of diffusion paths over tempering.

**Theoretical guarantees for deterministic transitions.** This is the most substantial improvement. We now use the lower-variance Hutch++ estimator and embed all deterministic methods within the penalty correction of Ceperley & Dewing (1999), a standard statistical-physics recipe recalled in Appendix C.1. This guarantees that deterministic AIS/SMC estimators are unbiased and consistent, and that deterministic RE admits the correct invariant distribution. Appendix D.4 is significantly clarified to ablate all components.

**Strengthening the realistic setting.** We extended the hyperparameter search, trained in continuous time, and added the DiffCLF objective, yielding only marginal improvements. Most importantly, we added a semi-ideal experiment (exact log-densities, learned scores) whose near-ideal performance directly identifies log-density estimation as the primary bottleneck, as unanimously requested by reviewers.

All changes are displayed in blue in the revised manuscript. We believe that these global changes, together with the local ones addressed in individual responses, exhaustively cover all concerns raised by the reviewers (and even several that were not explicitly raised) and significantly improve the overall quality of the manuscript.

---

### Decision · Action_Editor_q7ku · 2026-07-06

**Recommendation:** Accept with minor revision

**Additional Comments:**

While the reviewers appreciate the additional clarifications and experiments provided in the rebuttal, they note that the current empirical scope remains a limitation. Please validate the core findings on at least one standard community benchmark beyond Gaussian mixtures.

**Audience:**

Yes

**Audience Explanation:**

The findings are of interest to TMLR readers working on diffusion models, sampling problems, and Boltzmann generators.

**Claims And Evidence:**

Yes

**Claims Explanation:**

The claims made in the submission are supported by empirical evaluations and rigorous mathematical derivations.